# Histopathology-Genomics Multi-modal Structural Representation Learning for Data-Efficient Precision Oncology

**Kun Wu**[1,2], **Zhiguo Jiang**[3], **Xinyu Zhu**[2], **Jun Shi**[4], **Yushan Zheng**[1] *

[1]School of Engineering Medicine, Beihang University, Beijing, China.
[2]Image Processing Center, School of Astronautics, Beihang University, Beijing, China.
[3]Tianmushan Laboratory, Hangzhou, China.
[4]School of Software, Hefei University of Technology, Hefei, China.

## Abstract

Fusing histopathology images and genomics data with deep learning has significantly advanced precision oncology. However, genomics data is often missing due to its high acquisition cost and complexity in real-world clinical scenarios. Existing solutions aim to reconstruct genomics data from histopathology images. Nevertheless, these methods typically relied only on individual case and overlooked the potential relationships among cases. Additionally, they failed to take advantage of the authentic genomics data of diagnostically related cases that are accessible from training for inference. In this work, we propose a novel Multi-modal Structural Representation Learning (MSRL) framework for data-efficient precision oncology. We pre-train a histopathology-genomics multi-modal representation graph adopting Graph Structure Learning (GSL) to construct inter-case relevance based on the data inherently. During the fine-tuning stage, we dynamically capture structural relevance between the training cases and the acquired authentic cases for precise prediction. MSRL leverages prior inter-case associations and authentic genomics data from diagnosed cases based on the graph, which contributes to effective inference based on the single histopathology image modality. We evaluated MSRL on public TCGA datasets with 7,263 cases across various tasks, including survival prediction, cancer grading, and gene mutation prediction. The results demonstrate that MSRL significantly outperforms existing missing-genomics generation approaches with improvements of 1.44% to 3.12% in C-Index on survival prediction tasks and achieves comparable performance to multi-modal fusion methods. The code and data are available at https://github.com/WkEEn/MSRL.

## 1 Introduction

Histopathology whole slide images (WSIs) describe detailed visual information of morphology features, cellular organization, and phenotypic characteristics, which are considered the gold standard for the assessment of cancer (Shmatko et al., 2022; Dimitriou et al., 2019; Hegde et al., 2019; Shamai et al., 2022; Kather et al., 2019). Genomics data provide quantitative molecular characteristics of the microenvironment, which is significant for precision medicine and targeted therapies (Zhang et al., 2021; Eraslan et al., 2019; Zhou et al., 2019; Jaume et al., 2024a; Kopp et al., 2020; Li et al., 2022b). Recently, an increasing number of studies on histopathology-genomics multi-modal learning have shown superior capabilities in precision oncology by providing a more personalized representation that comprehensively reflects the case's status (Chen et al., 2020a; 2021; Zhou & Chen, 2023; Jaume et al., 2024b). Fusing morphological information from histology and molecular information from genomics data is the prevalent paradigm of multi-modal algorithms (Vale-Silva & Rohr, 2021; Mobadersany et al., 2018; Weng et al., 2019) as shown in Figure 1(a). However, the acquisition cost of genomics data is still very high, resulting in the lack of complete paired histopathology-genomics data in many practical inferences.

---

*Corresponding author: Yushan Zheng (e-mail: yszheng@buaa.edu.cn).

Figure 1: Differences between proposed MSRL and previous methods, where (a) shows multi-modal fusion methods, (b) and (c) depict WSI-based genomics reconstruction and distillation methods, where dashed lines indicate components used only during training and not involved in inference, and (d) presents our proposed MSRL, which incorporates authentic genomics data from diagnosed cases to assist gene reconstruction and inference.

To address this challenge, some researchers focus on utilizing the histopathology data to reconstruct the missing genomics features and then apply multi-modal framework for prediction. These methods typically follow two paradigms: (1) constructing auxiliary tasks during training to reconstruct existing genomics data and encouraging the model to build the associations between WSI and genomics features through reconstruction loss (Wu et al., 2024; Wang et al., 2025) or distillation loss (Xu et al., 2025) as shown in Figure 1(b); or (2) adopting generative approaches that condition on histopathology data inputs to synthesize missing genomics features (Zhou et al., 2025) as shown in Figure 1(c). These approaches partially addressed the problem of missing the modality during inference. However, these methods suffer from the following two limitations: (1) **Ignoring potential relevance among cases.** Reconstruction-based approaches typically rely solely on the histopathology image of the individual case to predict genomics features and focused on intra-case multi-modal alignment while overlooking crucial inter-case relevance for cancer diagnosis. (2) **Underutilization of available authentic genomics data.** During inference, the genomics modality is entirely synthesized by these methods, and they neglect the authentic genomics data of diagnostically related cases that are accessible from training data. This limits the authenticity and informativeness of the generated features.

In the paper, we propose a novel histopathology-genomics Multi-modal Structural Representation Learning (MSRL) framework for data-efficient precision oncology. We adopt graph structure learning to pre-train a multi-modal representation graph to construct inter-case relevance based on the data inherently and enable dynamic graph construction for downstream tasks to achieve inference with the single histopathology image modality. The data-driven graph pre-training captures relevance among diagnostic cases. We introduce authentic data as auxiliary structural guidance to exploit the information of genomics data in missing modality scenarios during inference. Furthermore, we fine-tune the WSI encoder with the guidance of the multi-modal representation graph and enhance its ability to capture fine-grained morphology characteristics for data-efficient prediction with the single image modality. The main contributions of this work are summarized as follows:

1. We propose a novel **Multi-modal Structural Representation Learning (MSRL)** framework that effectively addresses cancer diagnosis tasks in genomics data missing scenarios. **MSRL** facilitates the reconstruction of missing genomics features by leveraging multi-modal relevance among cases and jointly enhance the representation capability of the WSI encoder, which significantly promotes data-efficient precision oncology.

2. We pre-train a multi-modal representation graph via self-supervised graph structure learning on a large-scale pan-cancer histopathology–genomics dataset. The pre-trained graph captures structural relevance among cases and is further utilized to guide dynamic graph construction during downstream inference, which contributes to effective prediction with the single WSIs modality. The framework leverages authentic genomics data as auxiliary supervision to reconstruct the missing modality data for inference, effectively maximizing the utility of available molecular information.

3. We evaluated our method on publicly available The Cancer Genome Atlas (TCGA) datasets for survival prediction and four precision diagnosis tasks. Experimental results show that our approach significantly outperforms existing methods designed for missing modality scenarios, and achieves comparable performance with multi-modal fusion methods. Moreover, our method demonstrates superior generalization capabilities on the external CPTAC datasets, which highlights its potential for real-world clinical applications.

## 2 RELATED WORK

**Fusion-based Multi-modal methods** Multi-modal approaches that integrate WSIs with genomics data offer a more comprehensive and objective representation for cancer diagnosis. Chen et al. (2021) proposed the MCAT framework to learn how histology patches attend to genes for survival prediction. Zhou & Chen (2023) utilized two parallel encoder-decoder structures to align WSI and genomics representations. Jaume et al. (2024b) tokenized genomics data and used a memory efficient multi-modal Transformer to model interactions between pathway and histology patch tokens for prognosis. Xu & Chen (2023) and Zhang et al. (2024) leveraged optimal transport and prototypical information bottleneck theory, respectively, to fully enable the transfer and integration of information between the two modalities.

**Multi-modal methods with missing modality** Existing multi-modal fusion methods generally rely on the assumption that complete histopathology–genomics data pairs are available. However, acquiring genomics data is often costly and impractical, which results in missing modalities in real-world scenarios. To address this limitation, recent studies have explored methods for reconstructing paired genomics features from available histopathology images. One approach involves an auxiliary task during training to reconstruct the missing genomics data. Wang et al. (2024b; 2025) were the first to propose using initialized prompts to replace missing genomics data to enable interaction with histopathology features through the cross-attention module. The prompts reconstruct the missing modality representation guided by available genomics data during training, and then these learned prompts are subsequently applied for inference. Wu et al. (2024) introduced a proxy gene-reconstruction branch to guide the WSI encoder to extract genomics-related features from image modality. Zhou et al. (2025) proposed a conditional Latent Differentiation VAE (LD-VAE) to generate the missing genomics data from WSIs. Xu et al. (2025) employed a learnable prompt to substitute for the missing gene data and leveraged the powerful LLM (Large Language Model) to distill prognostic knowledge into the prompt learning process. Jin et al. (2025) ensured the decoder's ability to complete the multi-modal information by integrating missing data reconstruction with knowledge distillation. These methods rely on individual case data to reconstruct genomics features, while overlooking inter-case relevance that could more effectively guide the recovery of missing modality. Moreover, during inference, they failed to leverage the auxiliary information of available genomics data, despite evidence (Shu et al., 2024) showing that existing database knowledge can significantly enhance cancer diagnosis. As a result, current approaches struggle to faithfully reconstruct high-density genomics information for effective inference.

**Graph Structure Learning (GSL)** Graph Neural Networks (GNNs) have demonstrated strong performance in modeling structural dependencies and supporting downstream inference. A typical GNN takes as input a set of nodes and edges, where the edge set encodes structural relationships between nodes. However, in many real-world scenarios, such prior structures are difficult to define or are contaminated by noise, for example, the relationships between cancer patients are not explicitly known. Such unreliable structures limit the representational power of GNNs.

To address this challenge, data-driven *Graph Structure Learning (GSL)* (Franceschi et al., 2019; Liu et al., 2022a) has emerged as an effective paradigm. The core idea of GSL is to incorporate the learning of graph structure into the task-driven learning process. Unlike traditional GNNs with fixed edge sets, GSL dynamically learns the edge structure during training based on node features and constructs an optimized graph topology. Li et al. (2023) summarized a general GSL pipeline: a *Graph Learner* takes an initial graph $\mathcal{G} = \{\mathbf{X}, \mathbf{A}\}$ as input—where $\mathbf{A}$ is the adjacency matrix and $\mathbf{X}$ is the node feature matrix—and outputs a refined graph $\mathcal{G}^r = \{\mathbf{X}, \mathbf{A}^r\}$, which is then fed into a GNN for representation learning and task inference. Notably, the original adjacency matrix $\mathbf{A}$ is not required; the graph learner can construct $\mathbf{A}^r$ adaptively from the node features. Liu et al. (2022b), Li et al. (2022a), and Zhao et al. (2023) introduced contrastive learning to enable unsupervised GSL, eliminating the need for task labels and allowing the model to mine latent data correlations. Shen et al. (2024) further applied unsupervised GSL to multiplex graphs, offering a new perspective for structure learning in multi-modal learning.

## 3 METHODS

Our proposed framework consists of two stages: Multi-modal Structural Representation Pre-training and Fine-tuning. *In the first stage, as shown in Figure 2.(a)*, we leverage TCGA pan-cancer cases with

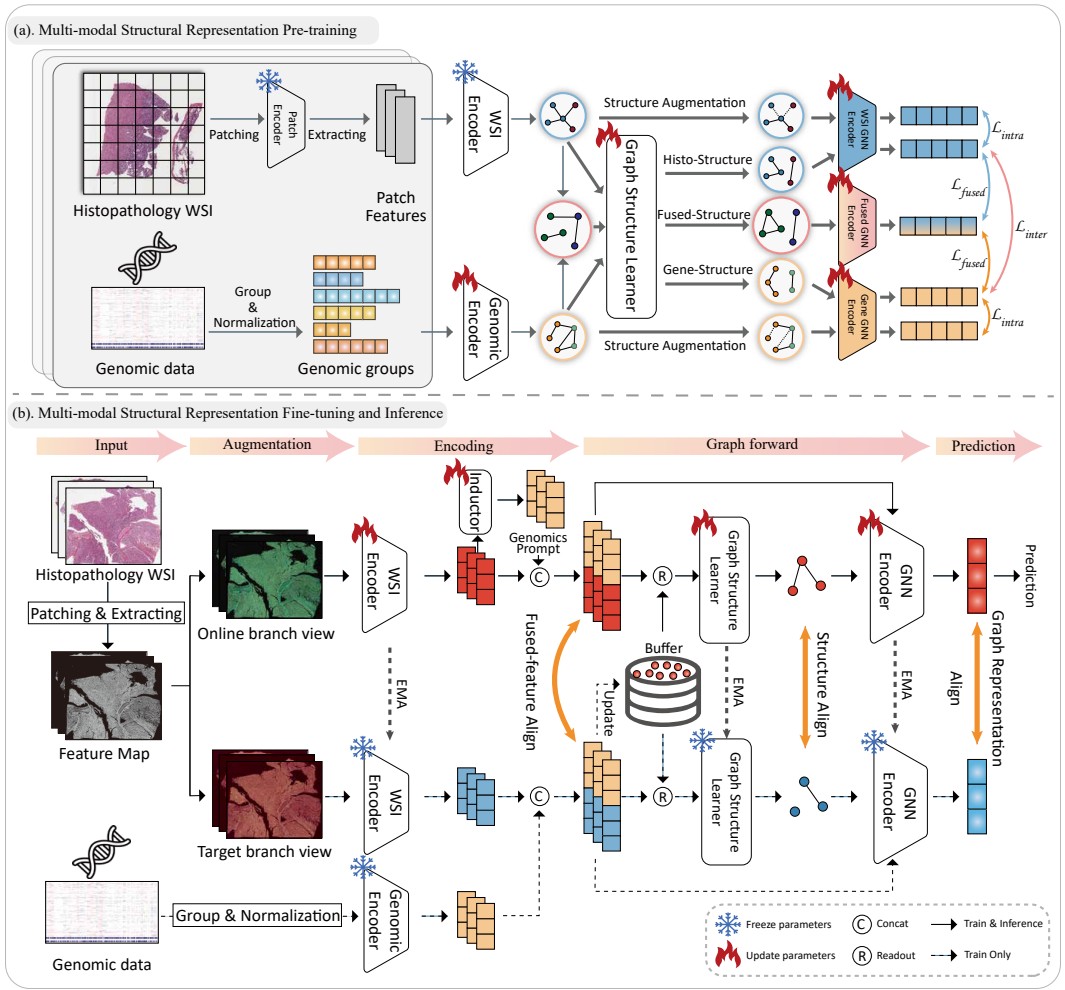

Figure 2: The framework of proposed MSRL, where (a) illustrates the multi-modal structural representation pre-training with TCGA pan-cancer dataset and (b) shows the fine-tuning to dynamically construct graphs for current cases and cases from the buffer, which leverages inter-case relevance and available authentic genomics data for inference.

complete multi-modal data for self-supervised pre-training. This process aligns the representation spaces of WSI and genomics and simultaneously establishes multi-modal inter-case relevance. The abundance of pan-cancer data facilitates Graph Structure Learning (GSL) in constructing more generalized relevance patterns. Furthermore, the aligned genomics encoder can be more effectively applied to downstream tasks. *In the Fine-tuning stage, as shown in Figure 2.(b)*, we apply the pre-trained GSL to dynamically construct graphs for the current cases in the mini-batch and previous cases from the training buffer. This leverages inter-case relevance and utilizes available authentic genomics data as auxiliary information for effective fine-tuning. Based on the above, we propose the Multi-modal Structural Representation Learning (MSRL) that jointly learn multi-modal case representations and the graph relevance among cases.

## 3.1 FORMULATION

**Histopathology Representation** For the $k$-th case, the WSI is cropped into $m^k$ non-overlapping patches after segmenting the foreground tissue region. We leverage a powerful pathology foundation model for patch-level feature extraction and WSI encoder initialization. Specifically, the ViT-giant pretrained on GigaPath (Xu et al., 2024) is used to extract patch features, denoted as $\mathbf{X}_I^k \in \mathbb{R}^{m^k \times d_p}$, where $d_p$ is the dimension of the patch feature. The Transformer-based LongNet (Ding et al., 2023) pretrained on GigaPath is used to initialize the WSI encoder $\phi_{\mathbf{H}}(\cdot)$. The WSI representation is the [*CLS*] token of $\phi_{\mathbf{H}}(\mathbf{X}_I^k)$ formulated as $\mathbf{h}^k \in \mathbb{R}^{d_h}$, where $d_h$ is the dimension of WSI feature.

---

**Algorithm 1:** *Graph Structure Learner (GSL)*

---

**Input:** The node features $\mathbf{X} \in \mathbb{R}^{K \times d_x}$;
**Output:** Refined adjacency matrix $\mathbf{A}^r$

1   $\mathbf{E}^{(0)} \leftarrow \mathbf{X}$, and the learner contains $L$ layers;
2   **for** $l = 1$ **to** $L$ **do**
3      **for** $i = 1$ **to** $K$ **do**
4         $z_i^{(l)} = \mathbf{E}_i^{(l-1)} \odot \omega^{(l)}$, where $\odot$ is Hadamard product and $\omega$ is the learnable weight vector;
5      $\mathbf{E}^{(l)} = \sigma([z_1^{(l)}, \ldots, z_K^{(l)}]^T)$, where $\sigma$ is the nonlinear activation function;
6   $\mathbf{A}^r = post\_processing(\mathbf{S}); \mathbf{S} \leftarrow \mathbf{E}^{(L)}(\mathbf{E}^{(L)})^T;$ `// The post-processing steps are` `described in Appendix D.2.`
7   **return** *Refined adjacency matrix* $\mathbf{A}^r$

---

**Genomics Representation** We collect bulk gene expression profiles for each case, where each profile contains thousands of expression values. Following prior work (Chen et al., 2021; Zhou & Chen, 2023; Jaume et al., 2024b) and the biological functional grouping of genes, we divide the genes into $n$ groups (Liberzon et al., 2015). Each group is encoded using an independent SNN network (Klambauer et al., 2017), denoted as $\mathbf{X}_G^k \in \mathbb{R}^{n \times d_g}$, where $d_g$ is the dimension of each grouped gene feature. The genomics feature of the case is the mean pooling of these groups formulated as $\mathbf{g}^k \in \mathbb{R}^{d_g}$, where $d_g = d_h = d$.

**Graph Representation** We initialize three graphs: $\mathcal{G}_\mathbf{H} = \{\mathbf{H}, \mathbf{A_H}\}$, $\mathcal{G}_\mathbf{G} = \{\mathbf{G}, \mathbf{A_G}\}$, and $\mathcal{G}_\mathbf{F} = \{\mathbf{F}, \mathbf{A_F}\}$, representing the histopathology graph, gene graph, and fused multi-modal graph, respectively. Specifically, $\mathbf{H} = \{\mathbf{h}^k\}_{k=1}^K$ and the adjacency matrix $\mathbf{A_H} \in [0,1]^{K \times K}$ is initialized by applying K-Nearest Neighbors (KNN) clustering on $\mathbf{H}$, where $K$ is the total number of cases. The graphs $\mathcal{G}_\mathbf{G}$ and $\mathcal{G}_\mathbf{F}$ are constructed in the same manner. We fuse multi-modal features with a simple strategy: $\mathbf{f}^k = concatenate(\mathbf{g}^k, \mathbf{h}^k) \in \mathbb{R}^{2d}$, in order to evaluate the contribution of case-level structural representation learning rather than complex fusion techniques.

## 3.2 MULTI-MODAL STRUCTURAL REPRESENTATION PRE-TRAINING

We leverage the pan-cancer data and introduce self-supervised GSL to achieve discriminative and generalized multi-modal structural representation learning, as illustrated in Figure.2.(a). The core component of GSL is a parameterized graph learner that adaptively infers the optimal refined structure from the input node features (Li et al., 2023; Liu et al., 2022b; Li et al., 2022a; Zhao et al., 2023; Shen et al., 2024), which takes the initial graph as input. The definition of $GSL$ are described in Algorithm 1. Subsequently, we detail the role of GSL during pre-training.

**Intra-modality GSL** The diagnostic relevance among cases is inherently reflected within the identical modality. We adopt a contrastive learning paradigm (Chen et al., 2020b; Tian et al., 2020) to uncover intra-modality relevance among cases. For the pathology graph $\mathcal{G}_\mathbf{H} = \{\mathbf{H}, \mathbf{A_H}\}$, we input the node features into the pathology-specific graph structure learner and obtain the refined pathology-modality adjacency matrix, denoted as $\mathbf{A}_\mathbf{H}^r = GSL_\mathbf{H}(\mathbf{H})$. Additionally, we apply random edge dropout and edge addition to generate an augmented adjacency matrix, denoted as $\mathbf{A}_\mathbf{H}^{aug}$. Then, both views are passed through the pathology-specific Graph Convolutional Network (GCN) encoder (Kipf & Welling, 2016), which is formulated as follows

$$\mathbf{Z_H} = GCN_\mathbf{H}(\mathbf{A}_\mathbf{H}^r, \mathbf{H}) \in \mathbb{R}^{K \times d_f}, \mathbf{Z}_\mathbf{H}^{aug} = GCN_\mathbf{H}(\mathbf{A}_\mathbf{H}^{aug}, \mathbf{H}) \in \mathbb{R}^{K \times d_f}, \tag{1}$$

The InfoNCE loss (Liang et al., 2023) is employed as the intra-modal constraint to maximize the agreement between the node representations of the same case and minimize the similarity between those of different cases within each modality. The loss is defined as follows

$$\mathcal{L}_{InfoNCE}(\mathbf{Z_H}; \mathbf{Z}_\mathbf{H}^{aug}) = -\sum_{k=1}^K \log \frac{\exp(sim(z_k, z_k^{aug})/\tau)}{\sum_{i=1}^K \exp(sim(z_k, z_i^{aug})/\tau)}, \tag{2}$$

where $sim(\cdot, \cdot)$ denotes the cosine similarity between two representations, and $\tau$ is a temperature hyperparameter. We accordingly obtain the structure construction for the genomics modality $\mathbf{Z_G}$ and $\mathbf{Z}_\mathbf{G}^{aug}$, and the intra-modality structure learning loss function formulated as follows

$$\mathcal{L}_{intra} = \frac{1}{2}(\mathcal{L}_{InfoNCE}(\mathbf{Z_H}; \mathbf{Z}_\mathbf{H}^{aug}) + \mathcal{L}_{InfoNCE}(\mathbf{Z_G}; \mathbf{Z}_\mathbf{G}^{aug})). \tag{3}$$

**Inter-modality GSL** Histopathology and genomics describe a case's status from different perspectives. We introduce inter-modality GSL to align their respective structures within a unified representation space. The InfoNCE loss is utilized to maximize the agreement between the different modal representations of the same case, which is formulated as follows

$$\mathcal{L}_{inter} = \frac{1}{2}(\mathcal{L}_{InfoNCE}(\mathbf{Z_H}; \mathbf{Z_G}) + \mathcal{L}_{InfoNCE}(\mathbf{Z_G}; \mathbf{Z_H})). \tag{4}$$

**Fused-modality GSL** Multi-modal fusion representations provide a more comprehensive and objective description of case information. Therefore, we further introduce a structural constraint to ensure that the fused representations preserve pathology-specific and genomics-specific characteristics. The multi-modal structure learner is employed to refine the adjacency matrix, denoted as $\mathbf{A_F^r} = GSL_{\mathbf{F}}(\mathbf{F})$. Consequently, we obtain the graph-encoded multi-modal representations denoted as $\mathbf{Z_F} = GCN_{\mathbf{F}}(\mathbf{A_F^r}, \mathbf{F})$. We then employ the InfoNCE loss to maximize the agreement between the uni-modal representations and the fused multi-modal representation within the same case, which is formulated as follows

$$\mathcal{L}_{fused} = \frac{1}{2}(\mathcal{L}_{InfoNCE}(\mathbf{Z_F}; \mathbf{Z_H}) + \mathcal{L}_{InfoNCE}(\mathbf{Z_F}; \mathbf{Z_G})). \tag{5}$$

Based on the above components, the overall pre-training loss is formulated as $\mathcal{L}_{gsl} = \mathcal{L}_{intra} + \mathcal{L}_{inter} + \mathcal{L}_{fused}$. The multi-perspective constraints facilitate training robust multi-modal graph structure learners and construct an effective multi-modal structural representation graph.

## 3.3 MULTI-MODAL STRUCTURAL REPRESENTATION FINE-TUNE AND INFERENCE

In this stage, we construct an online-target dual-branch architecture along with a buffer mechanism, as depicted in Figure 2(b). The online branch performs efficient task prediction using only WSI data. The target branch, which is only activated during training, receives complete multi-modal data and guides the WSI uni-modal input online branch to learn inter-case relevance representations, thereby completing the missing genomics information. Furthermore, the buffer stores the complete multi-modal representations of the training samples, serving as a key component for incorporating authentic data to construct inter-case relevance. Subsequently, we detail the specifics of each component.

**The Buffer mechanism** In the missing gene inference scenario, we define the training dataset as $\mathcal{D}_{train} = \{\mathbf{X}_G^s, \mathbf{X}_I^s, y^s\}_{s=1}^{S_1}$, and the testing dataset as $\mathcal{D}_{test} = \{\mathbf{X}_I^s, y^s\}_{s=1}^{S_2}$, where $S_1$ and $S_2$ denote the number of training and test cases, respectively, and $y$ is the task label. The feature buffer is initialized using the available authentic data of training cases, denoted as $\mathcal{D}_{buffer} = \{\mathbf{f}^s\}_{s=1}^{S_1} = \{concatenate(\phi_{\mathbf{G}}(\mathbf{X}_G^s), \phi_{\mathbf{H}}(\mathbf{X}_I^s))\}_{s=1}^{S_1}$, where $\phi_{\mathbf{G}}$ is the pre-trained genomics encoder of first stage and $\phi_{\mathbf{H}}$ is the WSI encoder of foundation model. The learned features of the target branch from authentic multi-modal data are utilized to update the buffer with the First-In-First-Out (FIFO) strategy, which ensures the buffer representations remain up-to-date.

**Dual-branch training with complete modalities** The target branch utilizes the relevance derived from complete multi-modal training data to guide the online branch in learning the missing genomics information. Inspired by prior work (Chen & Lu, 2023), we employ mixup-based augmentations to generate distinct input views for the online and target branches, respectively. We design the **Inductor** module to address the absence of genomics data in the online branch, as illustrated in the encoding phase of Figure 2(b). Specifically, the **Inductor** module adopts the same SNN network architecture (Klambauer et al., 2017) as the genomics encoder, but takes the WSI representation as input and outputs a genomics prompt that serves as a placeholder for the missing data. This prompt, combined with the WSI representation, constitutes the online branch's multi-modal representation. The missing gene information is subsequently completed through the construction of multi-modal relevance during the Graph Forward (GF) process, which is presented in Algorithm 2.I

**Hierarchical Alignment** We introduce hierarchical losses to jointly optimize multi-modal representation learning and structure learning, as illustrated in Algorithm 2.II. Firstly, we impose constraints on the features derived both preceding and succeeding the graph learning phases, which ensures the stable and comprehensive learning of authentic multi-modal data representations by the online branch and is formulated as follows

$$\mathcal{L}_{f\_align} = \mathcal{L}_{InfoNCE}(\mathbf{f}, \hat{\mathbf{f}}), \ \mathcal{L}_{g\_align} = \mathcal{L}_{InfoNCE}(\mathbf{Z}, \hat{\mathbf{Z}}), \tag{6}$$

---

**Algorithm 2:** Multi-modal Structural Representation Fine-tune and Inference

---

1    **I.Function** $Graph\ Forward\ (GF)$**:**
     **Input:** Slide features $\mathbf{h}$, Gene features $\mathbf{g}$, Buffer $D_{\text{buffer}}$
     **Output:** $\mathbf{f}, \mathbf{A}^r, \mathbf{Z}$
2      $\mathbf{f} \leftarrow concatenate(\mathbf{g}, \mathbf{h})$ ;
3      $\mathbf{F} \leftarrow Readout(D_{\text{buffer}}, \mathbf{f})$ `// Combining features of the current cases with`
          `the others in the buffer.`
4      $\mathbf{Z} \leftarrow GCN(\mathbf{A}^r, \mathbf{F}); \mathbf{A}^r \leftarrow GSL(\mathbf{F})$ `// Definition of GSL as shown in`
          `Algorithm 1.`
5      **return** $\mathbf{f}, \mathbf{A}^r, \mathbf{Z}$;

6    **II.Fine-tuning procedure with complete modalities:**
7      $\phi_H \leftarrow$ pre-trained WSI encoder,     $\phi_G \leftarrow$ pre-trained Gene encoder;
8      `// Encoding`
9      $\mathbf{g} \leftarrow Inductor(\mathbf{h}), \quad \mathbf{h} \leftarrow \phi_H(\mathbf{X}_I^{\text{aug}})$ ;                   `// Online branch input`
10      $\hat{\mathbf{g}} \leftarrow \phi_G(\mathbf{X}_G), \quad \hat{\mathbf{h}} \leftarrow \phi_H(\hat{\mathbf{X}}_I^{\text{aug}})$ ;                   `// Target branch input`
11      `// Graph Forward`
12      $\mathbf{f}, \mathbf{A}^r, \mathbf{Z} \leftarrow GF_{\text{online}}(\mathbf{h}, \mathbf{g}, D_{\text{buffer}})$;
13      $\hat{\mathbf{f}}, \hat{\mathbf{A}}^r, \hat{\mathbf{Z}} \leftarrow GF_{\text{target}}(\hat{\mathbf{h}}, \hat{\mathbf{g}}, D_{\text{buffer}})$;
14      $D_{\text{buffer}} \leftarrow update(D_{\text{buffer}}, \hat{\mathbf{f}})$;
15      Calculate losses: $\mathcal{L}_{f\_align}(\mathbf{f}, \hat{\mathbf{f}}), \mathcal{L}_{g\_align}(\mathbf{Z}, \hat{\mathbf{Z}}), \mathcal{L}_{s\_align}(\mathbf{A}^r, \hat{\mathbf{A}}^r), \mathcal{L}_{\text{task}}(\mathbf{Z}, y)$;

16    **III.Inference procedure with WSI modality:**
17      $\mathbf{g} \leftarrow Inductor(\mathbf{h}), \quad \mathbf{h} \leftarrow \phi_H(\mathbf{X}_I)$;
18      $\mathbf{f}, \mathbf{A}^r, \mathbf{Z} \leftarrow GF_{\text{online}}(\mathbf{h}, \mathbf{g}, D_{\text{buffer}})$;
19      Prediction with $\mathbf{Z}$;

---

Moreover, we adopt sparsity-balanced binary cross-entropy (BCE) loss (Duan et al., 2024) to align graph structures between $\mathbf{A}^r$ and $\hat{\mathbf{A}}^r$. For the target graph, we assume there are $c_0$ zero and $c_1$ non-zero elements in $\hat{\mathbf{A}}^r$, where $c_0 \gg c_1$. To balance the loss between zeros and non-zeros, we apply scaling factors on each element loss as follows

$$\mathcal{L}_{s\_align} = BCE(\mathbf{A}^r, \hat{\mathbf{A}}^r) = \alpha_0 \sum_{i=1}^{c_0} \mathcal{L}_i^0 + \alpha_1 \sum_{j=1}^{c_1} \mathcal{L}_j^1, \quad \alpha_0 = \frac{c_0 + c_1}{2c_0}, \; \alpha_1 = \frac{c_0 + c_1}{2c_1}, \quad (7)$$

where $\mathcal{L}^0$ and $\mathcal{L}^1$ denote the loss calculated for zero and non-zero elements in $\hat{\mathbf{A}}^r$. The joint loss calculated in the fine-tune stage is formulated as $\mathcal{L}_{fine\_tune} = \mathcal{L}_{f\_align} + \mathcal{L}_{g\_align} + \mathcal{L}_{s\_align} + \mathcal{L}_{task}$. The loss function is used exclusively to update the online branch, while the target branch is updated using an EMA (He et al., 2020) strategy to prevent representation collapse.

**Inference and Prediction with WSI modality** We apply the online branch for task inference as illustrated in Algorithm 2.III, where the trained Inductor is capable of estimating the missing genomics data. The integration of authentic genomics data from the buffer assists in completion of the missing genomics and facilitates data-efficient inference based on the single WSI modality.

## 4   EXPERIMENTS

### 4.1   DATASETS AND SETTINGS

**Datasets:** We collect and curate n=7,263 cases with WSI-gene pairs from the TCGA pan-cancer dataset containing 32 cancer subtypes across 12 primary tumor sites. We construct a pre-training dataset including 6,361 cases without any diagnostic information for MSRL first-stage. For fine-tuning, we evaluate the proposed MSRL framework on six TCGA cohorts and two external CPTAC cohorts. Details of fine-tuning datasets and prep-rocessing of data are provided in the Appendix C.

**Experimental Settings:** All test data are excluded from the first-stage pre-training. We implement two model variants: $MSRL_H$ excludes genomics data during fine-tuning, with no genomics input to the target branch, no Inductor in the online branch, and the buffer contains WSI embeddings rather than fused feature. $MSRL_{multi}$ incorporates authentic genomics data in the online branch and replaces the Inductor with the pre-trained $\phi_G$. We introduced with several foundation models to

serve as pre-trained baselines for comparison (Shao et al., 2025; Xu et al., 2024; Wang et al., 2024a; Ding et al., 2025). All foundation models utilized their respective patch encoders and initialized the slide encoders with public pre-trained weights. Then, we fine-tuned the slide encoder across all downstream tasks to ensure the objectivity and fairness of the reported results.

Table 1: The C-Index (mean ± std) on five survival prediction tasks, where "h." and "g." indicate rely on WSI and genomics, respectively. The cyan background represents methods trained with multi-modality data but inference with WSI. The best, second-best overall, and the best in cyan background results are highlighted in **bold red**, **underlined bold**, and **bold**, respectively.(†:p-value $<0.05$;‡:p-value $<0.01$)

| Model | Modality | BLCA (N=357) | BRCA (N=680) | STAD (N=318) | HNSC (N=392) | COADREAD (N=298) | Overall |
|---|---|---|---|---|---|---|---|
| SNN | g. | $0.5588 \pm 0.0314$‡ | $0.5816 \pm 0.0396$‡ | $0.5784 \pm 0.0409$‡ | $0.5456 \pm 0.0585$‡ | $0.5896 \pm 0.0512$‡ | 0.5708 |
| CLAM (Lu et al., 2021) | h. | $0.5304 \pm 0.0178$‡ | $0.5286 \pm 0.0746$‡ | $0.5482 \pm 0.0421$‡ | $0.5160 \pm 0.0331$‡ | $0.5740 \pm 0.0308$‡ | 0.5394 |
| SetMIL(Zhao et al., 2022) | h. | $0.5351 \pm 0.0742$‡ | $0.5692 \pm 0.0323$‡ | $0.5404 \pm 0.0511$‡ | $0.5280 \pm 0.0573$‡ | $0.5814 \pm 0.0717$‡ | 0.5508 |
| WiKG (Li et al., 2024) | h. | $0.5531 \pm 0.0204$‡ | $0.5827 \pm 0.0983$‡ | $0.5617 \pm 0.0983$‡ | $0.5303 \pm 0.0354$‡ | $0.5904 \pm 0.0517$‡ | 0.5636 |
| TransMIL (Shao et al., 2021) | h. | $0.5632 \pm 0.0273$‡ | $0.5372 \pm 0.0293$‡ | $0.5762 \pm 0.0464$‡ | $0.5570 \pm 0.0276$‡ | $0.6164 \pm 0.0977$‡ | 0.5686 |
| PANTHER (Song et al., 2024) | h. | $0.5712 \pm 0.0541$‡ | $0.6208 \pm 0.0997$‡ | $0.6219 \pm 0.0598$‡ | $0.5594 \pm 0.0550$‡ | $0.6101 \pm 0.0500$‡ | 0.5967 |
| FEATHER (Shao et al., 2025) | h. | $0.5306 \pm 0.0340$‡ | $0.5698 \pm 0.0247$‡ | $0.5570 \pm 0.0420$‡ | $0.5277 \pm 0.0302$‡ | $0.5796 \pm 0.0284$‡ | 0.5530 |
| CHIEF (Wang et al., 2024a) | h. | $0.5606 \pm 0.0890$‡ | $0.5762 \pm 0.0768$‡ | $0.5668 \pm 0.0601$‡ | $0.5338 \pm 0.0838$‡ | $0.5822 \pm 0.0672$‡ | 0.5639 |
| GigaPath (Xu et al., 2024) | h. | $0.5656 \pm 0.0291$‡ | $0.6282 \pm 0.0191$‡ | $0.6176 \pm 0.0346$‡ | $0.5580 \pm 0.0330$‡ | $0.6082 \pm 0.0180$‡ | 0.5954 |
| TITAN (Ding et al., 2025) | h. | $0.5756 \pm 0.0864$‡ | $0.6182 \pm 0.0711$‡ | $0.6306 \pm 0.0838$‡ | $0.5652 \pm 0.0381$‡ | $0.6138 \pm 0.0340$‡ | 0.6007 |
| MSRL$_\mathbf{H}$ | h. | $0.5774 \pm 0.0221$‡ | $0.6398 \pm 0.0251$‡ | $0.6626 \pm 0.0427$‡ | $0.5676 \pm 0.0289$‡ | $0.6182 \pm 0.0174$‡ | 0.6131 |
| MCAT (Chen et al., 2021) | g.+h. | $0.6038 \pm 0.0130$‡ | $0.6654 \pm 0.0182$‡ | $0.7064 \pm 0.0262$‡ | $0.6164 \pm 0.0536$‡ | $0.6358 \pm 0.0684$‡ | 0.6455 |
| MOTCat (Xu & Chen, 2023) | g.+h. | $0.6097 \pm 0.0540$‡ | $0.6689 \pm 0.0671$‡ | $0.7086 \pm 0.0522$‡ | $0.6175 \pm 0.0637$‡ | $0.6489 \pm 0.0346$‡ | 0.6507 |
| CMTA (Zhou & Chen, 2023) | g.+h. | $0.6110 \pm 0.0098$‡ | $0.6708 \pm 0.0323$‡ | $0.7110 \pm 0.0090$‡ | $0.6214 \pm 0.0470$‡ | $0.6580 \pm 0.0177$‡ | 0.6547 |
| PIBD (Zhang et al., 2024) | g.+h. | $0.6116 \pm 0.0318$† | $0.6738 \pm 0.0406$‡ | $0.7188 \pm 0.0267$‡ | $0.6244 \pm 0.0434$† | $0.6578 \pm 0.0654$‡ | 0.6573 |
| LD-CVAE $_{multi}$ (Zhou et al., 2025) | g.+h. | $0.6210 \pm 0.0131$‡ | $0.6712 \pm 0.0199$‡ | **0.7201** $\pm 0.0395$‡ | $0.6302 \pm 0.0303$‡ | $0.6602 \pm 0.0224$‡ | 0.6605 |
| SurvPath (Jaume et al., 2024b) | g.+h. | **0.6288** $\pm 0.0184$‡ | $0.6866 \pm 0.0209$‡ | $0.7194 \pm 0.0524$‡ | $0.6328 \pm 0.0256$‡ | $0.6712 \pm 0.0150$‡ | 0.6683 |
| DisPro$_{multi}$ (Xu et al., 2025) | g.+h. | $0.6267 \pm 0.0423$‡ | **0.6931** $\pm 0.0372$† | $0.7097 \pm 0.0403$‡ | **0.6390** $\pm 0.0580$† | **0.6770** $\pm 0.0479$† | **0.6691** |
| MSRL$_{multi}$ | g.+h. | **0.6368** $\pm 0.0327$† | **0.7012** $\pm 0.0302$‡ | **0.7236** $\pm 0.0411$† | **0.6456** $\pm 0.0263$‡ | **0.6896** $\pm 0.0301$† | **0.6794** |
| G-HANet (Wang et al., 2025) | g.+h.→h. | $0.5806 \pm 0.0149$‡ | $0.6418 \pm 0.0138$‡ | $0.6782 \pm 0.0489$‡ | $0.5770 \pm 0.0278$‡ | $0.6216 \pm 0.0184$‡ | 0.6246 |
| LD-CVAE (Zhou et al., 2025) | g.+h.→h. | $0.5954 \pm 0.0104$‡ | $0.6430 \pm 0.0146$‡ | $0.6938 \pm 0.0495$‡ | $0.5960 \pm 0.0286$‡ | $0.6280 \pm 0.0211$‡ | 0.6313 |
| DisPro (Xu et al., 2025) | g.+h.→h. | $0.6058 \pm 0.0269$‡ | $0.6734 \pm 0.0352$† | $0.6803 \pm 0.0424$‡ | $0.6053 \pm 0.0610$‡ | $0.6418 \pm 0.0342$‡ | 0.6414 |
| MSRL | g.+h.→h. | **0.6192** $\pm 0.0184$ | **0.6808** $\pm 0.0277$ | **0.7050** $\pm 0.0523$ | **0.6182** $\pm 0.0015$ | **0.6554** $\pm 0.0166$ | **0.6558** |

## 4.2 COMPARISONS WITH STATE-OF-THE-ARTS

### 4.2.1 SURVIVAL PREDICTION

Survival prediction aims to estimate the time-to-event outcomes for patients (Zadeh & Schmid, 2020; Jaume et al., 2024b). We use the concordance index (C-Index) as the evaluation metric. The formulation of survival prediction and the calculation of the C-Index are provided in the Appendix D.1. Table 1 presents the experimental results of the proposed MSRL compared with several state-of-the-art methods.

MSRL with WSI-only inference substantially outperforms existing uni-modal WSI methods. In particular, it achieves a 5.91% improvement in C-index over PANTHER, the state-of-the-art uni-modal WSI method without pre-training. Foundation models exhibit effective WSI representation capabilities pre-trained by large-scale data. For example, TITAN leverages region-level text-image multi-modal pre-training and significantly surpasses the performance of the remaining uni-modal training approaches. Nevertheless, MSRL still outperformed TITAN by 5.51%. These results demonstrate that MSRL can further enhance the diagnostic utility of WSI in real-world scenarios.

MSRL addresses the challenge of missing genomics data during inference more effectively than existing methods. Reconstruction-based methods achieve significantly better inference performance than all unimodal approaches, which effectively address the challenge of missing genomics data for inference. G-HANet performed differential analysis on the original genomic sequences and then reconstructed the remaining sequences during training. However, the heterogeneity among test cases results in significant biases in the reconstructed genomic data. The Variational Autoencoder of LD-CVAE struggled to fit the low-rank data distribution where the sample size is much smaller than the feature dimension ($d = 768$), resulting in significant noise during sampling in inference. These shortcomings result in G-HANet and LD-CVAE falling behind MCAT by 2.09% and 1.42% in C-index. MSRL avoids the rough situation that LD-VAE is stuck in and employs structure-guided reconstruction based on authentic original genomic data, which contribute to improvements in by 3.12% and 2.45% compared to G-HANet and LD-CVAE, respectively. DisPro distills prognostic knowledge into the genomics prompt during training. However, its inference procedure relies on self-scores computed within each individual case to select and aggregate tokens. This design overlooks

holistic case-level representations and inter-case relationships, which results in a c-index that is 1.44% lower than MSRL's. In addition, DisPro performs multi-stage uni-modal training, which makes it sensitive to the data distributions of both modalities and weakens its generalization ability at inference time. As shown in Table A1, DisPro's c-index drops by 4.77% when evaluated on out-of-domain data. In contrast, MSRL only a 1.02% performance decrease under the same setting, which exhibits strong generalization capacity.

Compared with multi-modal fusion methods, MSRL outperforms MCAT and CMTA by 1.03% and 0.11%, respectively. MCAT enhances WSI representations with genomic data unidirectionally, ignoring WSI's impact on genomic features. CMTA aligns WSI and genomic representations, but encodes genomic data using a WSI-specific module (Shao et al., 2021) that disrupts the sequence structure. MSRL's inter-modality constraint and genomic-specific encoding address these issues, which contribute to performance comparable to advanced multi-modal fusion methods, and enable data-efficient prediction using only the single image modality.

$MSRL_{multi}$ achieves the optimal performance and outperforms the second-best method by 1.03% in C-Index. This demonstrates that the multi-modal structural representation graph introduced by MSRL effectively enhances the integration of WSI and genomic features for survival prediction.

Table 2: The performance on four precision diagnosis tasks, where a cyan background represents methods trained with multi-modality data but inference with WSI, and the others are WSI uni-modal methods. The best and second-best results are highlighted in **bold red** and **underlined bold**, respectively.($\dagger$:p-value $<0.05$;$\ddagger$:p-value $<0.01$)

| Model | BRCA staging (n=944) | | NSCLC staging (n=893) | | EGFR mutation (n=627) | | HER2 status (n=482) | |
|---|---|---|---|---|---|---|---|---|
| | AUC | F1 score | AUC | F1 score | AUC | F1 score | AUC | F1 score |
| CLAM (Lu et al., 2021) | $0.577 \pm 0.0308^{\ddagger}$ | $0.535 \pm 0.0219^{\ddagger}$ | $0.590 \pm 0.0138^{\ddagger}$ | $0.557 \pm 0.0187^{\ddagger}$ | $0.765 \pm 0.0263^{\ddagger}$ | $0.702 \pm 0.0187^{\ddagger}$ | $0.628 \pm 0.0172^{\ddagger}$ | $0.500 \pm 0.0273^{\ddagger}$ |
| SetMIL (Zhao et al., 2022) | $0.580 \pm 0.0274^{\ddagger}$ | $0.542 \pm 0.0377^{\ddagger}$ | $0.597 \pm 0.0204^{\ddagger}$ | $0.563 \pm 0.0041^{\ddagger}$ | $0.779 \pm 0.0199^{\ddagger}$ | $0.705 \pm 0.0242^{\ddagger}$ | $0.667 \pm 0.0148^{\ddagger}$ | $0.507 \pm 0.0181^{\ddagger}$ |
| TransMIL (Shao et al., 2021) | $0.609 \pm 0.0203^{\ddagger}$ | $0.547 \pm 0.0140^{\ddagger}$ | $0.619 \pm 0.0182^{\ddagger}$ | $0.572 \pm 0.0258^{\ddagger}$ | $0.800 \pm 0.0286^{\ddagger}$ | $0.712 \pm 0.0252^{\ddagger}$ | $0.674 \pm 0.0295^{\ddagger}$ | $0.512 \pm 0.0382^{\ddagger}$ |
| DSMIL (Li et al., 2021) | $0.600 \pm 0.0097^{\ddagger}$ | $0.563 \pm 0.0374^{\ddagger}$ | $0.627 \pm 0.0123^{\ddagger}$ | $0.575 \pm 0.0274^{\ddagger}$ | $0.813 \pm 0.0122^{\ddagger}$ | $0.726 \pm 0.0353^{\ddagger}$ | $0.681 \pm 0.0470^{\ddagger}$ | $0.514 \pm 0.0245^{\ddagger}$ |
| WiKG (Li et al., 2024) | $0.619 \pm 0.0182^{\ddagger}$ | $0.567 \pm 0.0249^{\ddagger}$ | $0.640 \pm 0.0204^{\ddagger}$ | $0.587 \pm 0.0088^{\ddagger}$ | $0.814 \pm 0.0081^{\ddagger}$ | $0.733 \pm 0.0197^{\ddagger}$ | $0.690 \pm 0.0150^{\ddagger}$ | $0.515 \pm 0.0426^{\ddagger}$ |
| PANTHER (Song et al., 2024) | $0.643 \pm 0.0124^{\ddagger}$ | $0.574 \pm 0.0272^{\ddagger}$ | $0.648 \pm 0.0340^{\ddagger}$ | $0.611 \pm 0.0282^{\ddagger}$ | $0.820 \pm 0.0154^{\ddagger}$ | $0.749 \pm 0.0191^{\ddagger}$ | $0.698 \pm 0.0386^{\ddagger}$ | $0.541 \pm 0.0312^{\ddagger}$ |
| CHIEF (Wang et al., 2024a) | $0.602 \pm 0.0169^{\ddagger}$ | $0.566 \pm 0.0300^{\ddagger}$ | $0.635 \pm 0.0270^{\ddagger}$ | $0.589 \pm 0.0448^{\ddagger}$ | $0.804 \pm 0.0304^{\ddagger}$ | $0.713 \pm 0.0340^{\ddagger}$ | $0.657 \pm 0.0110^{\ddagger}$ | $0.517 \pm 0.0637^{\ddagger}$ |
| GigaPath (Xu et al., 2024) | $0.625 \pm 0.0074^{\ddagger}$ | $0.570 \pm 0.0214^{\ddagger}$ | $0.645 \pm 0.0180^{\ddagger}$ | $0.590 \pm 0.0188^{\ddagger}$ | $0.817 \pm 0.0313^{\ddagger}$ | $0.743 \pm 0.0088^{\ddagger}$ | $0.691 \pm 0.0186^{\ddagger}$ | $0.537 \pm 0.0155^{\ddagger}$ |
| FEATHER (Shao et al., 2025) | $0.627 \pm 0.0085^{\ddagger}$ | $0.571 \pm 0.0056^{\ddagger}$ | $0.646 \pm 0.0159^{\ddagger}$ | $0.593 \pm 0.0150^{\ddagger}$ | $0.816 \pm 0.0061^{\ddagger}$ | $0.748 \pm 0.0240^{\ddagger}$ | $0.689 \pm 0.0259^{\ddagger}$ | $0.538 \pm 0.0620^{\ddagger}$ |
| TITAN (Ding et al., 2025) | $0.648 \pm 0.0044^{\ddagger}$ | $0.583 \pm 0.0438^{\ddagger}$ | $0.639 \pm 0.0326^{\ddagger}$ | $0.614 \pm 0.0166^{\ddagger}$ | $0.822 \pm 0.0287^{\ddagger}$ | $0.751 \pm 0.0150^{\ddagger}$ | $0.693 \pm 0.0067^{\ddagger}$ | $0.546 \pm 0.0157^{\ddagger}$ |
| $MSRL_{\mathbf{H}}$ | $\underline{\mathbf{0.652}} \pm 0.0095^{\dagger}$ | $\underline{\mathbf{0.586}} \pm 0.0271^{\dagger}$ | $\underline{\mathbf{0.655}} \pm 0.0133^{\dagger}$ | $\underline{\mathbf{0.625}} \pm 0.0186^{\dagger}$ | $0.826 \pm 0.0200^{\ddagger}$ | $0.758 \pm 0.0173^{\ddagger}$ | $0.704 \pm 0.0569^{\ddagger}$ | $0.550 \pm 0.0372^{\ddagger}$ |
| G-HANet (Wang et al., 2025) | $0.632 \pm 0.0263^{\ddagger}$ | $0.572 \pm 0.0108^{\ddagger}$ | $0.634 \pm 0.0416^{\ddagger}$ | $0.614 \pm 0.0410^{\ddagger}$ | $0.830 \pm 0.0181^{\ddagger}$ | $0.762 \pm 0.0150^{\ddagger}$ | $0.715 \pm 0.0452^{\ddagger}$ | $0.576 \pm 0.0423^{\ddagger}$ |
| LD-CVAE (Zhou et al., 2025) | $0.646 \pm 0.0309^{\ddagger}$ | $0.582 \pm 0.0231^{\ddagger}$ | $0.650 \pm 0.0264^{\ddagger}$ | $0.619 \pm 0.0279^{\ddagger}$ | $\underline{\mathbf{0.836}} \pm 0.0215^{\ddagger}$ | $\underline{\mathbf{0.765}} \pm 0.0177^{\ddagger}$ | $\underline{\mathbf{0.717}} \pm 0.0254^{\ddagger}$ | $\underline{\mathbf{0.587}} \pm 0.0295^{\ddagger}$ |
| MSRL | $\mathbf{0.664} \pm 0.0263$ | $\mathbf{0.593} \pm 0.0277$ | $\mathbf{0.661} \pm 0.0102$ | $\mathbf{0.638} \pm 0.0108$ | $\mathbf{0.842} \pm 0.0206$ | $\mathbf{0.770} \pm 0.0165$ | $\mathbf{0.730} \pm 0.0223$ | $\mathbf{0.606} \pm 0.0274$ |

### 4.2.2 PRECISION DIAGNOSIS

We conduct four precision diagnosis tasks on two cancer staging datasets and two molecular prediction dataset. The experimental results shown in Table 2 demonstrate that the proposed MSRL and its variant $MSRL_{\mathbf{H}}$ achieve the highest performance across all tasks. Specifically, MSRL outperforms the second-best LD-CVAE in the AUCs/F1 scores for the four tasks by 1.8%/1.1%, 1.1%/1.9%, 0.6%/0.5%, and 1.3%/1.9%, respectively.

**Degradation of WSI encoder capabilities during gene reconstruction.** The diagnostic criteria of cancer staging relies on morphological characteristics, which makes it mainly dependent on WSIs. In the two cancer staging tasks, G-HANet is 1.1% lower than the WSI method, PANTHER, in the AUC. Moreover, LD-CVAE introducing genomics data during training, fails to achieve a significant performance improvement. This indicates that existing gene reconstruction methods compromise the WSI encoder's representation ability due to noisy high-dimensional genomics data being introduced during training the WSI encoder from scratch, resulting in bias in morphological feature learning.

**MSRL effectively enhances WSI encoder capabilities.** GigaPath is the baseline WSI encoder in our method. The WSI unimodal $MSRL_{\mathbf{H}}$ improves F1 scores by 1.6%, 3.5%, 1.5%, and 1.3% across four tasks compared to the baseline and outperforms existing missing modality methods in cancer staging with MSRL pre-training. Introducing genomics data further improves MSRL performance, which shows that MSRL pre-training effectively leverages the structural information of cases to enhance the WSI encoder representation. Authentic genomics data guidance constructs effective case-level relevance and further strengthens the WSI encoder.

## 5 MODEL ANALYSIS

### 5.1 ABLATION ANALYSIS

Table 3 presents MSRL structure ablation results on survival prediction. In particular, KNN (cosine) and KNN (Euclidean) correspond to adjacency matrices that are statically constructed using cosine similarity and Euclidean distance, respectively, instead of being learned by the graph learner. $\text{MSRL}_{\text{online\_buffer}}$ updates the buffer with online branch features rather than target branch features, and $\text{MSRL}_{\text{random\_GSL}}$ denotes a variant where the GSL is randomly initialized without pre-training.

MSRL outperforms $\text{MSRL}_{\text{random\_GSL}}$ across all datasets, which demonstrates the effectiveness of the pre-training. Moreover, $\text{MSRL}_{\text{random\_GSL}}$ performs better than both KNN-based methods, which suggests that GSL captures implicit and comprehensive diagnostic relevance, not only similarity as represented by KNN. The performance drop in $\text{MSRL}_{\text{online\_buffer}}$ demonstrates the effectiveness of introducing authentic data from the target branch to construct relevance during inference.

Table 3: The results of the structure ablation on five survival prediction datasets.($\dagger$:p-value $<0.05$;$\ddagger$:p-value $<0.01$)

| Model | BLCA (N=357) | BRCA (N=680) | STAD (N=318) | HNSC (N=392) | COADREAD (N=298) | Overall |
|---|---|---|---|---|---|---|
| KNN (Euclidean) | $0.5692 \pm 0.0293^{\ddagger}$ | $0.6283 \pm 0.0211^{\ddagger}$ | $0.6326 \pm 0.0361^{\ddagger}$ | $0.5626 \pm 0.0306^{\ddagger}$ | $0.6117 \pm 0.0118^{\ddagger}$ | $0.6009(\downarrow 0.0549)$ |
| KNN (cosine) | $0.5774 \pm 0.0234^{\ddagger}$ | $0.6371 \pm 0.0257^{\ddagger}$ | $0.6414 \pm 0.0411^{\ddagger}$ | $0.5681 \pm 0.0313^{\ddagger}$ | $0.6188 \pm 0.0141^{\ddagger}$ | $0.6086(\downarrow 0.0472)$ |
| $\text{MSRL}_{\text{random\_GSL}}$ | $0.5984 \pm 0.0216^{\dagger}$ | $0.6694 \pm 0.0331^{\ddagger}$ | $0.6812 \pm 0.0320^{\ddagger}$ | $0.5988 \pm 0.0253^{\dagger}$ | $0.6368 \pm 0.0156^{\ddagger}$ | $0.6369(\downarrow 0.0189)$ |
| $\text{MSRL}_{\text{online\_buffer}}$ | $0.6094 \pm 0.0222^{\dagger}$ | $0.6716 \pm 0.0307^{\dagger}$ | $0.6948 \pm 0.0492^{\dagger}$ | $0.6036 \pm 0.0275^{\dagger}$ | $0.6462 \pm 0.0149^{\dagger}$ | $0.6451(\downarrow 0.0107)$ |
| MSRL | $\mathbf{0.6192} \pm 0.0184$ | $\mathbf{0.6808} \pm 0.0277$ | $\mathbf{0.7050} \pm 0.0523$ | $\mathbf{0.6182} \pm 0.0015$ | $\mathbf{0.6554} \pm 0.0166$ | $\mathbf{0.6558}$ |

Table 4 shows that removing the alignment of the graph structure $\mathcal{L}_{s\_align}$ results in the most substantial performance drop, which confirms that the effectiveness and authenticity of the structure are fundamental to our framework. Notably, when the dataset size is relatively small, graph learning without structural constraints tends to suffer from instability and fails to achieve convergence. We performed the comprehensive experimental analysis on generalization of MSRL, the pre-training loss components, the buffer mechanism, and the parameter settings in Appendix E.

Table 4: Ablation results of the loss function during the fine-tuning on validation datasets.

| | | | | | | | |
|---|---|---|---|---|---|---|---|
| $\mathcal{L}_{f\_align}$ | ✓ | | ✓ | ✓ | ✓ | | |
| $\mathcal{L}_{g\_align}$ | ✓ | ✓ | | ✓ | | ✓ | |
| $\mathcal{L}_{s\_align}$ | ✓ | ✓ | ✓ | | | | ✓ |
| BLCA(N=357) | 0.607 | 0.588 ($\downarrow$ 0.019) | 0.599($\downarrow$ 0.008) | 0.584($\downarrow$ 0.023) | 0.570($\downarrow$ 0.037) | 0.564($\downarrow$ 0.043) | 0.573($\downarrow$ 0.034) |
| BRCA(N=680) | 0.672 | 0.644($\downarrow$ 0.028) | 0.661($\downarrow$ 0.011) | 0.623($\downarrow$ 0.049) | 0.583($\downarrow$ 0.089) | 0.577($\downarrow$ 0.095) | 0.592($\downarrow$ 0.080) |
| STAD(N=318) | 0.729 | 0.694($\downarrow$ 0.035) | 0.703($\downarrow$ 0.026) | 0.687($\downarrow$ 0.042) | 0.657($\downarrow$ 0.072) | Non-convergence | 0.663($\downarrow$ 0.066) |
| HNSC(N=392) | 0.622 | 0.609($\downarrow$ 0.013) | 0.617($\downarrow$ 0.005) | 0.598($\downarrow$ 0.024) | 0.563($\downarrow$ 0.059) | 0.558($\downarrow$ 0.064) | 0.587($\downarrow$ 0.035) |
| COADREAD(N=298) | 0.661 | 0.647($\downarrow$ 0.014) | 0.656($\downarrow$ 0.006) | 0.635($\downarrow$ 0.026) | 0.558($\downarrow$ 0.103) | Non-convergence | 0.577($\downarrow$ 0.084) |

### 5.2 VISUALIZATION ANALYSIS

We visualize the pre-trained multi-modal pan-cancer graph in Figure A1. The results demonstrate that the self-supervised GSL not only captures the relevance among similar WSIs but also uncovers potential RNA-related connections across heterogeneous WSIs, which contribute to constructing a more comprehensive graph representation and enable efficient inference based on unimodal data. Furthermore, Figure A2 confirms that leveraging authentic data facilitates a more effective construction of inter-case relevance. A more detailed analysis is provided in Appendix B.

## 6 CONCLUSION

The proposed MSRL framework jointly optimizes representation learning and structure learning. Extensive experiments on TCGA demonstrate that MSRL significantly outperforms existing methods and effectively addresses the challenges of missing modality with case-level relevance construction. However, there are some limitations in the current work as described in Appendix F. In future work, we will further extend these analyses and refine the MSRL framework to broaden its applicability to more diverse scenarios.

## 7 REPRODUCIBILITY STATEMENT

We believe that the MSRL framework is not only effective for histopathology–genomics tasks but also holds research value for other hierarchical multi-modal problems, such as broader domains involving molecular structures and protein expression. To this end, we provide a detailed code demo in the supplementary materials. Appendix C describes in detail the setup of the publicly available TCGA dataset used in our study, and Appendix D specifies the task definitions, model parameters, and the software/hardware environment adopted in the experiments. We hope these materials will sufficiently ensure the reproducibility of our approach.

## 8 ACKNOWLEDGEMENTS

This work was partly supported by the National Natural Science Foundation of China (Grant No. 62571015 and 62171007), partly supported by the Beijing Natural Science Foundation (Grant No. 7242270), partly supported by the Anhui Provincial Natural Science Foundation (Grant No. 2408085MF162).

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

APPENDIX

## A  THE STATEMENT OF LLMS USAGE

Here we explicitly declare that large language models (LLMs, e.g., the ChatGPT series) did not participate in any of the preliminary research work, including but not limited to literature review, idea formulation, method design, code implementation, experimental design, data processing, result organization and analysis, or figure generation. Their involvement was limited solely to the final stage of manuscript preparation, specifically for text polishing tasks such as grammar and spelling checks, refinement of certain expressions, and minor LaTeX table formatting adjustments. Importantly, they did not contribute to early-stage tasks such as drafting the article outline or designing the paragraph structure.

## B  VISUALIZATION

### B.1  THE PRE-TRAINED GRAPH OF PAN-CANCER DATASET

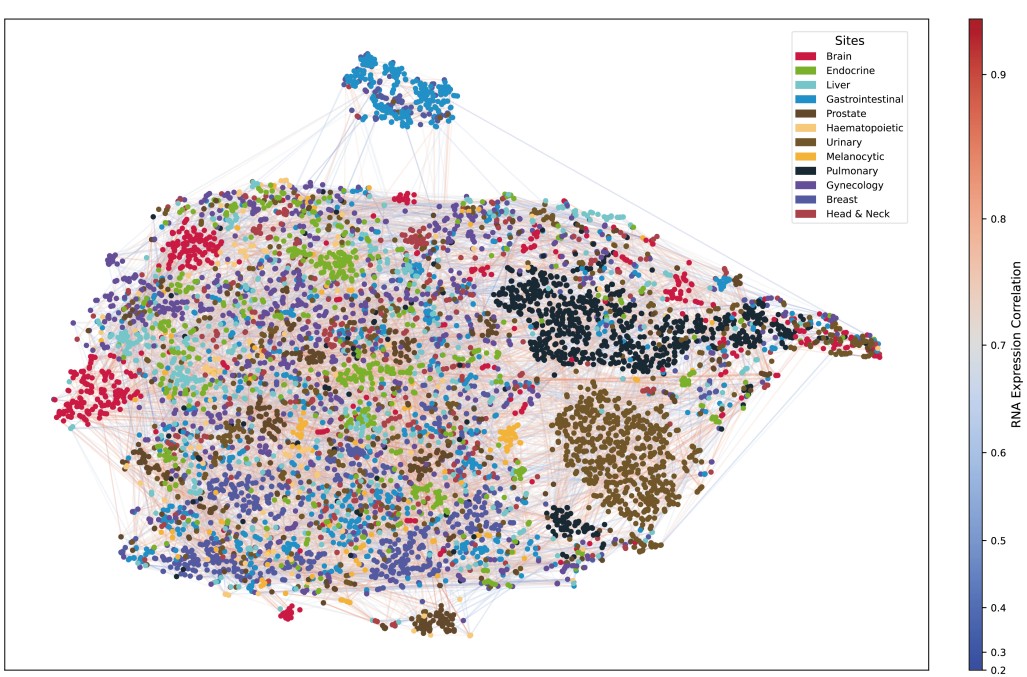

Figure A1: The graph of the pan-cancer dataset constructed by the pre-trained fused-modal GSL, where each node denotes the WSI representation of a case and the coordinates are clustered by t-SNE Van der Maaten & Hinton (2008), and each edge is weighted by the Pearson correlation coefficient between RNA expression of cases denoted by the two nodes.

We cluster Van der Maaten & Hinton (2008) the WSI representations of TCGA pan-cancer dataset, which are encoded by LongNet Ding et al. (2023) pre-trained by GigaPath Xu et al. (2024). Then, the pre-trained fused-modal Graph Structure Learner (GSL) is utilized to construct the adjacency matrix of the dataset. The completed graph of pre-trained pan-cancer dataset is shown as Figure A1, where each edge is weighted by the pearson correlation coefficient between RNA expression of cases denoted by the two nodes. The following observations are summarized:

**The foundation model demonstrates effective generalization ability.** Despite the absence of the TCGA dataset in the pre-training dataset of GigaPath, it exhibits the capacity to discern the morphological characteristics of TCGA WSIs. The figure illustrates that the WSI representations of the same site are aggregated into clusters.

**The pre-trained GSL of proposed MSRL can efficiently capture the morphological relationships of the cases.** The fused-modal GSL constructs denser edges for nodes within the same cluster, sparser edges for clusters between different sites, and also constructs denser associations for different clusters of the same site that are far from each other. It indicates that the fused-GSL can effectively capture the structural associations between cases in histo-morphology.

**The pre-trained GSL of proposed MSRL can efficiently capture the structural associations of cases at the molecular level.** The correlation between gene expression levels within a given cluster is high, while the correlation between different clusters is low. Furthermore, the different gene expression levels of distant clusters from the same site reflect the genetic heterogeneity between cancer subtypes.

**The molecular associations enhance the connectivity of the foundation model embeddings.** The distant cases in the feature space are also linked, which are corrected by the influence of gene correlation. These connections enhanced by genomics data play a crucial role in facilitating MSRL to achieve the precision multi-modal fusion.

## B.2 VISUALIZATION OF GRAPH STRUCTURE LEARNER

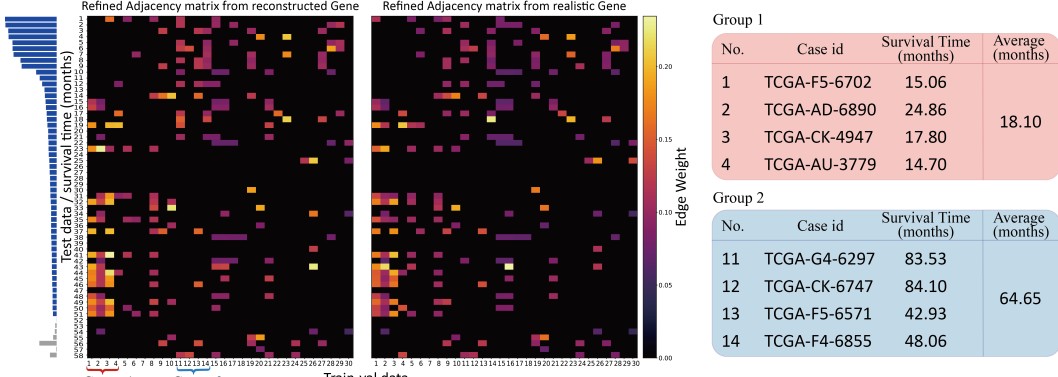

Figure A2: The adjacency matrix heatmaps obtained by GSL based on reconstructed and authentic genomic data, where each row shows edge weights between test cases and training cases. The survival times of two training groups are presented.

To further evaluate the effectiveness and authenticity of the structure learned by MSRL, we visualize the adjacency matrices generated by GSL from both authentic and reconstructed genomics data in COADREAD dataset, as shown in Figure A2. We input the WSI and genomics data of the test cases (n=58) into the target branch, and the output adjacency matrix of $\hat{GSL}$ is arranged in descending order based on the column sums, retaining the top 30 columns to obtain the "Refined Adjacency Matrix from Authentic Gene". Meanwhile, we input the WSI of the test cases into the online branch, reconstruct the genomics data, and obtain the output of GSL. The consistent 30 columns with the target branch are selected to generate the "Refined Adjacency Matrix from Reconstructed Gene". In Figure A2, each row of the heatmap represents the edge weights between test cases and training cases. The blue bars in the left histogram represent the survival time (in months) of the corresponding test cases, and the gray bars indicate that the case is deceased. Two observations are as follows: **(1) The online branch** GSL **can effectively learn the structural relationships from authentic data.** The heatmaps of both matrices show high consistency, especially in the high-weight edges (yellow areas). This indicates that the online branch's WSI-based Inductor can effectively reconstruct the authentic genomics data. And then GSL can build meaningful relationships between previously unseen test cases and real diagnostic training cases. This ensures that MSRL can perform efficient multi-modal task inference in real-world scenarios with WSI unimodality. **(2) Fine-tuned** GSL **can capture task-specific knowledge.** GSL significantly separates the test cases into high-risk group (rows 41 to 51) and low-risk group (rows 1 to 10), which show high association with group 1 (columns 1 to 4) and group 2 (columns 11 to 13) of the training cases. The average survival time of cases in Group 1 is

only 18.10 months and it is 64.65 months in Group 2. This demonstrates that the fine-tuned GSL can learn task-related diagnostic information among GSL and effectively promote precision oncology.

## C    DATASETS

**Dataset:**    We evaluate the proposed MSRL framework on six TCGA cohorts during fine-tuning. Specifically, we collect Bladder Urothelial Carcinoma (BLCA, n=359), Breast Invasive Carcinoma (BRCA, n=680), Stomach Adenocarcinoma (STAD, n=318), Head and Neck Squamous Cell Carcinoma (HNSC, n=392), and Colon and Rectum Adenocarcinoma (COADREAD, n=298) for survival prediction.  For precision diagnosis, we consider Non-Small Cell Lung Cancer (NSCLC) staging (n=893), Epidermal Growth Factor Receptor (EGFR) mutation status (n=627), BRCA staging (n=944), and BRCA human epidermal growth factor receptor-2 (HER2) status prediction (n=482, subset of BRCA staging dataset). We used 8 datasets for fine-tuning, and every dataset was split into a training-val dataset (containing 3,607 cases) and a testing dataset (containing 902 cases) with a 4:1 ratio, following a five-fold cross-validation strategy.. To ensure a sufficiently large pre-training dataset, we compiled an additional 2,754 cases from the TCGA pan-cancer database, which is along with the training-val dataset to form the pre-training dataset (containing 6,361 cases). In total, this paper used 7,263 cases (6,361 + 902), of which the testing dataset had no overlap with either the pre-training dataset or the training-val dataset. The model achieving the best performance on the validation set is selected for reporting results on the test set.

Additionally, we collected two public datasets from the CPTAC project for external evaluation. Specifically, CPTAC-HNSC dataset comprises WSI data and clinical survival information for 106 patients and CPTAC-BRCA dataset contains WSI data and diagnostic stage information for 111 patients. We ensured these datasets share no overlap with the TCGA project.

**Histopathology image collection:**    All WSIs data come from $20\times$ magnification hematoxylin and eosin (H&E)-stained slides. We crop each slide into $256\times256$-pixel patches and extract 1536-dimensional features using the ViT-giant model pre-trained on GigaPath (Xu et al., 2024). The WSI representation of MSRL is the 768-dimensional [*CLS*] token of output of LongNet (Ding et al., 2023) pretrained on GigaPath.  Among the compared foundation models, both CHIEF and FEATHER use ABMIL (Ilse et al., 2018) as their slide encoder, and TITAN adopts a ViT slide encoder. Notably, all the non foundation model methods utilize the same patch encoder as GigaPath.

**Genomics data collection:**    We collect raw genomics data for each corresponding case from TCGA. Following the Hallmarks resource in the Human Molecular Signatures Database (MSigDB) (Liberzon et al., 2015; Subramanian et al., 2005), we select 4241 genes and divide them into 50 groups and then apply log-normalization. Each group is encoded into a 768-dimensional feature with an independent SNN network (Klambauer et al., 2017).

## D    EXPERIMENTS

### D.1    SURVIVAL PREDICTION:

Survival prediction is to estimate cases' time-to-event outcomes.  Following previous research (Zadeh & Schmid, 2020), this task is defined by two components: censorship status and event time. Censorship status denoted as $c$, where $c = 0$ indicates the case's death was observed, and $c = 1$ indicates the cases' last known follow-up. Event time denoted as $t$, representing the time between diagnosis and observed death if $c = 0$, or between diagnosis and last follow-up, *i.e.* survival time, if $c = 1$. We estimate event time by dividing the time into non-overlapping intervals $(t_{j-1}, t_j)$ for $j \in [1, ..., n]$, based on quartiles of survival times ($c = 1$), and denote these intervals as $y_j$, rather than predicting the exact event time $t$ directly. This converts the problem into a classification task with censorship. Then, each case is represented by $(Z, y_j, c)$, where $Z$ is the representation of the case. We design a classifier where each output logit $\hat{y}_j$ predicted by the network corresponds to a specific time interval. Based on this, we define the discrete hazard function as

$$f_{\text{hazard}}(y_j|Z) = \sigma(\hat{y}_j),$$

where $\sigma$ denotes the sigmoid activation function. $f_{\text{hazard}}(y_j|Z)$ gives the probability that the patient dies within the time interval $(t_{j-1}, t_j)$. We then introduce the discrete survival function as

$$f_{\text{surv}}(y_j|Z) = \prod_{k=1}^{j} \left(1 - f_{\text{hazard}}(y_k|Z)\right),$$

which represents the probability that the patient survives up to the time interval $(t_{j-1}, t_j)$. Afterwards, we construct the negative log-likelihood (NLL) survival loss (Zadeh & Schmid, 2020) to optimize this task, as formulated as follows

$$\mathcal{L}_{NLL}\left(Z^{(i)}, y_j^{(i)}, c^{(i)}\}_{i=1}^{D}\right) = \sum_{i=1}^{D} -c^{(i)} \log\left(f_{\text{surv}}\left(y_j^{(i)} \mid Z^{(i)}\right)\right) \tag{8}$$

$$+ (1 - c^{(i)}) \log\left(f_{\text{surv}}\left(y_j^{(i)} - 1 \mid Z^{(i)}\right)\right) \tag{9}$$

$$+ (1 - c^{(i)}) \log\left(f_{\text{hazard}}\left(y_j^{(i)} \mid Z^{(i)}\right)\right) \tag{10}$$

where, $N_{\mathcal{D}}$ represents the total number of cases in the dataset. The loss ensures that the model assigns high survival probabilities to patients alive at last follow-up, correctly models survival up to death time for deceased patients, and accurately predicts the time of death when observed. A detailed mathematical explanation is provided in (Zadeh & Schmid, 2020). We finally take the negative sum of all logits to predict a patient-level risk score, which is used to categorize patients into different risk groups and to stratify them accordingly.

**C-Index:** The Concordance Index (C-Index) is a metric used to evaluate the consistency between predicted ordered sequences and true sequences. In survival prediction, C-Index measures how accurately the model ranks cases according to their survival times. The C-Index ranges from 0.5 to 1, where 0.5 indicates random prediction and 1 indicates perfect prediction. Specifically, the C-Index calculates the proportion of all comparable case pairs for which the predicted order matches the ground-truth order of survival times. For a pair of cases $(case_i, case_j)$, if $case_j$'s survival time is longer than $case_j$'s and the model predicts a lower risk for $case_i$ than for patient $case_j$, this pair is called a "concordant pair." The formulation of the C-Index is as follows

$$\text{c-index} = \frac{1}{D(D-1)} \sum_{i=1}^{D} \sum_{j=1}^{D} I(t_i < t_j)(1 - c_j),$$

where $I(\cdot)$ is the indicator function, which takes the value 1 if the argument is true, and 0 otherwise.

### D.2 EXPERIMENTAL IMPLEMENTATIONS:

In the pre-training, a graph with 6,361 nodes was constructed, where each node contains a 768-dimensional WSI feature and a 768-dimensional genomic feature. The Graph Structure Learner employed a 2-layer architecture, and the Graph Convolutional Network (GCN) consisted of 3 layers to output final node embeddings of 256 dimensions. The model was trained using the Adam optimizer with an initial learning rate of 1e-4 for 400 epochs. During fine-tuning, the AdamW optimizer was applied for 50 epochs with a 10-epoch warmup. The learning rate followed a cosine annealing schedule with a maximum learning rate of 1e-5 and a minimum of 1e-7. To validate the generalization of MSRL and baseline methods on out-of-distribution cases, we directly applied models trained on TCGA-HNSC and TCGA-BRCA to the CPTAC-HNSC and CPTAC-BRCA datasets for testing, respectively. All implementations were carried out in Python 3.9, PyTorch 2.0 and CUDA 12.4 on a computer cluster with six Nvidia GeForce 4090 GPUs.

**Post-processing of GSL:** The post-processing aims to refine the sketched adjacency matrix $\mathbf{S}$ into a sparse, non-negative, symmetric, and normalized adjacency matrix $\mathbf{A}^r$. The three steps are as follows:

1. **Sparsification.** The obtained similarity matrix $S$ is typically dense and requires sparsification. For each node, we employ a KNN-based method and retain the top $K$ connected edges and set the remaining connections to zero, specifically as follows: $\mathbf{S}_{ij}^{(sp)} = q_{sp}(\mathbf{S}_{ij}) = \mathbf{S}_{ij} \ if \ \mathbf{S}_{ij} \in \text{top-k}(\mathbf{S}_i) \ else \ 0 \ ;$

2. **Symmetrization and Activation**. To ensure bidirectional connections between nodes and guarantee non-negative edge values, we perform the following additional processing: $\mathbf{S}^{(sym)} = q_{sym}\left(q_{act}\left(\mathbf{S}^{(sp)}\right)\right) = \frac{\sigma_q(\mathbf{S}^{(sp)}) + \sigma_q(\mathbf{S}^{(sp)})^{\mathrm{T}}}{2}$, where $\sigma(\cdot)$ is the activation function;

3. **Normalization**. To ensure the edge weights fall within the range $[0, 1]$, the final processing step is as follows: $\mathbf{A}^r = q_{norm}\left(\mathbf{S}^{(sym)}\right) = \left(\mathbf{D}^{(sym)}\right)^{-\frac{1}{2}} \mathbf{S}^{(sym)} \left(\mathbf{D}^{(sym)}\right)^{-\frac{1}{2}}$, where $D^{(sym)}$ is the degree matrix of $S^{(sym)}$.

# E SUPPLEMENTARY RESULTS AND DISCUSSION

## E.1 GENERALIZATION ANALYSIS

**Our framework presents no additional difficulty in generalization.** Genomic data from out-of-distribution (OOD) cases often suffers from complex sources and various protocols. Direct inclusion of such data can cause significant batch effects. In contrast, our method utilizes solely WSI unimodal inference. Its generalization only rely on the representation capability of the slide encoder. Therefore, the generalization capability of our method when facing out-of-distribution data is consistent with other WSI encoder-based inference frameworks and does not introduce additional challenges.

**Our cross-modal retrieval strategy even boosts the generalization ability.** Existing uni-modal inference methods typically focus on reconstructing raw genomic data for individual case, where the reconstruction performance is also affected by OOD issues. In contrast, MSRL models genomic correlations between cases within an aligned representation space and then use the real source-domain genomic representation for multi-modal prediction. This strategy naturally avoids the multi-source heterogeneity inherent in raw genomic data during training. Therefore, it ensures better generalization than reconstruction-based methods during the inference phase.

We constructed new external datasets to fully validate the generalization of MSRL as shown in Table A1.MSRL consistently achieved the best performance on both external datasets. Notably, MSRL$_H$ secured the top results even under conditions limited to WSI unimodal training. In scenarios addressing missing inference modalities, existing methods like G-HANet and LD-CAVE suffered performance drops exceeding 5% and 7% on CPTAC-HNSC and CPTAC-BRCA, respectively. In contrast, MSRL showed minimal declines of only 1.02% and 1.62%, outperforming most foundation models. These results demonstrate the superior generalization of MSRL. The performance of MSRL significantly decreases after removing the pre-training stage. Its generalization capability is also greatly diminished, where the C-index drops over 6%. This practically validates the necessity of our pre-training approach. As DisPro's prompts are specifically designed for survival prediction, we therefore excluded it from the cancer staging tasks.

Table A1: The generalization analysis results, where CPTAC-HNSC and CPTAC-BRCA are external test sets and $\Delta$ denotes performance declines.

| Method | Modality | TCGA-HNSC C-index | CPTAC-HNSC C-index | $\Delta$ | TCGA-BRCA AUC | CPTAC-BRCA AUC | $\Delta$ |
|---|---|---|---|---|---|---|---|
| FEATHER | h. | $0.5277 \pm 0.0302$ | $0.5197 \pm 0.0214$ | $-0.80\%$ | $0.627 \pm 0.0085$ | $0.5782 \pm 0.0142$ | $-4.88\%$ |
| CHIEF | h. | $0.5338 \pm 0.0388$ | $0.5246 \pm 0.0301$ | $-0.92\%$ | $0.602 \pm 0.0169$ | $0.5608 \pm 0.0198$ | $-4.12\%$ |
| GigaPath | h. | $0.5580 \pm 0.0330$ | $0.5278 \pm 0.0545$ | $-3.02\%$ | $0.625 \pm 0.0074$ | $0.5824 \pm 0.0457$ | $-4.26\%$ |
| TITAN | h. | $0.5652 \pm 0.0381$ | $0.5341 \pm 0.0610$ | $-3.11\%$ | $0.648 \pm 0.0044$ | $0.6094 \pm 0.0541$ | $-3.86\%$ |
| MSRL$_H$ | h. | $0.5676 \pm 0.0289$ | $0.5461 \pm 0.0125$ | $-2.15\%$ | $0.652 \pm 0.0095$ | $0.6099 \pm 0.0401$ | $-4.21\%$ |
| G-HANet | g.+h.$\rightarrow$h. | $0.5770 \pm 0.0278$ | $0.5238 \pm 0.0786$ | $-5.32\%$ | $0.632 \pm 0.0263$ | $0.5612 \pm 0.0634$ | $-7.08\%$ |
| LD-CVAE | g.+h.$\rightarrow$h. | $0.5960 \pm 0.0286$ | $0.5418 \pm 0.0469$ | $-5.42\%$ | $0.646 \pm 0.0309$ | $0.5728 \pm 0.0309$ | $-7.32\%$ |
| DisPro | g.+h.$\rightarrow$h. | $0.6053 \pm 0.0610$ | $0.5576 \pm 0.0692$ | $-4.77\%$ | – | – | – |
| MSRL w/o Pre | g.+h.$\rightarrow$h. | $0.5988 \pm 0.0253$ | $0.5364 \pm 0.0408$ | $-6.24\%$ | $0.639 \pm 0.0335$ | $0.5856 \pm 0.0381$ | $-5.34\%$ |
| MSRL | g.+h.$\rightarrow$h. | $\mathbf{0.6182 \pm 0.0015}$ | $\mathbf{0.6080 \pm 0.0118}$ | $-1.02\%$ | $\mathbf{0.664 \pm 0.0263}$ | $\mathbf{0.6478 \pm 0.0416}$ | $-1.62\%$ |

## E.2 BUFFER ANALYSIS

**Resource consumption:** Graph-based inference does not lead to significant additional resource consumption. We evaluated computational cost on 136 test WSIs of the BRCA dataset. We measured the model FLOPs and average time required of each WSI for patch feature extraction, and WSI encoding and task inference as shown in Table R2. Compared to other methods, our approach does not introduce notable increases in inference time. To simulate large-scale datasets, we additionally

increased the buffer size in multiples. The WSI inference time increased by only 0.607 seconds—less than 2% of the patch feature extraction time, even with the 50X buffer size. Therefore, our method is not a bottleneck in terms of time or resource consumption for practical applications.

Table A2: The results of resource consumption.

| Model | Patch Feature Extracting | | WSI-level Encoding and Inference | | Total | |
|---|---|---|---|---|---|---|
| | Time | FLOPs | Time | FLOPs | Time | FLOPs |
| CMTA | 34.87s | $4.68 \times 10^5$ G | 0.997s | $9.78 \times 10^0$ G | 35.867s | $\approx 4.68 \times 10^5$ G |
| SurvPath | 34.87s | $4.68 \times 10^5$ G | 0.439s | $2.26 \times 10^0$ G | 35.309s | $\approx 4.68 \times 10^5$ G |
| G-HANet | 34.87s | $4.68 \times 10^5$ G | 0.415s | $6.03 \times 10^0$ G | 35.285s | $\approx 4.68 \times 10^5$ G |
| LD-CVAE | 34.87s | $4.68 \times 10^5$ G | 0.487s | $2.65 \times 10^2$ G | 35.357s | $\approx 4.68 \times 10^5$ G |
| MSRL(buffer=434) | 34.87s | $4.68 \times 10^5$ G | 0.534s | $1.78 \times 10^2$ G | 35.404s | $\approx 4.68 \times 10^5$ G |
| MSRL(buffer=4340) | 34.87s | $4.68 \times 10^5$ G | 0.542s | $2.45 \times 10^2$ G | 34.412s | $\approx 4.68 \times 10^5$ G |
| MSRL(buffer=8680) | 34.87s | $4.68 \times 10^5$ G | 0.701s | $3.97 \times 10^2$ G | 35.571s | $\approx 4.68 \times 10^5$ G |
| MSRL(buffer=21700) | 34.87s | $4.68 \times 10^5$ G | 1.141s | $1.33 \times 10^3$ G | 36.011s | $\approx 4.69 \times 10^5$ G |

**Buffer size analysis:** We additionally construct experiments assessing the impact of changing buffer size on task performance as shown in Table A3. Results indicate that a smaller buffer size leads to weaker model performance. The performance reduction is more significant when the number of training samples is limited. When the buffer size is zero (i.e., removing the buffer), the model performance drops significantly. Furthermore, our approach constructs structural correlations between the current case and historical cases. Consequently, removing the buffer leads to the loss of functionality for both GSL and GCN. This demonstrates that introducing authentic data is key to MSRL's effective inference. It also shows that a sufficient data scale better assists the model in learning missing genomic information.

Table A3: The ablation results of the buffer size on validation datasets.

| Buffer size | 100% | 75% | 50% | 25% | 0% |
|---|---|---|---|---|---|
| BLCA (N=357) | 0.607 | 0.595 (-0.012) | 0.589 (-0.018) | 0.573 (-0.034) | 0.533 (-0.074) |
| BRCA (N=680) | 0.672 | 0.669 (-0.003) | 0.664 (-0.008) | 0.653 (-0.019) | 0.591 (-0.081) |
| STAD (N=318) | 0.729 | 0.713 (-0.016) | 0.705 (-0.024) | 0.691 (-0.038) | 0.611 (-0.118) |
| HNSC (N=392) | 0.622 | 0.615 (-0.007) | 0.611 (-0.011) | 0.602 (-0.020) | 0.545 (-0.077) |
| COADREAD (N=298) | 0.661 | 0.655 (-0.006) | 0.640 (-0.021) | 0.621 (-0.039) | 0.557 (-0.104) |

### E.3 PARAMETER ANALYSIS:

**Pre-training data settings:** We have added experiments to evaluate the effectiveness of H-G pairing and different scales of pre-training data. We constructed GSL models using 20%, 50%, and 80% of the pre-training data, and we also constructed a GSL using a fully pre-trained dataset with random H-G pairings. Table A4 presents the validation metrics of the pre-training models for five survival prediction tasks under different settings. Based on these experiments, we draw the following conclusions:

1. Using more pretraining data effectively improves model performance. The model performance is optimal when using the full dataset, and the performance improvement from a 50% to an 80% data increase is significantly higher than the improvement from 20% to 50%. This indicates that more data benefits the pretraining of GSL, which is consistent with the scaling law.

2. The pairing of H-G data is essential for multimodal research. The model performance using the fully pretraining dataset with random pairing is lower than that of the model pretraining with 20% paired data, and in some cases, the results are close to random predictions (for example, the c-index for HNSC and COADREAD was only 0.507 and 0.502, respectively). Compared with the method of training with WSI data only, H-G random pairing will introduce noise and reduce model performance.

**Pre-training loss ablation studies:** We have added the ablation study for the three loss components used during pre-training, as shown in Table A5. The results indicate that removing the individual loss $\mathcal{L}_{fuse}$ leads to the most significant performance reduction. This is because $\mathcal{L}_{fuse}$ constrains the

Table A4: The results on the validation set of the various data settings.

| Data setting | BLCA | BRCA | STAD | HNSC | COADREAD |
|---|---|---|---|---|---|
| random pairing pre-training data | 0.522 | 0.534 | 0.511 | 0.507 | 0.502 |
| WSI only | 0.574 | 0.593 | 0.582 | 0.536 | 0.524 |
| 20% pre-training data/1272 cases | 0.615 | 0.688 | 0.609 | 0.551 | 0.533 |
| 50% pre-training data/3181 cases | 0.627 | 0.694 | 0.613 | 0.556 | 0.541 |
| 80% pre-training data/5089 cases | 0.647 | 0.720 | 0.644 | 0.577 | 0.548 |
| 100% pre-training data/6361 cases | 0.651 | 0.744 | 0.658 | 0.593 | 0.558 |

multi-modal representation of the cases, playing a vital role in integrating multi-modal information for the downstream tasks. $\mathcal{L}_{inter}$ is the key constraint for aligning the cross-modal representations of the same case. $\mathcal{L}_{intra}$ enhances the discriminative power of the individual modal data representations. Removing any of these losses negatively impacts the downstream task performance.

Table A5: Ablation results of the pre-training losses on validation datasets.

| | | | | | | | |
|---|---|---|---|---|---|---|---|
| $\mathcal{L}_{inter}$ | ✓ | | ✓ | ✓ | ✓ | | |
| $\mathcal{L}_{intra}$ | ✓ | ✓ | | ✓ | | ✓ | |
| $\mathcal{L}_{fuse}$ | ✓ | ✓ | ✓ | | | | ✓ |
| BLCA | 0.607 | 0.582 ($\downarrow$0.025) | 0.571 ($\downarrow$0.036) | 0.567 ($\downarrow$0.040) | 0.522 ($\downarrow$0.085) | 0.514 ($\downarrow$0.093) | 0.556 ($\downarrow$0.051) |
| BRCA | 0.672 | 0.640 ($\downarrow$0.032) | 0.646 ($\downarrow$0.026) | 0.636 ($\downarrow$0.036) | 0.538 ($\downarrow$0.134) | 0.524 ($\downarrow$0.148) | 0.563 ($\downarrow$0.109) |
| STAD | 0.729 | 0.656 ($\downarrow$0.073) | 0.670 ($\downarrow$0.059) | 0.646 ($\downarrow$0.083) | 0.560 ($\downarrow$0.169) | 0.557 ($\downarrow$0.172) | 0.585 ($\downarrow$0.144) |
| HNSC | 0.622 | 0.603 ($\downarrow$0.019) | 0.594 ($\downarrow$0.028) | 0.569 ($\downarrow$0.053) | 0.530 ($\downarrow$0.092) | 0.536 ($\downarrow$0.086) | 0.559 ($\downarrow$0.063) |
| COADREAD | 0.661 | 0.621 ($\downarrow$0.040) | 0.629 ($\downarrow$0.032) | 0.613 ($\downarrow$0.048) | 0.545 ($\downarrow$0.116) | 0.543 ($\downarrow$0.118) | 0.573 ($\downarrow$0.088) |

**K value ablation studies:** Table A6 shows the validation metrics of the prognostic tasks for different values of K. The results indicate that the model is relatively robust to variations in K (with metric fluctuations within 0.02). Overall, as the value of K increases, the model can gather more relevant case support, which leads to higher performance. However, the computational complexity also increases accordingly. Considering the balance between performance and resource consumption, we ultimately selected K=12 for all experiments.

Table A6: The results on the validation set of the various K values in GSL.

| K | BLCA | BRCA | STAD | HNSC | COADREAD |
|---|---|---|---|---|---|
| 4 | 0.633 | 0.728 | 0.639 | 0.579 | 0.546 |
| 8 | 0.648 | 0.734 | 0.644 | 0.582 | 0.551 |
| 12 | 0.651 | 0.744 | 0.658 | 0.593 | 0.558 |
| 16 | 0.654 | 0.747 | 0.659 | 0.602 | 0.562 |

### E.4    MISSING MODALITY TRAINING

We additionally constructed experiments detailing different genomic data missing rates. We did not design the method to handle modal omission during the initial stage. Therefore, to conduct this experiment, we adjusted the training strategy for the Target branch. If a case lacks genomic data, the Target branch does not participate in training; only the WSI-input Online branch is trained. The specific experimental results are shown in Table A7. Even when using only partially complete data, MSRL still surpasses all uni-modal training methods. Furthermore, MSRL remains superior to existing methods for uni-modal inference, even at a 30% missing rate. This demonstrates that MSRL can fully utilize existing multi-modal data to learn comprehensive and robust representations. This greatly enhances WSI's diagnostic value and data efficiency during the inference phase.

### E.5    THE KM ANALYSIS OF SURVIVAL PREDICTION

To further assess the effectiveness of MSRL in survival prediction, we divide all patients into low-risk and high-risk groups based on the median of the predicted risk scores from MSRL. Then, we apply Kaplan-Meier analysis to visualize the survival outcomes of both groups, as shown in Figure A3. Additionally, we conduct a Log-rank test to evaluate the statistical significance between the low-risk group (blue) and the high-risk group (red). A p-value of 0.05 or less is considered statistically

Table A7: The results of different genomic data missing rates.

| Method | Modality | BLCA | BRCA | STAD | HNSC | COADREAD | Overall |
|--------|----------|------|------|------|------|----------|---------|
| G-HANet | g.+h.→h. | 0.5806±0.0149 | 0.6418±0.0138 | 0.6782±0.0489 | 0.5770±0.0278 | 0.6216±0.0184 | 0.6246 |
| LD-CVAE | g.+h.→h. | 0.5954±0.0104 | 0.6430±0.0146 | 0.6938±0.0495 | 0.5960±0.0286 | 0.6280±0.0211 | 0.6313 |
| DisPro | g.+h.→h. | 0.6058±0.0269 | 0.6734±0.0352 | 0.6803±0.0424 | 0.6053±0.0610 | 0.6418±0.0342 | 0.6414 |
| MSRL missing 60% | g.+h.→h. | 0.6008±0.0125 | 0.6508±0.0577 | 0.6918±0.0699 | 0.5896±0.0570 | 0.6186±0.0401 | 0.6303 |
| MSRL missing 30% | g.+h.→h. | 0.6132±0.0668 | 0.6752±0.0809 | 0.7034±0.0830 | 0.6026±0.0505 | 0.6448±0.0587 | 0.6478 |
| MSRL missing 0% | g.+h.→h. | 0.6192±0.0184 | 0.6808±0.0277 | 0.7050±0.0523 | 0.6182±0.0015 | 0.6554±0.0166 | 0.6558 |

significant. The results show that p-values of all datasets are much smaller than 0.05, indicating significant differences between the groups.

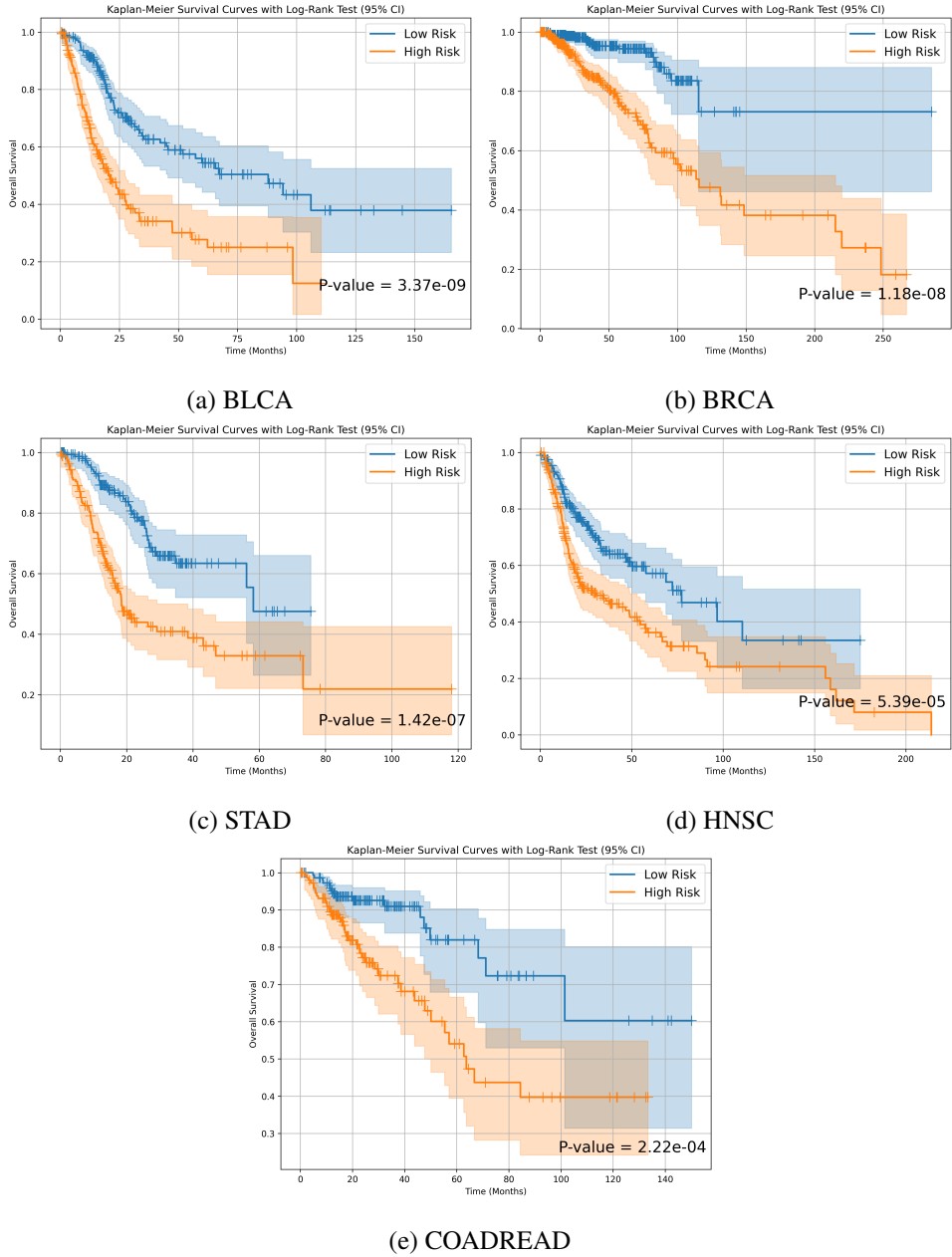

(a) BLCA

(b) BRCA

(c) STAD

(d) HNSC

(e) COADREAD

Figure A3: The curves of Kaplan-Meier analysis for all survival prediction datasets and the p-value of the Log-rank test.

## F    LIMITATIONS OF MSRL

There are three limitations of the current work: (1) Jointly optimizing node representations and graph structures easily leads to local optima, which makes it difficult to achieve task-optimal solutions. We proposed that pre-training and fine-tuning partially mitigate this challenge, but we still need to find a more stable optimization strategy during fine-tuning. (2) We have not yet introduced more structured textual information during MSRL's pre-training. Incorporating clinical reports could help MSRL learn more robust structural representations in multi-modal task inference. (3) We have not yet conducted a thorough analysis of the distribution of medical centers, ethnic composition, and regional characteristics in the TCGA dataset, nor have we examined their potential impact on the applicability of our method.

