# OpenReview forum: "Histopathology-Genomics Multi-modal Structural Representation Learning for Data-Efficient Precision Oncology"
_ICLR.cc/2026/Conference — ICLR 2026 Poster_

### Official Review · Reviewer_sJW6 · 2025-10-21

**Soundness:** 3
**Presentation:** 4
**Contribution:** 3
**Rating:** 6
**Confidence:** 4

**Summary:**

This paper proposes a two-stage framework, MSRL for Robust inference in precision oncology when genomics is missing at test time. Specifically, in stage one, self‑supervised multi‑modal Graph Structure Learning (GSL) is used to learn inter‑case structure over WSIs, genomics, and fused features. In stage two, a dual-branch with an online branch (WSI only, learns a genomics “prompt” via an SNN Inductor) and a target branch (uses authentic genomics during training, updated via EMA) was used for finetuning. In experiments, MSRL outperforms WSI‑only methods and prior “missing‑genomics reconstruction” approaches by 2.45–3.12% C‑Index on survival.

**Strengths:**

1.	Clarity of high-level idea and task framing: The authors make clear the distinction between conventional reconstruction and their structure guided reconstruction augmented with authentic training genomics.
2.	Methodological coherence: The pipeline that (i) pretrains a multi‑modal structural graph; (ii) transfers it to task-specific fine‑tuning; and (iii) retrieves structural support from training cases at inference is well motivated. The dual-branch (online/target with EMA) is a sensible way to stabilize learning and avoid collapse, and the hierarchical alignment losses are thoughtfully designed.
3.	Ablations and analyses: The KNN vs GSL ablations, loss ablations (especially showing $L_{salign}$ is critical), buffer source ablation, and buffer size/inference time analysis are useful and strengthen the paper.

**Weaknesses:**

1.	Evaluation fairness: The paper includes G‑HANet and LD‑CVAE (g.+h.→h) baselines that also aim to impute genomics. However, it is not clear whether they are allowed to retrieve/use training-case genomics at inference in the same way. If not, MSRL enjoys an extra source of information that makes comparisons less equitable.
2.	All results are intra TCGA: an external‑cohort generalization test is highly valuable. This is especially important because MSRL leverages structural similarity and retrieval. Generalization across centers is precisely where structural priors may shift.
3.	Report performance when only a fraction (e.g., 10–50\%) of training cases have genomics (partial-paired training) and when the buffer is limited.

**Questions:**

See weaknesses.

---

> ### Author Response · Authors · 2025-11-24
> **Response to Reviewer sJW6 (1/3)**
>
> Dear Reviewer,
>
> we sincerely thanks for your thorough review and appreciate your recognition of the soundness, presentation and contribution our work. Your comments are very helpful for improving our paper. We have added more experiments and provided clearer explanations to address your concerns. Our detailed responses are as follows:
>
> ---
>
>
> ## Response to Generalization
>
> **1. Mechanism of Generalization**
>
> - **Our framework presents no additional difficulty in generalization.**
> Genomic data from out-of-distribution cases often suffers from complex sources and various protocols. Direct inclusion of such data can cause significant batch effects. In contrast, our method utilizes solely WSI unimodal inference. Its generalization only rely on the representation capability of the slide encoder. Therefore, the generalization capability of our method when facing out-of-distribution data is consistent with other WSI encoder-based inference frameworks and does not introduce additional challenges.
>
> - **Our cross-modal retrieval strategy even boosts the generalization ability.**
> Existing unimodal inference methods typically focus on reconstructing raw genomic data for individual cases, where the recontruction performance is also affected by OOD issues. In contrast, MSRL models genomic correlations between cases within an aligned representation space and then use the real source-domain genomic representation for muilti-modal prediciton. This strategy naturally avoids the multi-source heterogeneity inherent in raw genomic data during training. Therefore, it ensures better generalization than reconstruction-based methods during the inference phase.
>
> **2. External Evaluation Experiments**
> We constructed new external datasets to fully validate the generalizability of MSRL. The details are as follows:
>
> - **External Datasets:** We collected two public datasets from the CPTAC project: **CPTAC-HNSC** (comprising WSI data and clinical survival information for 106 patients) and **CPTAC-BRCA** (containing WSI data and diagnostic stage information for 111 patients). We ensured these datasets share no overlap with the TCGA project.
>
> - **Experimental Setting:** To validate the generalizability of MSRL and baseline methods on out-of-distribution cases, we directly applied models trained on TCGA-HNSC and TCGA-BRCA to the CPTAC-HNSC and CPTAC-BRCA datasets for testing, respectively. dditionally, we incorporated several foundation models known for high generalizability for comparison, such as FEATHER [1], CHIEF [2], TITAN [3], and GigaPath [4]. The specific experimental results are shown in Table R2.
>
> - **Results Analysis:**
>    * MSRL consistently achieved the best performance on both external datasets. Notably, MSRL_H secured the top results even under conditions limited to WSI unimodal training.
>    * In scenarios addressing missing inference modalities, existing methods like G-HANet and LD-CAVE suffered performance drops exceeding 5% and 7% on CPTAC-HNSC and CPTAC-BRCA, respectively. In contrast, MSRL showed minimal declines of only 1.02% and 1.62%, outperforming most foundation models. These results demonstrate the superior generalizability of MSRL.
>
> ##### Table R2: The results on external test sets, where "Δ" denotes performance declines.
> |method|TCGA-HNSC C-index|CPTAC-HNSC C-index|Δ|TCGA-BRCA AUC|CPTAC-BRCA AUC|Δ|
> |---|---|---|---|---|---|---|
> |FEATHER|0.5277±0.0302|0.5197±0.0214|-0.80%|0.627±0.0085|0.5782±0.0142|-4.88%|
> |CHIEF|0.5338±0.0388|0.5246±0.0301|-0.92%|0.602±0.0169|0.5608±0.0198|-4.12%|
> |GigaPath|0.5580±0.0330|0.5278±0.0545|-3.02%|0.625±0.0074|0.5824±0.0457|-4.26%|
> |TITAN|0.5652±0.0381|0.5341±0.0610|-3.11%|0.648±0.0044|0.6094±0.0541|-3.86%|
> ||||||||
> |MSRL_H|0.5676±0.0289|0.5461±0.0125|-2.15%|0.652±0.0095|0.6099±0.0401|-4.21%|
> |G-HANet|0.5770±0.0278|0.5238±0.0786|-5.32%|0.632±0.0263|0.5612±0.0634|-7.08%|
> |LD-CVAE|0.5960±0.0286|0.5418±0.0469|-5.42%|0.646±0.0309|0.5728±0.0309|-7.32%|
> |MSRL|0.6182±0.0015|0.6080±0.0118|-1.02%|0.664±0.0263|0.6478±0.0416|-1.62%|
>
> *We will add these results to the appendix of original manuscript.*
>
> [1] Shao, Daniel, et al. "Do Multiple Instance Learning Models Transfer?." ICML 2025.
>
> [2] Wang, Xiyue, et al. "A pathology foundation model for cancer diagnosis and prognosis prediction." Nature 2024.
>
> [3] Ding, Tong, et al. "A multimodal whole-slide foundation model for pathology." Nature Medicine 2025.
>
> [4] Xu, Hanwen, et al. "A whole-slide foundation model for digital pathology from real-world data." Nature 2024.

---

> > ### Author Response · Authors · 2025-11-24
> > **Response to Reviewer sJW6 (2/3)**
> >
> > ## Response to Missing Data Setting
> >
> > **Our core work focuses on training with complete multimodal data while only using WSI unimodal data for inference.** This is intended to achieve data-efficiency during inference, not to specifically focus on training using incompletely paired multimodal data.
> >
> > **1. Practical Scenario:**
> > In the clinical diagnostic environment, Whole Slide Images (WSI) have become routine data. Their low acquisition cost and convenient collection method result in a large volume of available data. However, genomic data remains scarce due to its high cost, difficulty in collection, and patient privacy issues. The inverse scenario, where genomic data is available but WSI is missing, is rare in routine clinical practice.
> >
> > **2. Motivation:**
> > We aim to fully leverage existing complete multimodal data (public datasets and historical center data) for model training. During inference, however, we rely solely on low-cost WSI unimodal data. This approach maximizes the diagnostic value of WSI, allowing it to achieve performance comparable to multimodal inference. This fulfills our purpose of achieving data-efficiency during inference.
> >
> > **3. Experimental Setup:**
> > Following your suggestion, we have included experiments detailing different genomic data missing rates. We did not design the method to handle modal omission during the initial stage. Therefore, to conduct this experiment, we adjusted the training strategy for the Target branch. If a case lacks genomic data, the Target branch does not participate in training; only the WSI-input Online branch is trained.
> >
> > The specific experimental results are shown in Table R3. Even when using only partially complete data, MSRL still surpasses all unimodal training methods. Furthermore, MSRL remains superior to existing methods for unimodal inference, even at a 30% missing rate. This demonstrates that MSRL can fully utilize existing multimodal data to learn comprehensive and robust representations. This greatly enhances WSI’s diagnostic value and data efficiency during the inference phase.
> >
> > ##### Table R3: The results of different genomic data missing rates.
> > |Method|Modality|BLCA|BRCA|STAD|HNSC|COADREAD|overall|
> > |---|---|---|---|---|---|---|---|
> > |G-HANet|g.+h.→h.|0.5806±0.0149|0.6418±0.0138|0.6782±0.0489|0.5770±0.0278|0.6216±0.0184|0.6246|
> > |LD-CVAE|g.+h.→h.|0.5954±0.0104|0.6430±0.0146|0.6938±0.0495|0.5960±0.0286|0.6280±0.0211|0.6313|
> > |||||||||
> > |MSRL missing 60%|g.+h.→h.|0.6008±0.0125|0.6508±0.0577|0.6918±0.0699|0.5896±0.0570|0.6186±0.0401|0.6303|
> > |MSRL missing 30%|g.+h.→h.|0.6132±0.0668|0.6752±0.0809|0.7034±0.0830|0.6026±0.0505|0.6448±0.0587|0.6478|
> > |MSRL missing 0%|g.+h.→h.|0.6192±0.0184|0.6808±0.0277|0.7050±0.0523|0.6182±0.0015|0.6554±0.0166|0.6558|
> >
> > *We will add these results to the Table 1 of original manuscript.*

---

> > > ### Author Response · Authors · 2025-11-24
> > > **Response to Reviewer sJW6 (3/3)**
> > >
> > > ## Response to Evaluation fairness
> > > The "non-isolated" nature of MSRL represents one of the core contributions of this study. Existing frameworks rely solely on information from the individual case. Consequently, they fail to fully explore the relevance between cases. To fairly demonstrate the specific contribution of GSL, we conducted a comparative experiment using a static KNN graph construction method instead of GSL as shown in Table R1 (the Table 3 in the original paper). The results show that the C-index drops by 5% when using KNN. This performance is even lower than that of G-HANet and LD-CVAE. This finding indicates that merely accessing information from other cases does not improve performance. Crucially, the improvement relies on the dynamic relevance construction provided by GSL.
> > > ##### Table R1:The results of the structure ablation on five survival prediction datasets (Table 3 in the original paper)
> > > Model|BLCA|BRCA|STAD|HNSC|COADREAD|Overall
> > > ---|---|---|---|---|---|---
> > > KNN(euclidean)|0.569±0.029|0.628±0.021|0.632±0.036|0.562±0.030|0.611±0.011|0.600
> > > KNN(cosin)|0.577±0.023|0.637±0.025|0.641±0.041|0.568±0.031|0.618±0.014|0.608
> > > MSRL_random_GSL|0.598±0.021|0.669±0.033|0.681±0.032|0.599±0.025|0.637±0.015|0.637
> > > MSRL|0.619±0.018|0.681±0.028|0.705±0.052|0.618±0.002|0.655±0.017|0.656
> > >
> > > ---
> > >
> > > ## Response to Buffer size
> > >
> > > One of the core contributions of this paper is leveraging authentic genomics data to assist during inference for cases missing genomic data. The **buffer** is designed specifically for this purpose.
> > >
> > >  We added experiments assessing the impact of changing buffer size on task performance as shown in Table R4. Results indicate that a smaller buffer size leads to weaker model performance. The performance reduction is more significant when the number of training samples is limited. When the buffer size is zero (i.e., removing the buffer), the model performance drops significantly. This demonstrates that introducing authentic data is key to MSRL's effective inference. It also shows that a sufficient data scale better assists the model in learning missing genomic information.
> > >
> > > ##### Table R4: The ablation results of the buffer size on validation datasets
> > > |buffer size|100%|75%|50%|25%|0%|
> > > |---|---|---|---|---|-|
> > > |BLCA(N=357)|0.607|0.595(-0.012)|0.589(-0.018)|0.573(-0.034)|0.533(-0.074)|
> > > |BRCA(N=680)|0.672|0.669(-0.003)|0.664(-0.008)|0.653(-0.019)|0.591(-0.081)|
> > > |STAD(N=318)|0.729|0.713(-0.016)|0.705(-0.024)|0.691(-0.038)|0.611(-0.118)|
> > > |HNSC(N=392)|0.622|0.615(-0.007)|0.611(-0.011)|0.602(-0.020)|0.545(-0.077)|
> > > |COADREAD(N=298)|0.661|0.655(-0.006)|0.640(-0.021)|0.621(-0.039)|0.557(-0.104)|
> > >
> > > *We will add these results to the appendix of original manuscript.*
> > >
> > >
> > > ---
> > >
> > > We would like to express our sincere gratitude once again for the time and effort you have dedicated to this review. We hope that our responses have satisfactorily addressed your concerns. Should you have any further questions, please do not hesitate to discussions with us.
> > >
> > > Best,
> > >
> > > Authors

---

> ### Comment · Reviewer_sJW6 · 2025-11-26
>
> I appreciate the responses from the authors. My concerns are mostly addressed. Therefore, I keep my positive rating unchanged.
>
> By the way, please update your revised manuscript to reflect changes with colored texts.

---

> > ### Author Response · Authors · 2025-11-27
> > **Response to the revised manuscript**
> >
> > Dear reviewer，
> >
> > We appreciate your prompt response. The revised manuscript has been uploaded, with all modifications highlighted in blue. If you have any remaining concerns, we would be happy to discuss them further.
> >
> > Best,
> >
> > Authors

---

### Official Review · Reviewer_GyVr · 2025-11-01

**Soundness:** 2
**Presentation:** 1
**Contribution:** 3
**Rating:** 4
**Confidence:** 5

**Summary:**

This work proposes a Multi-modal Structural Representation Learning (MSRL) framework that pre-trains a histopathology-genomics multi-modal representation graph via graph structure learning to capture inherent inter-case relevance using paired TCGA pan-cancer data. Then, during fine-tuning, with the guidance of multimodal representation, an inductor is learned to reconstruct genomic features, which then serves as a proxy when genomic data is missing during inference.

**Strengths:**

1. The idea of structural alignment based on graph learning is new for addressing the problem of missing genomic data.
2. The performance of the proposed method is better when using genomic data only during inference, although there is a gap compared to the model with complete multimodal data for inference.

**Weaknesses:**

1. It is hard to follow the manuscript with so many notations. Moreover, a lack of intuitive insights behind the model design makes it difficult to understand the problem the authors wanted to address.
2. The related works are not comprehensive for both Fusion-based Multi-modal methods and Multi-modal methods with missing modality. For example, DisPro [1], a multimodal model for survival prediction that is robust to missing modalities, is considerably closer to the proposed method. It should be discussed and compared in the experimental section.
3. So many important details are missing in the methodology section. For example, it is confusing how the buffer assists in the modeling. And what did Readout stand for? Did the model use complete data during training? Or both missing data and complete data.
4. Data contamination issue. Given that the pretrained data is from TCGA and the finetuning datasets also come from TCGA, although they cannot see the label, there is still data contamination. External validation is necessary to validate the generalizable ability of the proposed method. For example, CPTAC is recommended to validate the model’s generalization.
5. Given that the proposed method can handle both missing and complete modalities during inference, the performance of complete modalities should be presented for fair comparison to multimodal fusion approaches. This can deepen the understanding of the performance boundary of the proposed model.
6. The ablation studies are incomplete.

    1) The effectiveness of various loss components in the pretraining should be validated.

    2) It is confusing about what MSRLonline_buffer represents.
    3) Each component of alignment loss (fused feature, structure, and graph alignment) should be investigated.
    4) The effectiveness of the buffer should be explored by removing it.
    5) The target branch can be validated by removing it and the performance combination between the online branch and the genomic data.
    6) To showcase the superiority of the proposed pretraining method, various pathology foundation models should be compared by replacing the pretrained weights, e.g. UNI [2], Virchow[3], and especially mSTAR [4], which also leveraged genomic data to enhance pathology.
7. Given that the model can handle missing modality, various missing ratios should be investigated.

[1] Xu Y, Zhou F, Zhao C, et al. Distilled Prompt Learning for Incomplete Multimodal Survival Prediction[C]//Proceedings of the Computer Vision and Pattern Recognition Conference. 2025: 5102-5111.

[2] Chen R J, Ding T, Lu M Y, et al. Towards a general-purpose foundation model for computational pathology[J]. Nature medicine, 2024, 30(3): 850-862.

[3] Vorontsov E, Bozkurt A, Casson A, et al. A foundation model for clinical-grade computational pathology and rare cancers detection[J]. Nature medicine, 2024, 30(10): 2924-2935.

[4] Xu Y, Wang Y, Zhou F, et al. A multimodal knowledge-enhanced whole-slide pathology foundation model[J]. arXiv preprint arXiv:2407.15362, 2024.

**Questions:**

See weakness.

---

> ### Author Response · Authors · 2025-11-24
> **Response to Reviewer GyVr (1/5)**
>
> Dear Reviewer,
>
> we sincerely thanks for your thorough review and appreciate your recognition our contribution. Your comments are very helpful for improving our paper. We have added more experiments and provided clearer explanations to address your concerns. Our detailed responses are as follows:
>
>
> ---
> ## Response to Presentation and Clarity
>
> **1. About the Notations**
> WSI is the giga-pixel ultra-high resolution image. Genomic data also has thousands of dimensions. This complexity means that constructing case-level representations is an complicated process. Therefore, introducing a substantial number of notations is unavoidable to clearly describe variables at different hierarchical levels. We would be grateful if you could let us know if you identify any errors or ambiguities among these notations.
>
> **2. About the Motivation and Method Design**
> Existing methods addressing the lack of genomic data during inference typically focus only on individual case information. They neglect the inference value of authentic cases. Consequently, our motivation and corresponding solutions are as follows:
>
> * **Motivation I:** Constructing inter-case relevance.
>     **Method:** Introducing dynamic construction via Graph Structure Learning (GSL).
> * **Motivation II:** Leveraging authentic data to guide WSI unimodal inference.
>     **Method:** Establishing a buffer to store authentic data representations and implementing dynamic updates.
>
> In terms of the specific workflow: the pre-training stage utilizes pan-cancer data to mine inter-case relevance and align multimodal representations of cases. This initializes the GSL and the genomic encoder. The fine-tuning stage leverages authentic multimodal data and the buffer structure. It guides the WSI unimodal data to perform efficient task inference.
>
> **3. About the Details in the Methodology Section**
> The **Readout** module operates based on the `case_id` of the current batch samples. It extracts the representations of all cases not included in the current batch from the buffer. These are then combined with the features $\mathbf{f}$ of the current batch. This combined result serves as the input for the next GSL step, as described in line 295 of the original manuscript.
>
>
> ---
>
>
> ## Response to Details in the Buffer
>
> One of the core contributions of this paper is leveraging authentic genomics data to assist during inference for cases missing genomic data. The **buffer** is designed specifically for this purpose. The details are as follows:
>
> 1.  **Buffer Content.** The training samples for the downstream task consist of complete WSI-Genomics multimodal data. The buffer stores the multimodal representations encoded from the training samples' WSI and authentic genomic data. (As stated in lines 280–281 of the original manuscript.)
>
> 2.  **Buffer Function.** Representations within the buffer are jointly sent to the GSL with the case representations that have missing genomic data. Relevance construction with the authentic data representations facilitates learning the missing gene information representation. This process then enables task inference. (As stated in lines 295–296 of the original manuscript.)
>
> 3.  **Buffer Update.** During training, the Target branch, which is input with real paired multimodal data, learns representations that update the buffer. (As stated in lines 292–293 of the original manuscript.)
>
> 4.  **Buffer-Related Experimental Validation.** In the original paper, we ablated the buffer's update mechanism (see Table 3) and evaluated the impact of different buffer sizes on inference efficiency (see Table A1). Furthermore, we added experiments assessing the impact of changing buffer size on task performance as shown in Table R1. Results indicate that a smaller buffer size leads to weaker model performance. The performance reduction is more significant when the number of training samples is limited. When the buffer size is zero (i.e., **removing the buffer**), the model performance drops significantly. Furthermore, our approach constructs structural correlations between the current case and historical cases. Consequently, *removing the buffer leads to the loss of functionality for both GSL and GCN*. This demonstrates that introducing authentic data is key to MSRL's effective inference. It also shows that a sufficient data scale better assists the model in learning missing genomic information.
>
> ##### Table R1: The ablation results of the buffer size on validation datasets
> |buffer size|100%|75%|50%|25%|0%|
> |---|---|---|---|---|-|
> |BLCA(N=357)|0.607|0.595(-0.012)|0.589(-0.018)|0.573(-0.034)|0.533(-0.074)|
> |BRCA(N=680)|0.672|0.669(-0.003)|0.664(-0.008)|0.653(-0.019)|0.591(-0.081)|
> |STAD(N=318)|0.729|0.713(-0.016)|0.705(-0.024)|0.691(-0.038)|0.611(-0.118)|
> |HNSC(N=392)|0.622|0.615(-0.007)|0.611(-0.011)|0.602(-0.020)|0.545(-0.077)|
> |COADREAD(N=298)|0.661|0.655(-0.006)|0.640(-0.021)|0.621(-0.039)|0.557(-0.104)|
>
> *We will add these results to the appendix of original manuscript.*

---

> ### Author Response · Authors · 2025-11-24
> **Response to Reviewer GyVr (2/5)**
>
> ## Response to Comparison methods
> 1. Thank you for your valuable suggestion to increase the number of comparison methods. Therefore, we have added two new multimodal methods, MOTCat [1] and PIBD [2]. Regarding the suggestion to compare with foundation models: our method utilizes **slide-level** pre-training. The pathology foundation models you suggested are **patch-level** pre-trained, which makes a direct comparison infeasible. Therefore, we have added comparisons with the latest published slide-level foundation models FEATHER [3], CHIEF [4], and TITAN [5], as shown in Table R2.
>    - Based on the results, LD-CVAE [6] and SurvPath [7], as presented in the original paper, remain the state-of-the-art (SOTA) among multimodal fusion methods.
>    - MSRL_H is able to outperform existing foundation models even without introducing multimodal training for downstream tasks. This demonstrates that MSRL's pre-training effectively enhances the representation capability of the slide encoder.
> ##### Table R2: The C-Index (mean ± std) on five survival prediction tasks.
> |Method|Modality|BLCA|BRCA|STAD|HNSC|COADREAD|overall|
> |---|---|---|---|---|---|---|---|
> |FEATHER|h.|0.5306±0.0340|0.5698±0.0247|0.5570±0.0420|0.5277±0.0302|0.5796±0.0284|0.5530|
> |CHIEF|h.|0.5606±0.0890|0.5762±0.0768|0.5668±0.0601|0.5338±0.0838|0.5822±0.0672|0.5639|
> |TITAN|h.|0.5756±0.0864|0.6182±0.0711|0.6306±0.0838|0.5652±0.0381|0.6138±0.0340|0.6007|
> |MSRL_H|h.|0.5774±0.0221|0.6398±0.0251|0.6626±0.0427|0.5676±0.0289|0.6182±0.0174|0.6131|
> |||||||||
> |MOTCat|g.+h.|0.6097±0.0540|0.6689±0.0671|0.7086±0.0522|0.6175±0.0637|0.6489±0.0346|0.6507|
> |PIBD|g.+h.|0.6116±0.0318|0.6738±0.0406|0.7188±0.0267|0.6244±0.0434|0.6578±0.0654|0.6573|
> |||||||||
> |LD-CVAE_multi|g.+h.|0.6210±0.0131|0.6712±0.0199|0.7201±0.0395|0.6302±0.0303|0.6602±0.0224|0.6605|
> |SurvPath|g.+h.|0.6288±0.0184|0.6866±0.0209|0.7194±0.0524|0.6328±0.0256|0.6712±0.0150|0.6683|
> |||||||||
> |MSRL|g.+h.→h.|0.6192±0.0184|0.6808±0.0277|0.7050±0.0523|0.6182±0.0015|0.6554±0.0166|0.6558|
>
> *We will add these results to the Table 1 of original manuscript.*
>
>
> 2. As for DisPro[8], we firstly directly compared our results with those reported in their paper. As shown in Table R3, MSRL's performance on TCGA BRCA, which is the closest dataset used by both, is closer to that of SurvPath than DisPro. This, to some extent, demonstrates MSRL's superiority. We are currently reproducing the DisPro results under our experimental settings to ensure a more objective and fair comparison. However, DisPro requires training three models for a single task, with each model involving fine-tuning an LLM. This leads to a longer experimental cycle under the five-fold cross-validation setting. We guarantee that these related experiments will be completed and results released within the next week.
> ##### Table R3: The comparison of DisPro and MSRL.
> ||DisPro|MSRL|
> |---|---|---|
> |Dataset |TCGA-BLCA(n=372)|TCGA-BLCA(n=357)|
> |patch encoder|Vit (pretrained in UNI)|Vit (pre-trained in GigaPath)|
> |WSI aggregation|LLM+Top-K pooling|LongNet (pre-trained in GigaPath)|
> |Gene setting| 330 pathways|50 pathways|
> |c-Index of SurvPath|0.6572|0.6288|
> |c-Index of proposed method|0.6315|0.6192|
> |performance gap with SurvPath|-2.57%|-0.96%|
>
> [1] Xu, Yingxue, and Hao Chen. "Multimodal optimal transport-based co-attention transformer with global structure consistency for survival prediction." CVPR 2023
>
> [2] Zhang, Yilan, et al. "Prototypical information bottlenecking and disentangling for multimodal cancer survival prediction." ICLR 2024
>
> [3] Shao, Daniel, et al. "Do Multiple Instance Learning Models Transfer?." ICML 2025.
>
> [4] Wang, Xiyue, et al. "A pathology foundation model for cancer diagnosis and prognosis prediction." Nature 2024.
>
> [5] Ding, Tong, et al. "A multimodal whole-slide foundation model for pathology." Nature Medicine 2025.
>
> [6] Zhou, Junjie, et al. "Robust Multimodal Survival Prediction with Conditional Latent Differentiation Variational AutoEncoder." CVPR 2025.
>
> [7] Jaume,et al. "Modeling dense multimodal interactions between biological pathways and histology for survival prediction." CVPR 2024.
>
> [8] Xu, Yingxue, et al. "Distilled Prompt Learning for Incomplete Multimodal Survival Prediction." CVPR 2025.

---

> ### Author Response · Authors · 2025-11-24
> **Response to Reviewer GyVr (3/5)**
>
> ## Response to Data contamination and generalization
> **1. The experimental setup ensures there is no data contamination whatsoever at the **case-level** (patient-level).** All fine-tuning datasets were split at the case-level into training-validation and test sets using a 4:1 ratio. The 3,607 cases in the training-validation sets were included in the pre-training data (totaling 6,361 cases). All 902 cases in the test sets did not participate in pre-training. These sets constitute the entirety of the 7,263 cases (6,361 + 902) used in this paper. Furthermore, the baseline GigaPath, which MSRL uses, is pretrained on in-house WSIs from Providence, a large US health network [9]. This dataset has no intersection with TCGA, providing further assurance that all patient information used for testing is absent from the pre-training stage.
>
> **2. Mechanism of Generalization**
>
> - **Our framework presents no additional difficulty in generalization.**
> Genomic data from out-of-distribution cases often suffers from complex sources and various protocols. Direct inclusion of such data can cause significant batch effects. In contrast, our method utilizes solely WSI unimodal inference. Its generalization only rely on the representation capability of the slide encoder. Therefore, the generalization capability of our method when facing out-of-distribution data is consistent with other WSI encoder-based inference frameworks and does not introduce additional challenges.
>
> - **Our cross-modal retrieval strategy even boosts the generalization ability.**
> Existing unimodal inference methods typically focus on reconstructing raw genomic data for individual cases, where the recontruction performance is also affected by OOD issues. In contrast, MSRL models genomic correlations between cases within an aligned representation space and then use the real source-domain genomic representation for muilti-modal prediciton. This strategy naturally avoids the multi-source heterogeneity inherent in raw genomic data during training. Therefore, it ensures better generalization than reconstruction-based methods during the inference phase.
>
> **3. External Evaluation Experiments**
> We constructed new external datasets to fully validate the generalizability of MSRL. The details are as follows:
>
> - **External Datasets:** We collected two public datasets from the CPTAC project: **CPTAC-HNSC** (comprising WSI data and clinical survival information for 106 patients) and **CPTAC-BRCA** (containing WSI data and diagnostic stage information for 111 patients). We ensured these datasets share no overlap with the TCGA project.
>
> - **Experimental Setting:** To validate the generalizability of MSRL and baseline methods on out-of-distribution cases, we directly applied models trained on TCGA-HNSC and TCGA-BRCA to the CPTAC-HNSC and CPTAC-BRCA datasets for testing, respectively. Additionally, we incorporated several foundation models [3][4][5] known for high generalizability for comparison. The specific experimental results are shown in Table R4.
>
> - **Results Analysis:**
>    * MSRL consistently achieved the best performance on both external datasets. Notably, MSRL_H secured the top results even under conditions limited to WSI unimodal training.
>    * In scenarios addressing missing inference modalities, existing methods like G-HANet and LD-CAVE suffered performance drops exceeding 5% and 7% on CPTAC-HNSC and CPTAC-BRCA, respectively. In contrast, MSRL showed minimal declines of only 1.02% and 1.62%, outperforming most foundation models. These results demonstrate the superior generalizability of MSRL.
>
> ##### Table R4: The results on external test sets, where "Δ" denotes performance declines.
> |method|TCGA-HNSC C-index|CPTAC-HNSC C-index|Δ|TCGA-BRCA AUC|CPTAC-BRCA AUC|Δ|
> |---|---|---|---|---|---|---|
> |FEATHER|0.5277±0.0302|0.5197±0.0214|-0.80%|0.627±0.0085|0.5782±0.0142|-4.88%|
> |CHIEF|0.5338±0.0388|0.5246±0.0301|-0.92%|0.602±0.0169|0.5608±0.0198|-4.12%|
> |GigaPath|0.5580±0.0330|0.5278±0.0545|-3.02%|0.625±0.0074|0.5824±0.0457|-4.26%|
> |TITAN|0.5652±0.0381|0.5341±0.0610|-3.11%|0.648±0.0044|0.6094±0.0541|-3.86%|
> ||||||||
> |MSRL_H|0.5676±0.0289|0.5461±0.0125|-2.15%|0.652±0.0095|0.6099±0.0401|-4.21%|
> |G-HANet|0.5770±0.0278|0.5238±0.0786|-5.32%|0.632±0.0263|0.5612±0.0634|-7.08%|
> |LD-CVAE|0.5960±0.0286|0.5418±0.0469|-5.42%|0.646±0.0309|0.5728±0.0309|-7.32%|
> |MSRL|0.6182±0.0015|0.6080±0.0118|-1.02%|0.664±0.0263|0.6478±0.0416|-1.62%|
>
> [9] Xu, Hanwen, et al. "A whole-slide foundation model for digital pathology from real-world data." Nature 2024.

---

> > ### Author Response · Authors · 2025-11-24
> > **Response to Reviewer GyVr (4/5)**
> >
> > ## Response to Missing Data Setting
> >
> > **Our core work focuses on training with complete multimodal data while only using WSI unimodal data for inference.** This is intended to achieve data-efficiency during inference, not to specifically focus on training using incompletely paired multimodal data.
> >
> > **1. Practical Scenario:**
> > In the clinical diagnostic environment, Whole Slide Images (WSI) have become routine data. Their low acquisition cost and convenient collection method result in a large volume of available data. However, genomic data remains scarce due to its high cost, difficulty in collection, and patient privacy issues. The inverse scenario, where genomic data is available but WSI is missing, is rare in routine clinical practice.
> >
> > **2. Motivation:**
> > We aim to fully leverage existing complete multimodal data (public datasets and historical center data) for model training. During inference, however, we rely solely on low-cost WSI unimodal data. This approach maximizes the diagnostic value of WSI, allowing it to achieve performance comparable to multimodal inference. This fulfills our purpose of achieving data-efficiency during inference.
> >
> > **3. Experimental Setup:**
> > Following your suggestion, we have included experiments detailing different genomic data missing rates. We did not design the method to handle modal omission during the initial stage. Therefore, to conduct this experiment, we adjusted the training strategy for the Target branch. If a case lacks genomic data, the Target branch does not participate in training; only the WSI-input Online branch is trained.
> >
> > The specific experimental results are shown in Table R5. Even when using only partially complete data, MSRL still surpasses all unimodal training methods. Furthermore, MSRL remains superior to existing methods for unimodal inference, even at a 30% missing rate. This demonstrates that MSRL can fully utilize existing multimodal data to learn comprehensive and robust representations. This greatly enhances WSI’s diagnostic value and data efficiency during the inference phase.
> >
> > **4. MSRL\_multi** We implemented a complete multiple modalities version of the method in the original manuscript, which we named **MSRL\_multi** (see lines 345–346). This version outperforms existing multimodal fusion methods across all datasets (see Table 1 in the original manuscript).
> >
> > ##### Table R5: The results of different genomic data missing rates.
> > |Method|Modality|BLCA|BRCA|STAD|HNSC|COADREAD|overall|
> > |---|---|---|---|---|---|---|---|
> > |MSRL_multi|g.+h.|0.6368±0.0327|0.7012±0.0302|0.7236±0.0411|0.6456±0.0263|0.6896±0.0301|0.6794|
> > |||||||||
> > |G-HANet|g.+h.→h.|0.5806±0.0149|0.6418±0.0138|0.6782±0.0489|0.5770±0.0278|0.6216±0.0184|0.6246|
> > |LD-CVAE|g.+h.→h.|0.5954±0.0104|0.6430±0.0146|0.6938±0.0495|0.5960±0.0286|0.6280±0.0211|0.6313|
> > |||||||||
> > |MSRL missing 60%|g.+h.→h.|0.6008±0.0125|0.6508±0.0577|0.6918±0.0699|0.5896±0.0570|0.6186±0.0401|0.6303|
> > |MSRL missing 30%|g.+h.→h.|0.6132±0.0668|0.6752±0.0809|0.7034±0.0830|0.6026±0.0505|0.6448±0.0587|0.6478|
> > |MSRL missing 0%|g.+h.→h.|0.6192±0.0184|0.6808±0.0277|0.7050±0.0523|0.6182±0.0015|0.6554±0.0166|0.6558|
> >
> > *We will add these results to the Table 1 of original manuscript.*

---

> ### Author Response · Authors · 2025-11-24
> **Response to Reviewer GyVr (5/5)**
>
> ## Response to ablation studies
> 1. **Pre-training** We have added the ablation study for the three loss components used during pre-training, as shown in Table R6. The results indicate that removing the individual loss $\mathcal{L}\_{fuse}$ leads to the most significant performance reduction. This is because $\mathcal{L}\_{fuse}$ constrains the multimodal representation of the cases, playing a vital role in integrating multimodal information for the downstream tasks. $\mathcal{L}\_{inter}$ is the key constraint for aligning the cross-modal representations of the same case. $\mathcal{L}\_{intra}$ enhances the discriminative power of the individual modal data representations. Removing any of these losses negatively impacts the downstream task performance.
>
> ##### Table R6: The ablation results of the loss function during the pre-training on validation datasets
> |Dataset|All losses|$\mathcal{L}\_{intra}$+$\mathcal{L}\_{fuse}$|$\mathcal{L}\_{inter}$+$\mathcal{L}\_{fuse}$|$\mathcal{L}\_{inter}$+$\mathcal{L}\_{intra}$|$\mathcal{L}\_{inter}$|$\mathcal{L}\_{intra}$|$\mathcal{L}\_{fuse}$|
> |---|---|---|---|---|---|---|---|
> |BLCA|0.607|0.582(↓0.025)|0.571(↓0.036)|0.567(↓0.040)|0.522(↓0.085)|0.514(↓0.093)|0.556(↓0.051)|
> |BRCA|0.672|0.640(↓0.032)|0.646(↓0.026)|0.636(↓0.036)|0.538(↓0.134)|0.524(↓0.148)|0.563(↓0.109)|
> |STAD|0.729|0.656(↓0.073)|0.670(↓0.059)|0.646(↓0.083)|0.560(↓0.169)|0.557(↓0.172)|0.585(↓0.144)|
> |HNSC|0.622|0.603(↓0.019)|0.594(↓0.028)|0.569(↓0.053)|0.530(↓0.092)|0.536(↓0.086)|0.559(↓0.063)|
> |COADREAD|0.661|0.621(↓0.040)|0.629(↓0.032)|0.613(↓0.048)|0.545(↓0.116)|0.543(↓0.118)|0.573(↓0.088)|
>
> *We will add these results to the appendix of original manuscript.*
>
> 2. **MSRL\_online\_buffer Update Mechanism** The MSRL\_online\_buffer updates the buffer using features from the online branch rather than the target branch features, as indicated in line 429 of the original manuscript. This specific setting was introduced to validate the necessity of the buffer storing authentic data representations derived from the target branch.
>
> 3. **Hierarchical Alignment Loss** The ablation experiments for each component of the Hierarchical Alignment loss are presented in Table 4 of the original manuscript. The results clearly demonstrate the necessity of every component.
>
> 4. The ablation experiment for removing the buffer is shown in Table R1 (where **Buffer Size = 0%**).
>
> 5. **Target branch** We constructed the ablation experiment for removing the Target branch following your suggestion, as shown in Table R7. The Target branch receives complete multimodal data as input. Its core function is to guide the WSI unimodal input of the Online branch to learn multimodal information representation and achieve efficient unimodal inference. **When the Target branch is removed, the unimodal inference task cannot be accomplished.** Compared to the multimodal MSRL\_multi, removing the Target branch leads to a significant decrease in performance. Furthermore, the standard deviation increases significantly, indicating that the model's robustness is severely compromised.
>
> ##### Table R7: The results of removing target branch.
> |Method|Modality|BLCA|BRCA|STAD|HNSC|COADREAD|overall|
> |---|---|---|---|---|---|---|---|
> |MSRL_multi|g.+h.|0.6368±0.0327|0.7012±0.0302|0.7236±0.0411|0.6456±0.0263|0.6896±0.0301|0.6794|
> |MSRL_multi w/o Target|g.+h.|0.6025±0.0932|0.6631±0.1021|0.7102±0.0907|0.6154±0.1015|0.6378±0.0982|0.6458|
>
> ---
>
> We would like to express our sincere gratitude once again for the time and effort you have dedicated to this review. We hope that our responses have satisfactorily addressed your concerns. We look forward to further discussions with you.
>
> Best,
>
> Authors

---

> > ### Author Response · Authors · 2025-11-27
> > **Response to comparison with DisPro**
> >
> > We have completed the comparison experiments with DisPro [1] and added the results to Table RA and Table RB.
> >
> > DisPro distills prognostic knowledge into the genomics prompt during training. However, its inference procedure relies on self-scores computed within each individual case to select and aggregate tokens. This design overlooks holistic case-level representations and inter-case relationships, which results in a c-index that is 1.44\% lower than MSRL’s as shown in Table RA. In addition, DisPro performs multi-stage uni-modal training, which makes it sensitive to the data distributions of both modalities and weakens its generalization ability at inference time. As shown in Table RB, DisPro’s c-index drops by 4.77% when evaluated on out-of-domain data. In contrast, MSRL only a 1.02% performance decrease under the same setting, which exhibits strong generalization capacity.
> >
> > ##### Table RA: The C-Index (mean ± std) on five survival prediction tasks. (A part of the Table 1 in the paper)
> > |Method|Modality|BLCA|BRCA|STAD|HNSC|COADREAD|overall|
> > |---|---|---|---|---|---|---|---|
> > |**DisPro_multi**|**g.+h.**|**0.6267±0.0423**|**0.6931±0.0372**|**0.7097±0.0403**| **0.6390±0.0580**| **0.6770±0.0479**|**0.6691**|
> > |MSRL_multi|g.+h.|0.6368±0.0327| 0.7012±0.0302| 0.7236±0.0411|0.6456±0.0263|0.6896±0.0301|0.6794|
> > |||||||||
> > |**DisPro**|**g.+h.→h.**|**0.6058±0.0269**| **0.6734±0.0352**|**0.6803±0.0424**|**0.6053±0.0610**|**0.6418±0.0342**|**0.6414**|
> > |MSRL|g.+h.→h.|0.6192±0.0184|0.6808±0.0277|0.7050±0.0523|0.6182±0.0015|0.6554±0.0166|0.6558|
> >
> >
> > ##### Table RB: The generalization analysis results, where CPTAC-HNSC and CPTAC-BRCA are external test sets and "Δ" denotes performance declines. (A part of the Table A1 in the paper)
> > |method|TCGA-HNSC C-index|CPTAC-HNSC C-index|Δ|TCGA-BRCA AUC|CPTAC-BRCA AUC|Δ|
> > |---|---|---|---|---|---|---|
> > |FEATHER|0.5277±0.0302|0.5197±0.0214|-0.80%|0.627±0.0085|0.5782±0.0142|-4.88%|
> > |CHIEF|0.5338±0.0388|0.5246±0.0301|-0.92%|0.602±0.0169|0.5608±0.0198|-4.12%|
> > |GigaPath|0.5580±0.0330|0.5278±0.0545|-3.02%|0.625±0.0074|0.5824±0.0457|-4.26%|
> > |TITAN|0.5652±0.0381|0.5341±0.0610|-3.11%|0.648±0.0044|0.6094±0.0541|-3.86%|
> > ||||||||
> > |MSRL_H|0.5676±0.0289|0.5461±0.0125|-2.15%|0.652±0.0095|0.6099±0.0401|-4.21%|
> > |G-HANet|0.5770±0.0278|0.5238±0.0786|-5.32%|0.632±0.0263|0.5612±0.0634|-7.08%|
> > |LD-CVAE|0.5960±0.0286|0.5418±0.0469|-5.42%|0.646±0.0309|0.5728±0.0309|-7.32%|
> > |**DisPro**|**0.6053±0.0610**|**0.5576±0.0692**|**-4.77%**|--|--|--|
> > |MSRL|0.6182±0.0015|0.6080±0.0118|-1.02%|0.664±0.0263|0.6478±0.0416|-1.62%|
> >
> > *As DisPro’s prompts are specifically designed for survival prediction, we therefore excluded it from the cancer staging tasks.*
> >
> > The revised manuscript has been uploaded, with all modifications highlighted in blue. If you have any remaining concerns, we would be happy to discuss them further.
> >
> > Best,
> >
> > Authors
> >
> > [1] Xu, Yingxue, et al. "Distilled Prompt Learning for Incomplete Multimodal Survival Prediction." CVPR 2025.

---

> > > ### Comment · Reviewer_GyVr · 2025-11-28
> > >
> > > Thanks to the authors’ detailed responses. Some of my concerns have been addressed. However, I still have some questions as follows:
> > >
> > > **Regarding the motivation and method design**, I understand that each methodology may potentially overlook certain dimensions. The key issue is not whether any aspects are ignored, but rather whether the overlooked aspects are of sufficient significance for the task you are resolving. In other words, it is essential to articulate clearly why the particular aspects you emphasize warrant consideration, which is the basis of this work.
> > >
> > > **Regarding the compared PFMs**, although the PFMs I suggested are patch-level models, the proposed method is still supposed to be compared to them. For example, the results of patch-level PFMs can be presented by using the standard ABMIL [1] as the aggregator trained on the features extracted by these PFMs. Since ABMIL is very simple and highly efficient, if the proposed approach, despite its complexity, cannot outperform these PFMs with simple ABMIL, what then is the rationale for the proposed method’s existence? Based on this, I consider such an experiment to be fundamentally necessary.
> > >
> > > Additionally, I wonder whether the non-FM methods included in the comparison consistently employ the Gigapath patch feature extractor, maintaining alignment with the approach adopted in the proposed method for fair comparison.
> > >
> > > [1] Attention-based Deep Multiple Instance Learning, ICML, 2018

---

> ### Author Response · Authors · 2025-11-28
> **Response to Reviewer GyVr**
>
> Dear Reviewer,
>
> Thank you for your prompt reply. Our detailed responses are as follows:
>
> ---
> ### **Response to the motivation concern**
>
> The basis of our method is leveraging inter-case relevance to link a WSI with the most relevant authentic genomic information, enabling data-efficient single-modality inference using only WSI while achieving performance comparable to multimodal models.
>
>
> The designs throughout the work are all aimed at achieving this goal, and the two core components can be summarized as:
> - a **GSL-based dynamic graph** to model inter-case relevance, and
> - a **dynamically updated buffer** to store representations derived from authentic genomic data.
>
> In the revised manuscript, we now state the motivation of each module more clearly before its technical design, covering the two-stage framework, the losses used in pre-training, the buffer mechanism, and the dual-branch fine-tuning strategy. These revisions can be found in **Section 3 (METHODS)** starting from **line 156**, with all changes highlighted.
>
> If there are specific modules in which motivations remain unclear, we would appreciate it if you could point them out, and we will be happy to clarify further.
>
> ---
>
> ### **Response to the concern regarding comparisons with PFMs**
>
> We **did not omit comparisons with PFMs**. In the revised manuscript, we had added quantitative comparisons with several **slide-level PFMs** and these results are reported in **Table 1, Table 2, and Table A1**. These PFMs are trained by further pre-training a slide encoder on top of patch-level PFMs. Their frameworks are summarized in **Table RC**, and we use their respective patch encoders for consistent comparison.
>
> Notably, both **CHIEF** and **FEATHER** adopt **ABMIL** for the slide encoder, as suggested by the reviewer. Our results show that **MSRL consistently outperforms ABMIL-based PFMs in both internal and external evaluations (as shown in Tables R4 and R5).** Furthermore, when using the same baseline PFM (**GigaPath**), MSRL achieves clear improvements across all tasks, demonstrating its advantage over current SOTA PFMs.
>
> #### **Table RC. Frameworks of the compared PFMs.**
> | PFM | Patch encoder | Slide encoder |
> |---|---|---|
> | CHIEF [1] | ViT (pre-trained in CTransPath [5]) | ABMIL |
> | FEATHER [2] | ViT (pre-trained in UNI [6]) | ABMIL |
> | TITAN [3] | ViT (pre-trained in CONCH [7]) | ViT |
> | GigaPath [4] | ViT (pre-trained in GigaPath) | LongNet |
>
> Regarding patch-level PFMs: as shown in Fig. 2(a) in the manuscript, the pre-training stage of our framework requires a trained slide encoder to construct case-level representations. Using patch-level PFMs alone would break this structural requirement, so our comparison focuses on **slide-level PFMs**, which are built upon patch-level PFMs.
>
> ---
>
> ### **Response to the concern regarding non-FM methods**
>
> All non-FM baselines adopt the **same GigaPath patch feature extractor** to ensure fair comparison. Following your suggestion, we have added an explicit clarification in the revised manuscript (lines 894–895).
>
> ---
>
> If the reviewer have additional concerns or would like further experimental verification, we would be grateful if you could indicate them, and we will address them promptly.
>
> Best,
>
> Authors
>
>
> [1] Wang, Xiyue, et al. "A pathology foundation model for cancer diagnosis and prognosis prediction." Nature 2024.
>
> [2] Shao, Daniel, et al. "Do Multiple Instance Learning Models Transfer?." ICML 2025.
>
> [3] Ding, Tong, et al. "A multimodal whole-slide foundation model for pathology." Nature Medicine 2025.
>
> [4] Xu, Hanwen, et al. "A whole-slide foundation model for digital pathology from real-world data." Nature 2024.
>
> [5] Wang, X. et al. "Transformer-based unsupervised contrastive learning for histopathological image classification." Med. Image Anal. 2022.
>
> [6] Chen, Richard J., et al. "Towards a general-purpose foundation model for computational pathology." Nature medicine 2024.
>
> [7] Lu, Ming Y., et al. "A visual-language foundation model for computational pathology." Nature medicine 2024.

---

### Official Review · Reviewer_vuS4 · 2025-11-03

**Soundness:** 2
**Presentation:** 1
**Contribution:** 2
**Rating:** 4
**Confidence:** 5

**Summary:**

This paper introduces Multi-modal Structural Representation Learning (MSRL), a novel framework to address the issue of missing genomics data in precision oncology. The core contribution is a two-stage approach: first, it pre-trains a graph structure learner on paired WSI-genomics data to capture inter-case relationships. Second, during fine-tuning and inference, it leverages this learned structure and a feature buffer of authentic training cases to guide predictions from WSI data alone. The method demonstrates improved performance over existing reconstruction-based approaches on survival prediction and other diagnostic tasks.

**Strengths:**

The primary strength of this paper lies in its novel reframing of the missing modality problem in computational pathology. Instead of treating each case in isolation, the authors introduce a structural, context-aware paradigm. The originality stems from the creative integration of self-supervised Graph Structure Learning (GSL) with a dual-branch fine-tuning strategy, effectively creating a system that learns how to leverage a knowledge base of related cases for inference. This is a significant departure from standard reconstruction methods. The quality of the work is supported by extensive experiments on multiple TCGA cohorts, demonstrating not only superior performance but also the value of each component through thorough ablation studies.

**Weaknesses:**

**1. Methodological Concerns on Scalability and Buffer Management:**

The core mechanism of MSRL relies on a feature buffer of historical cases during inference, which raises concerns about its long-term viability. The paper lacks a discussion on the strategy for maintaining and updating this crucial component over time. The current approach seems to imply a static buffer constructed from the training set, which would quickly become outdated in a real clinical setting where new data arrives continuously and data distributions may shift (e.g., due to new equipment or patient demographics). To address this, the authors should elaborate on a long-term maintenance strategy. For instance, would a First-In-First-Out (FIFO) queue be appropriate, or could a more intelligent, uncertainty-based or diversity-based sampling strategy be used to keep the buffer representative and up-to-date?

**2. Experimental Concerns on Fairness of Comparison and Robustness:**

A critical issue lies in the fairness of the experimental comparison. MSRL's inference process is "non-isolated" as it actively queries a knowledge base (the feature buffer) of labeled training samples. In contrast, the baseline methods (e.g., G-HANet, LD-CVAE), while trained on the same data, perform inference in an "isolated" manner for each new case. This paradigm difference may grant MSRL an inherent and potentially unfair advantage. To disentangle the benefits of the proposed GSL-based model from the benefits of this "open-book exam" paradigm, the authors should implement a stronger, paradigm-aligned baseline. A simple yet powerful baseline would be to use a standard image retrieval model to find the *K* most similar cases for a new WSI from the training set, and then simply average their corresponding ground-truth genomic data to serve as the reconstructed features. This would help clarify whether the performance gain truly comes from the sophisticated structural learning or primarily from leveraging nearest neighbors' information.

Additionally, the study's reliance on the relatively standardized TCGA dataset limits its generalizability. Real-world clinical data is notoriously heterogeneous, suffering from significant batch effects. It is unclear how robust MSRL is to such noise, as the GSL module could be misled by technical artifacts rather than true biological signals. The authors should strengthen their evaluation by conducting cross-cohort or cross-institutional validation experiments to provide a more convincing assessment of the method's real-world applicability.

**3. Lack of In-depth Interpretability Analysis:**

While the paper's title emphasizes "Structural Representation Learning," the analysis of the learned structure remains superficial. The core strength of this method should be its ability to uncover non-obvious, clinically meaningful relationships between patients. The authors should conduct several in-depth case studies to enhance interpretability. For instance, can they showcase a pair of patients with histologically distinct tumors (perhaps even from different cancer types) that MSRL connects with a strong edge? If so, can this connection be explained by shared molecular pathways, genetic mutations, or similar treatment responses documented in biomedical literature? Such analysis would greatly elevate the paper's scientific impact.

**4. Room for Improvement in Presentation and Clarity:**

The overall presentation of the paper could be significantly improved. The connection between the stated motivation and the proposed method is not immediately clear, requiring the reader to reread sections to grasp the core idea. The description of the methodology, along with the quality of the framework diagrams, could be refined for better clarity and intuition. A more direct and streamlined narrative would enhance the readability and accessibility of this otherwise promising work.

**Questions:**

1. **On Buffer Management:** Could you elaborate on the long-term strategy for the feature buffer? Given that a static buffer may become outdated in clinical practice due to data distribution shifts, have you considered dynamic updating mechanisms like a FIFO queue or more advanced sampling strategies (e.g., based on uncertainty or diversity) to maintain its relevance?

2. **On Fairness of Comparison:** The comparison to "isolated" inference baselines may be unfair due to MSRL's "non-isolated" use of a feature buffer. To better isolate the contribution of your proposed structural learning, would you be willing to add a stronger, paradigm-aligned baseline? For example, one that uses a standard image retrieval model to find K-Nearest Neighbors and averages their ground-truth genomics data for reconstruction. How do you expect MSRL would perform against such a baseline?

3. **On Robustness to Heterogeneity:** The experiments are confined to the standardized TCGA dataset. How would MSRL's Graph Structure Learner (GSL) perform in the presence of significant batch effects from heterogeneous real-world data? Have you considered or can you discuss how the model might distinguish true biological signals from technical artifacts during graph construction?

4. **On Interpretability of the Learned Structure:** Could you provide a concrete case study from your results to demonstrate the model's ability to uncover non-obvious, clinically relevant relationships? For example, can you highlight a pair of histologically dissimilar patients that were strongly connected by the GSL and explain the potential underlying molecular basis for this connection?

**Details Of Ethics Concerns:**

N/A.

---

> ### Author Response · Authors · 2025-11-24
> **Response to Reviewer vuS4 (1/4)**
>
> Dear Reviewer,
>
> we sincerely thanks for your thorough review and appreciate your recognition of the novelty and experimental construction. Your comments are very helpful for improving our paper. We have added more experiments and provided clearer explanations to address your concerns. Our detailed responses are as follows:
>
>
> ---
> ## Response to Scalability and Buffer Management
>
> **1. Dynamic Updating Mechanism**
> The buffer operates dynamically rather than acting as static storage. As stated in lines 292–293 in the original paper ("The feature buffer updated by target branch denoted as $\mathbf{D}\_{buffer}^{updated}$"), we update the corresponding case representations in the buffer during each training batch. This utilizes the learned features from authentic multimodal data (the output of the Target branch). This process ensures the buffer representations remain up-to-date. Consequently, this update mechanism essentially follows a First-In-First-Out (FIFO) strategy.
>
> **2. Rationale and Efficiency**
> Our primary focus is utilizing the buffer to provide authentic data for constructing inter-case relevance. We currently employ a basic buffer update method. This approach effectively highlights the contribution of introducing historical authentic data for relevance construction. We plan to refine the specific buffer maintenance method in future work. Additionally, we conducted experiments on the relationship between buffer size and model inference efficiency (see Table A1 in the original paper). The results indicate that even increasing the buffer size to 50 times the training data volume results in an inference time increase of less than 2%. Therefore, the buffer does not create a bottleneck for task inference efficiency.
>
>
> ---
> ## Response to Fairness of Comparison and Robustness (1/2)
>
> **1. Fairness of Comparison**
> The "non-isolated" nature of MSRL represents one of the core contributions of this study. Existing frameworks rely solely on information from the individual case. Consequently, they fail to fully explore the relevance between cases. To fairly demonstrate the specific contribution of GSL, we conducted a comparative experiment using a static KNN graph construction method instead of GSL (see Table 3 in the original paper). The results show that the C-index drops by 5% when using KNN. This performance is even lower than that of G-HANet and LD-CVAE. This finding indicates that merely accessing information from other cases does not improve performance. Crucially, the improvement relies on the dynamic relevance construction provided by GSL.

---

> ### Author Response · Authors · 2025-11-24
> **Response to Reviewer vuS4 (2/4)**
>
> ## Response to Fairness of Comparison and Robustness (2/2)
>
>
> **2. Mechanism of Generalization**
>
> - **Our framework presents no additional difficulty in generalization.**
> Genomic data from out-of-distribution cases often suffers from complex sources and various protocols. Direct inclusion of such data can cause significant batch effects. In contrast, our method utilizes solely WSI unimodal inference. Its generalization only rely on the representation capability of the slide encoder. Therefore, the generalization capability of our method when facing out-of-distribution data is consistent with other WSI encoder-based inference frameworks and does not introduce additional challenges.
>
> - **Our cross-modal retrieval strategy even boosts the generalization ability.**
> Existing unimodal inference methods typically focus on reconstructing raw genomic data for individual cases, where the recontruction performance is also affected by OOD issues. In contrast, MSRL models genomic correlations between cases within an aligned representation space and then use the real source-domain genomic representation for muilti-modal prediciton. This strategy naturally avoids the multi-source heterogeneity inherent in raw genomic data during training. Therefore, it ensures better generalization than reconstruction-based methods during the inference phase.
>
> **3. External Evaluation Experiments**
> We constructed new external datasets to fully validate the generalizability of MSRL. The details are as follows:
>
> - **External Datasets:** We collected two public datasets from the CPTAC project: **CPTAC-HNSC** (comprising WSI data and clinical survival information for 106 patients) and **CPTAC-BRCA** (containing WSI data and diagnostic stage information for 111 patients). We ensured these datasets share no overlap with the TCGA project.
>
> - **Experimental Setting:** To validate the generalizability of MSRL and baseline methods on out-of-distribution cases, we directly applied models trained on TCGA-HNSC and TCGA-BRCA to the CPTAC-HNSC and CPTAC-BRCA datasets for testing, respectively. Additionally, we incorporated several foundation models known for high generalizability for comparison, such as FEATHER [1], CHIEF [2], TITAN [3], and GigaPath [4]. The specific experimental results are shown in Table R1.
>
> - **Results Analysis:**
>    * MSRL consistently achieved the best performance on both external datasets. Notably, MSRL_H secured the top results even under conditions limited to WSI unimodal training.
>    * In scenarios addressing missing inference modalities, existing methods like G-HANet and LD-CAVE suffered performance drops exceeding 5% and 7% on CPTAC-HNSC and CPTAC-BRCA, respectively. In contrast, MSRL showed minimal declines of only 1.02% and 1.62%, outperforming most foundation models. These results demonstrate the superior generalizability of MSRL.
>
> ##### Table R1: The results on external test sets, where "Δ" denotes performance declines.
> |method|TCGA-HNSC C-index|CPTAC-HNSC C-index|Δ|TCGA-BRCA AUC|CPTAC-BRCA AUC|Δ|
> |---|---|---|---|---|---|---|
> |FEATHER|0.5277±0.0302|0.5197±0.0214|-0.80%|0.627±0.0085|0.5782±0.0142|-4.88%|
> |CHIEF|0.5338±0.0388|0.5246±0.0301|-0.92%|0.602±0.0169|0.5608±0.0198|-4.12%|
> |GigaPath|0.5580±0.0330|0.5278±0.0545|-3.02%|0.625±0.0074|0.5824±0.0457|-4.26%|
> |TITAN|0.5652±0.0381|0.5341±0.0610|-3.11%|0.648±0.0044|0.6094±0.0541|-3.86%|
> ||||||||
> |MSRL_H|0.5676±0.0289|0.5461±0.0125|-2.15%|0.652±0.0095|0.6099±0.0401|-4.21%|
> |G-HANet|0.5770±0.0278|0.5238±0.0786|-5.32%|0.632±0.0263|0.5612±0.0634|-7.08%|
> |LD-CVAE|0.5960±0.0286|0.5418±0.0469|-5.42%|0.646±0.0309|0.5728±0.0309|-7.32%|
> |MSRL|0.6182±0.0015|0.6080±0.0118|-1.02%|0.664±0.0263|0.6478±0.0416|-1.62%|
>
> *We will add these results to the appendix of original manuscript.*
>
> [1] Shao, Daniel, et al. "Do Multiple Instance Learning Models Transfer?." ICML 2025.
>
> [2] Wang, Xiyue, et al. "A pathology foundation model for cancer diagnosis and prognosis prediction." Nature 2024.
>
> [3] Ding, Tong, et al. "A multimodal whole-slide foundation model for pathology." Nature Medicine 2025.
>
> [4] Xu, Hanwen, et al. "A whole-slide foundation model for digital pathology from real-world data." Nature 2024.

---

> > ### Author Response · Authors · 2025-11-24
> > **Response to Reviewer vuS4 (3/4)**
> >
> > ## Response to Interpretability Analysis
> >
> > We provided a comprehensive analysis of the interpretability of the learned structure. This is detailed in Appendix B (Visualization) of the original manuscript.
> >
> > **1. Macroscopic Structural Correlation Analysis.**
> > Figure A1 in the original paper illustrates the structural correlations. These correlations exist between WSI representations and gene expression within the pan-cancer dataset following pre-training. Key findings include:
> > - **The pre-trained GSL can efficiently capture the morphological relationships of the cases.** The fused-modal GSL constructs denser edges for nodes within the same cluster, and also constructs denser associations for different clusters of the same site that are far from each other. It indicates that the fused-GSL can effectively capture the structural associations between cases in histo-morphology;
> >
> > - **The pre-trained GSL can efficiently capture the structural relevance of cases at the molecular level.** The correlation between gene expression levels within a given cluster is high. Furthermore, the different gene expression levels of distant clusters from the same site reflect the genetic heterogeneity between cancer subtypes.
> >
> > - **The molecular relevance enhance the connectivity of the foundation model embeddings.** The distant cases in the feature space are also linked, which are corrected by the influence of gene correlation. Even for patients exhibiting heterogeneity in pathological morphology, GSL effectively captures their relevance at the genomic level..These connections enhanced by genomics data play a crucial role in facilitating MSRL to achieve the precision multi-modal fusion.
> >
> > **2. Specific Case Correlation Analysis**
> > Figure A2 in the original manuscript details the case relevance constructed by GSL. This is demonstrated using the COADREAD prognosis task. Key findings include:
> > - **The GSL can effectively learn the structural relationships from authentic data.** The online branch’s WSI-based Inductor can effectively capture the genomics information from authentic data. And then GSL can build meaningful relationships between previously unseen test cases and diagnostic training cases. This ensures that MSRL can perform efficient multi-modal task inference in real-world scenarios with WSI unimodality.
> >
> > - **Fine-tuned GSL can capture task-specific knowledge.** The GSL significantly separates the test cases into high-risk group and low-risk group, which show high association with group 1 (columns 1 to 4) and group 2 (columns 11 to 13) of the training cases. The average survival time of cases in Group 1 is only 18.10 months and it is 64.65 months in Group 2. This demonstrates that the fine-tuned GSL can learn task-related diagnostic information among GSL and effectively promote precision oncology.

---

> > > ### Author Response · Authors · 2025-11-24
> > > **Response to Reviewer vuS4 (4/4)**
> > >
> > > ### Response to Presentation and Clarity
> > >
> > > **1. Connection between Motivation and Specific Methods** We have established clear motivations and proposed corresponding solutions.
> > >
> > > - **Motivation I:** Constructing inter-case relevance.
> > >    **Method:** Introducing dynamic construction via **Graph Structure Learning (GSL)**.
> > >
> > > - **Motivation II:** Leveraging authentic data to guide WSI unimodal inference.
> > >    **Method:** Establishing a **buffer** to store authentic data representations and implementing dynamic updates.
> > >
> > > **2. Reorganization of the Method Description** We have streamlined the methodology section and adopted a clearer, simplified procedural description below:
> > >
> > > - **Pre-training:** This stage utilizes pan-cancer data to mine inter-case relevance and align multimodal representations of cases. It serves to initialize the GSL and the genomics encoder. We define the $Graph\_{learn}(\mathcal{G}, aug=False)$ as follows:
> > >    1. **Input**
> > >       - $\mathcal{G}$: graph data, including node features $\mathbf{X}$ and adjacency matrix $\mathbf{A}$;
> > >       - $aug$: bool value, denoting the $\mathcal{G}$ is augmented or not;
> > >    2. **Processing**
> > >       - if $aug$:
> > >          - $\mathbf{A}^r = \mathbf{A}$
> > >       - else:
> > >          - $\mathbf{A}^r = GSL(\mathbf{X})$;
> > >       - $\mathbf{Z} = GCN(\mathbf{A}^r, \mathbf{X})$;
> > >    3. **Output**
> > >       - $\mathbf{Z}$: graph feature.
> > >
> > >    The pre-training stage is as follows:
> > >    - step 1: Initialize histopathology graph $\mathcal{G}\_H$ and augmented $\mathcal{G}\_H^{aug}$, gene graph $\mathcal{G}\_G$ and augmented $\mathcal{G}\_g^{aug}$, and fused multi-modal graph $\mathcal{G}\_F$;
> > >
> > >    - step 2: $\mathbf{Z}\_H = Graph\_{learn}(\mathcal{G}\_H), \mathbf{Z}\_H^{aug} = Graph\_{learn}(\mathcal{G}\_H^{aug}, Ture)$, $\mathbf{Z}\_G = Graph\_{learn}(\mathcal{G}\_G), \mathbf{Z}\_G^{aug} = Graph\_{learn}(\mathcal{G}\_G^{aug}, Ture)$, $\mathbf{Z}\_F = Graph\_{learn}(\mathcal{G}\_F)$;
> > >
> > >    - step 3: calculate $\mathcal{L}\_{\text{intra}}
> > >    =\tfrac{1}{2}\big(
> > >    \mathcal{L}\_{\text{InfoNCE}}( \mathbf{Z}\_{H}; \mathbf{Z}\_{H}^{aug})
> > >    +\mathcal{L}\_{\text{InfoNCE}}( \mathbf{Z}\_{G}; \mathbf{Z}\_{G}^{aug})
> > >    \big)$,
> > >    $\mathcal{L}\_{\text{inter}}
> > >    =\tfrac{1}{2}\big(
> > >    \mathcal{L}\_{\text{InfoNCE}}( \mathbf{Z}\_{H}; \mathbf{Z}\_{G})
> > >    +\mathcal{L}\_{\text{InfoNCE}}( \mathbf{Z}\_{G}; \mathbf{Z}\_{H})
> > >    \big)$,
> > >    $\mathcal{L}\_{\text{fused}}
> > >    =\tfrac{1}{2}\big(
> > >    \mathcal{L}\_{\text{InfoNCE}}( \mathbf{Z}\_{F}; \mathbf{Z}\_{H})
> > >    +\mathcal{L}\_{\text{InfoNCE}}( \mathbf{Z}\_{F}; \mathbf{Z}\_{G})
> > >    \big)$.
> > >
> > >
> > >
> > > - **Fine-tuning:** This stage leverages authentic multimodal data and the buffer structure. It guides the WSI unimodal data to perform efficient task inference. We define the $Branch(\mathbf{h}, \mathbf{g}, D\_{buffer})$ as follows:
> > >    1. **Input**
> > >       - $\mathbf{h}$: slide features;
> > >       - $\mathbf{g}$: gene features;
> > >       - $D\_{buffer}$: feature buffer of training cases;
> > >    2. **Processing**
> > >       - $\mathbf{f} = concatenate(\mathbf{g}, \mathbf{h})$;
> > >       - $\mathbf{F} = Readout(D\_{buffer}, \mathbf{f})$, denoting the combination of the current cases and other cases in the buffer;
> > >       - $\mathbf{A}^r = GSL(\mathbf{F})$;
> > >       - $\mathbf{Z} = GCN(\mathbf{A}^r, \mathbf{F})$;
> > >    3. **Output**
> > >       - $\mathbf{f}$: node features;
> > >       - $\mathbf{A}^r$: refined adjacency matrix;
> > >       - $\mathbf{Z}$: graph feature.
> > >
> > >    Then, the fine-tune stage is as follows:
> > >    - step 1: Initialize each module with the pre-trained weights;
> > >
> > >    - step 2: $\mathbf{h} = \phi\_H( X\_I^{aug}), \hat{\mathbf{h}} = \phi\_H( \hat{X}\_I^{aug}), \mathbf{g} = \phi\_G( X\_G)$;
> > >
> > >    - step 3: $\mathbf{f}$, $\mathbf{A}^r$, $\mathbf{Z}$ = **$Branch\_{online}(\mathbf{h}, Inductor(\mathbf{h}), D\_{buffer})$**;
> > >
> > >    - step 4: $\hat{\mathbf{f}}$, $\hat{\mathbf{A}}^r$, $\hat{\mathbf{Z}}$ = **$Branch\_{target}(\hat{\mathbf{h}}, \mathbf{g}, D\_{buffer})$**;
> > >
> > >    - step 5: $D\_{buffer} = update(D\_{buffer}, \hat{\mathbf{f}})$;
> > >
> > >    - step 6: $\mathcal{L}\_{falign}(\mathbf{f}, \hat{\mathbf{f}})$, $\mathcal{L}\_{galign}(\mathbf{Z}, \hat{\mathbf{Z}})$, $\mathcal{L}\_{salign}(\mathbf{A}^r, \hat{\mathbf{A}}^r)$, $\mathcal{L}\_{task}(\mathbf{Z}, y)$, where $y$ denotes the task labels;
> > >
> > > *We will incorporate this workflow into subsections 3.2 and 3.3 of the methodology section in the original manuscript. Furthermore, we will reorganize and streamline the content to ensure greater conciseness and clarity.*
> > >
> > > ---
> > >
> > > We would like to express our sincere gratitude once again for the time and effort you have dedicated to this review. We hope that our responses have satisfactorily addressed your concerns. Should you have any further questions, please do not hesitate to discussions with us.
> > >
> > > Best,
> > >
> > > Authors

---

> > > > ### Author Response · Authors · 2025-11-27
> > > > **Response to the revised manuscript**
> > > >
> > > > Dear Reviewer,
> > > >
> > > > The revised manuscript has been uploaded, with all modifications highlighted in blue. We have carefully refined and reorganized the method description section, simplified the notations, and adopted a clearer and more explicit presentation of the workflow to better convey our contributions. We hope that the revised manuscript addresses your concerns.
> > > >
> > > > Best,
> > > >
> > > > Authors

---

### Official Review · Reviewer_vXzx · 2025-11-05

**Soundness:** 3
**Presentation:** 3
**Contribution:** 3
**Rating:** 6
**Confidence:** 3

**Summary:**

The authors propose MSRL, a novel architecture for multi-modal survival analysis and modality completion. To address the limitations of prior methods, the underutilization of gene data and the neglect of inter-patient relationships, this paper introduces a new approach that trains a gene encoder using self-supervised graph learning. The framework is validated on both modality completion for survival analysis and WSI classification.

**Strengths:**

The proposed method aligns well with its stated motivation, offering targeted solutions for the challenges in modality completion. The staged training approach is novel, consists of first pre-training a gene encoder with self-supervised graph learning, followed by aligning encoders trained on single modalities. The experiments are comprehensive, covering two downstream tasks and including KM-plot analysis and GSL visualizations in the supplementary materials.

**Weaknesses:**

The paper directly uses cross-modal kNN to construct cross-modal maps. However, due to the inherent differences between WSI and genes, it is difficult to guarantee the balance of this mapping method.
The ablation studies are missing experiments on the losses used during the representation pre-training. Additionally, I am curious about the function and impact of the "Buffer" component.

**Questions:**

Explain why it is reasonable to directly connect WSI and genes using kNN while the large differences between the two modalities may lead to unbalanced mapping.
The experimental validation would be significantly strengthened by additional ablation studies. Specifically, an ablation on the individual components of the pre-training loss is required to dissect their respective contributions. Additionally, the role of the buffer component should be explicitly clarified, and its impact empirically validated

---

> ### Author Response · Authors · 2025-11-24
> **Response to Reviewer vXzx (1/2)**
>
> Dear Reviewer,
>
> we sincerely thanks for the time and effort you dedicated to reviewing this manuscript. We are also grateful for your recognition of the novelty of our method and the validation of our experiments. Your comments are very helpful for improving our paper. We have added more experiments and provided clearer explanations to address your concerns. Our detailed responses are as follows:
>
>
> ---
> ## Response to KNN Construction
>
> **1. KNN Application in Pre-training**
> In the first pre-training stage, we used the KNN method to initialize the graph adjacency matrix separately for the two modalities: WSI and genomics, resulting in $A\_H$ and $A\_G$. We did not perform cross-modal KNN graph construction.
>
> **2. Relevance Construction and Balance**
> In this paper, we construct **inter-case relevance**. Each node in the graph represents a case-level feature. Since the pre-training stage uses complete multimodal data, the corresponding nodes in the histopathology graph and the gene graph are always paired. Therefore, there is no risk of an unbalanced graph structure.
>
> ---
> ## Response to Ablation Studies of Pre-training
> We have added the ablation study for the three loss components used during pre-training, as shown in Table R1. The results indicate that removing the individual loss $\mathcal{L}\_{fuse}$ leads to the most significant performance reduction. This is because $\mathcal{L}\_{fuse}$ constrains the multimodal representation of the cases, playing a vital role in integrating multimodal information for the downstream tasks. $\mathcal{L}\_{inter}$ is the key constraint for aligning the cross-modal representations of the same case. $\mathcal{L}\_{intra}$ enhances the discriminative power of the individual modal data representations. Removing any of these losses negatively impacts the downstream task performance.
>
> ##### Table R1: The ablation results of the loss function during the pre-training on validation datasets
> |Dataset|All losses|$\mathcal{L}\_{intra}$+$\mathcal{L}\_{fuse}$|$\mathcal{L}\_{inter}$+$\mathcal{L}\_{fuse}$|$\mathcal{L}\_{inter}$+$\mathcal{L}\_{intra}$|$\mathcal{L}\_{inter}$|$\mathcal{L}\_{intra}$|$\mathcal{L}\_{fuse}$|
> |---|---|---|---|---|---|---|---|
> |BLCA|0.607|0.582(↓0.025)|0.571(↓0.036)|0.567(↓0.040)|0.522(↓0.085)|0.514(↓0.093)|0.556(↓0.051)|
> |BRCA|0.672|0.640(↓0.032)|0.646(↓0.026)|0.636(↓0.036)|0.538(↓0.134)|0.524(↓0.148)|0.563(↓0.109)|
> |STAD|0.729|0.656(↓0.073)|0.670(↓0.059)|0.646(↓0.083)|0.560(↓0.169)|0.557(↓0.172)|0.585(↓0.144)|
> |HNSC|0.622|0.603(↓0.019)|0.594(↓0.028)|0.569(↓0.053)|0.530(↓0.092)|0.536(↓0.086)|0.559(↓0.063)|
> |COADREAD|0.661|0.621(↓0.040)|0.629(↓0.032)|0.613(↓0.048)|0.545(↓0.116)|0.543(↓0.118)|0.573(↓0.088)|
>
> *We will add these results to the appendix of original manuscript.*

---

> > ### Author Response · Authors · 2025-11-24
> > **Response to Reviewer vXzx (2/2)**
> >
> > ## Response to the Buffer
> >
> > One of the core contributions of this paper is leveraging authentic genomics data to assist during inference for cases missing genomic data. The **buffer** is designed specifically for this purpose.
> >
> > **1. Buffer Content**
> > In the downstream task, the training samples consist of complete WSI-Genomics multimodal data. The buffer stores the multimodal representations, which are encoded from the training samples' WSI and authentic genomic data.
> >
> > **2. Buffer Function**
> > The representations within the buffer, along with the case representations that have missing genomic data, are jointly fed into the **Graph Structure Learner (GSL)**. By constructing relevance with the authentic data representations, the model learns the missing genomic information representation and enables task inference.
> >
> > **3. Buffer Update**
> > During training, the buffer is updated using the representations learned by the Target branch, which is input with the authentic, paired multimodal data.
> >
> > **4. Buffer-Related Experimental Validation**
> > In the original paper, we ablated the buffer's update mechanism (see Table 3 in the original paper) and evaluated the impact of different buffer sizes on inference efficiency (see Table A1 in the original paper). Furthermore, we have added experiments assessing the impact of changing the buffer size on task performance, as shown in Table R2. The results indicate that a smaller buffer size leads to weaker model performance. The performance reduction is more significant when the number of training samples is limited. This confirms that sufficient authentic data better assists the model in learning the missing genomic information.
> > ##### Table R2: The ablation results of the buffer size on validation datasets
> > |buffer size|100%|75%|50%|25%|
> > |---|---|---|---|---|
> > |BLCA(N=357)|0.607|0.595(-0.012)|0.589(-0.018)|0.573(-0.034)|
> > |BRCA(N=680)|0.672|0.669(-0.003)|0.664(-0.008)|0.653(-0.019)|
> > |STAD(N=318)|0.729|0.713(-0.016)|0.705(-0.024)|0.691(-0.038)|
> > |HNSC(N=392)|0.622|0.615(-0.007)|0.611(-0.011)|0.602(-0.020)|
> > |COADREAD(N=298)|0.661|0.655(-0.006)|0.640(-0.021)|0.621(-0.039)|
> >
> > *We will add these results to the appendix of original manuscript.*
> >
> > ---
> > We would like to express our sincere gratitude once again for the time and effort you have dedicated to this review. We hope that our responses have satisfactorily addressed your concerns. Should you have any further questions, please do not hesitate to discussions with us.
> >
> > Best,
> >
> > Authors

---

> > > ### Author Response · Authors · 2025-11-27
> > > **Response to the revised manuscript**
> > >
> > > Dear Reviewer,
> > >
> > > The revised manuscript has been uploaded, with all modifications highlighted in blue. If you have any remaining concerns, we would be happy to discuss them further.
> > >
> > > Best,
> > >
> > > Authors

---

### Official Review · Reviewer_EbNx · 2025-11-05

**Soundness:** 2
**Presentation:** 2
**Contribution:** 2
**Rating:** 2
**Confidence:** 4

**Summary:**

This paper proposes MSRL, a framework that integrates histopathology whole-slide images (WSIs) and genomics data to improve precision oncology, especially when genomics data are missing. Unlike prior methods that reconstruct genomics features from single cases, MSRL leverages inter-case relationships through graph structure learning to model structural relevance among patients. Based on large-scale evaluation from TCGA across survival prediction and precision diagnosis tasks, MSRL outperformed existing missing-modality reconstruction approaches in C-index and achieved performance comparable to full multi-modal fusion models.

**Strengths:**

1.	Introduces a Multi-modal Structural Representation Learning (MSRL) framework that uniquely combines graph structure learning with histopathology–genomics fusion, enabling inference from WSIs alone by leveraging inter-case relationships.

2.	Conducts extensive experiments on over 7,000 TCGA cases across multiple cancer types and tasks (survival prediction, staging, and mutation status), with comparisons to both unimodal and multi-modal baselines.

**Weaknesses:**

1.	While the paper provides a detailed overview of existing reconstruction-based solutions for handling missing genomics modalities, the discussion omits another important line of research — knowledge distillation or modality-distillation frameworks that transfer multi-modal information into a single-modality model for inference [1-2]. These approaches (e.g., modality distillation or teacher–student setups trained with full modalities but deployed with WSIs only) address the same practical problem more elegantly, without explicitly synthesizing genomics data. Including and contrasting these methods would strengthen the contextual positioning of MSRL, clarify its unique advantages over distillation-based solutions, and provide a more balanced view of the current landscape of missing-modality learning in computational pathology.

$\quad$ [1] Jin, Cheng, et al. "Genome-Anchored Foundation Model Embeddings Improve Molecular Prediction from Histology Images." arXiv preprint arXiv:2506.19681 (2025).

$\quad$ [2] Xu, Yingxue, et al. "Distilled Prompt Learning for Incomplete Multimodal Survival Prediction." Proceedings of the Computer Vision and Pattern Recognition Conference. 2025.

2.	The paper states that MSRL leverages authentic genomics data from diagnostically related cases during inference through graph structure learning. However, it remains unclear how this mechanism avoids potential information leakage between training and inference phases. In a realistic clinical setting, test patients may come from a distribution distinct from the training cohort, and their “related” cases’ genomics data would not be available. The manuscript would benefit from clarifying whether the model restricts graph construction to training data only, how it ensures that no test information is indirectly used, and whether any domain-shift analysis (e.g., across cancer subtypes or institutions) was performed to assess robustness when patient distributions differ. Without this clarification, it is difficult to judge the practical feasibility and generalizability of using authentic genomics profiles as auxiliary signals during inference.

3.	The paper pre-trains the MSRL model on a large-scale TCGA pan-cancer dataset and then evaluates it on TCGA subsets (e.g., BRCA, STAD, HNSC, COADREAD). While the authors mention that “all test data are excluded from the first-stage pre-training,” the training and evaluation still originate from the same overarching dataset and share highly similar data distributions. This setup risks information leakage or at least unfairly optimistic performance, as patient-level overlaps, stain distributions, or cancer-specific morphological priors could be implicitly retained from pre-training. To ensure a fair assessment of generalization, it would be valuable to include evaluations on external cohorts or at least cross-center TCGA splits to verify that the model’s performance is not inflated by intra-dataset correlations.

4.	The AUC values reported for BRCA and NSCLC staging tasks (around 0.66) are surprisingly low compared with prior WSI-based studies, where standard MIL models such as ABMIL often achieve near-perfect discrimination (>0.95 AUC) [1] for such coarse-grained classification tasks. This discrepancy raises concerns about dataset construction, label definition, or preprocessing consistency. The paper should clarify whether these staging labels were derived from TCGA clinical metadata (which may be incomplete or imbalanced), or whether additional filtering or downsampling was applied. Without this clarification, it is difficult to interpret the reported improvements or compare them meaningfully with prior literature.

$\quad$ [1] Chen, Richard J., et al. "Towards a general-purpose foundation model for computational pathology." Nature medicine 30.3 (2024): 850-862.

5.	The literature review in this work is not thorough enough. First of all, for the distillation-based missing modality modeling:

$\quad$ [1] Jin, Cheng, et al. "Genome-Anchored Foundation Model Embeddings Improve Molecular Prediction from Histology Images." arXiv preprint arXiv:2506.19681 (2025).

$\quad$ [2] Xu, Yingxue, et al. "Distilled Prompt Learning for Incomplete Multimodal Survival Prediction." Proceedings of the Computer Vision and Pattern Recognition Conference. 2025.

Meanwhile, for fusion-based multi-modal methods:

$\quad$ [3] Xu, Yingxue, and Hao Chen. "Multimodal optimal transport-based co-attention transformer with global structure consistency for survival prediction." Proceedings of the IEEE/CVF international conference on computer vision. 2023.

$\quad$ [4] Zhang, Yilan, et al. "Prototypical information bottlenecking and disentangling for multimodal cancer survival prediction." arXiv preprint arXiv:2401.01646 (2024).

Moreover, the MIL methods used for comparison did not show in the Related Work section.

**Questions:**

Please see the Weaknesses section.

---

> ### Author Response · Authors · 2025-11-24
> **Response to Reviewer EbNx (1/3)**
>
> Dear Reviewer,
>
> we sincerely thanks for your thorough review. Your comments are very helpful for improving our paper. We have added more experiments and provided clearer explanations to address your concerns. Our detailed responses are as follows:
>
> ---
> ## Response to the Literature Review
>
> **1. Clarification on Distillation-Based Methods**
> Distillation is a type of network constraint and it does not conflict with the solutions summarized in this paper for handling missing genomic data. Reference [1] uses a learnable prompt to substitute the missing genomic data and relies on supervision from a auxiliary task (survival loss). This approach falls into the category illustrated in Figure 1(b) of our original paper. Reference [2] uses similar "pseudo-pathway embeddings" to replace missing genomic data and explicitly constructs a reconstruction loss $\mathcal{L}\_rec$ (see page 15 of that paper). This strategy belongs to the category shown in Figure 1(c) of our original paper. In summary, the literature review section of our paper provides a high-level overview of current major methods for addressing modal omission during inference. However, it failed to include certain specific articles. We appreciate the reviewer's reminder and will further refine this section.
>
> **2. Comparison Methods**
> We have added comparative experiments for the two multimodal fusion methods you suggested, **MOTCat**[3] and **PIBD**[4], as shown in Table R1. Based on the results, LD-CVAE[5] and SurvPath[6], as presented in the original paper, remain the state-of-the-art (SOTA) among multimodal fusion methods.
> ##### Table R1: The C-Index (mean ± std) on five survival prediction tasks.
> |Method|Modality|BLCA|BRCA|STAD|HNSC|COADREAD|overall|
> |---|---|---|---|---|---|---|---|
> |MOTCat|g.+h.|0.6097±0.0540|0.6689±0.0671|0.7086±0.0522|0.6175±0.0637|0.6489±0.0346|0.6507|
> |PIBD|g.+h.|0.6116±0.0318|0.6738±0.0406|0.7188±0.0267|0.6244±0.0434|0.6578±0.0654|0.6573|
> |||||||||
> |LD-CVAE_multi|g.+h.|0.6210 ± 0.0131|0.6712 ± 0.0199|0.7201 ± 0.0395|0.6302 ± 0.0303|0.6602 ± 0.0224|0.6605|
> |SurvPath|g.+h.|0.6288 ± 0.0184|0.6866 ± 0.0209|0.7194 ± 0.0524|0.6328 ± 0.0256|0.6712 ± 0.0150|0.6683|
> |||||||||
> |MSRL|g.+h.→h.|0.6192±0.0184|0.6808±0.0277|0.7050±0.0523|0.6182±0.0015|0.6554±0.0166|0.6558|
>
> *We will add these results to the Table1 of original manuscript.*
>
> Moreover, we first directly compared our results with those reported in the original DisPro[1] paper. As shown in Table R2, MSRL's performance on TCGA BRCA, which is the closest dataset used by both, is closer to that of SurvPath than DisPro. This, to some extent, demonstrates MSRL's superiority. We are currently reproducing the DisPro results under our experimental settings to ensure a more objective and fair comparison. However, DisPro requires training three models for a single task, with each model involving fine-tuning an LLM. This leads to a longer experimental cycle under the five-fold cross-validation setting. We guarantee that these related experiments will be completed and results released within the next week.
> ##### Table R2: The comparison of DisPro and MSRL.
> ||DisPro|MSRL|
> |---|---|---|
> |Dataset |TCGA-BLCA(n=372)|TCGA-BLCA(n=357)|
> |patch encoder|Vit (pretrained in UNI)|Vit (pre-trained in GigaPath)|
> |WSI aggregation|LLM+Top-K pooling|LongNet (pre-trained in GigaPath)|
> |Gene setting| 330 pathways|50 pathways|
> |c-Index of SurvPath|0.6572|0.6288|
> |c-Index of proposed method|0.6315|0.6192|
> |performance gap with SurvPath|-2.57%|-0.96%|
>
> We were unable to perform experimental comparisons with Reference [2] because its code has not yet been made public.
>
>
> [1]  Xu, Yingxue, et al. "Distilled Prompt Learning for Incomplete Multimodal Survival Prediction." CVPR 2025.
>
> [2] Jin, Cheng, et al. "Genome-Anchored Foundation Model Embeddings Improve Molecular Prediction from Histology Images." arXiv preprint arXiv:2506.19681 (2025).
>
> [3] Xu, Yingxue, and Hao Chen. "Multimodal optimal transport-based co-attention transformer with global structure consistency for survival prediction." CVPR 2023.
>
> [4] Zhang, Yilan, et al. "Prototypical information bottlenecking and disentangling for multimodal cancer survival prediction." ICLR 2024.
>
> [5] LD-CVAE: Zhou, Junjie, et al. "Robust Multimodal Survival Prediction with Conditional Latent Differentiation Variational AutoEncoder." CVPR 2025.
>
> [6] Jaume,et al. "Modeling dense multimodal interactions between biological pathways and histology for survival prediction." CVPR 2024.

---

> ### Author Response · Authors · 2025-11-24
> **Response to Reviewer EbNx (2/3)**
>
> ## Response to Information Leakage and Generalizability of MSRL
>
>
> **1. The experimental setup ensures there is no information leakage whatsoever at the **case-level** (patient-level).** All fine-tuning datasets were split at the case-level into training-validation and test sets using a 4:1 ratio. The 3,607 cases in the training-validation sets were included in the pre-training data (totaling 6,361 cases). All 902 cases in the test sets did not participate in pre-training. These sets constitute the entirety of the 7,263 cases (6,361 + 902) used in this paper. Furthermore, the baseline GigaPath, which MSRL uses, is pretrained on in-house WSIs from Providence, a large US health network [7]. This dataset has no intersection with TCGA, providing further assurance that all patient information used for testing is absent from the pre-training stage.
>
> **2. Mechanism of Generalization**
>
> - **Our framework presents no additional difficulty in generalization.**
> Genomic data from out-of-distribution cases often suffers from complex sources and various protocols. Direct inclusion of such data can cause significant batch effects. In contrast, our method utilizes solely WSI unimodal inference. Its generalization only rely on the representation capability of the slide encoder. Therefore, the generalization capability of our method when facing out-of-distribution data is consistent with other WSI encoder-based inference frameworks and does not introduce additional challenges.
>
> - **Our cross-modal retrieval strategy even boosts the generalization ability.**
> Existing unimodal inference methods typically focus on reconstructing raw genomic data for individual cases, where the recontruction performance is also affected by OOD issues. In contrast, MSRL models genomic correlations between cases within an aligned representation space and then use the real source-domain genomic representation for muilti-modal prediciton. This strategy naturally avoids the multi-source heterogeneity inherent in raw genomic data during training. Therefore, it ensures better generalization than reconstruction-based methods during the inference phase.
>
> **3. External Evaluation Experiments**
> We constructed new external datasets to fully validate the generalizability of MSRL. The details are as follows:
>
> - **External Datasets:** We collected two public datasets from the CPTAC project: **CPTAC-HNSC** (comprising WSI data and clinical survival information for 106 patients) and **CPTAC-BRCA** (containing WSI data and diagnostic stage information for 111 patients). We ensured these datasets share no overlap with the TCGA project.
>
> - **Experimental Setting:** To validate the generalizability of MSRL and baseline methods on out-of-distribution cases, we directly applied models trained on TCGA-HNSC and TCGA-BRCA to the CPTAC-HNSC and CPTAC-BRCA datasets for testing, respectively. Additionally, we incorporated several foundation models known for high generalizability for comparison, such as FEATHER [8], CHIEF [9] and TITAN [10]. The specific experimental results are shown in Table R3.
>
> - **Results Analysis:**
>     *  MSRL consistently achieved the best performance on both external datasets. Notably, MSRL_H secured the top results even under conditions limited to WSI unimodal training.
>     *  In scenarios addressing missing inference modalities, existing methods like G-HANet and LD-CAVE suffered performance drops exceeding 5% and 7% on CPTAC-HNSC and CPTAC-BRCA, respectively. In contrast, MSRL showed minimal declines of only 1.02% and 1.62%, outperforming most foundation models. These results demonstrate the superior generalizability of MSRL.
>
> ##### Table R3: The results on external test sets, where "Δ" denotes performance declines.
> |method|TCGA-HNSC C-index|CPTAC-HNSC C-index|Δ|TCGA-BRCA AUC|CPTAC-BRCA AUC|Δ|
> |---|---|---|---|---|---|---|
> |FEATHER|0.5277±0.0302|0.5197±0.0214|-0.80%|0.627±0.0085|0.5782±0.0142|-4.88%|
> |CHIEF|0.5338±0.0388|0.5246±0.0301|-0.92%|0.602±0.0169|0.5608±0.0198|-4.12%|
> |GigaPath|0.5580±0.0330|0.5278±0.0545|-3.02%|0.625±0.0074|0.5824±0.0457|-4.26%|
> |TITAN|0.5652±0.0381|0.5341±0.0610|-3.11%|0.648±0.0044|0.6094±0.0541|-3.86%|
> ||||||||
> |MSRL_H|0.5676±0.0289|0.5461±0.0125|-2.15%|0.652±0.0095|0.6099±0.0401|-4.21%|
> |G-HANet|0.5770±0.0278|0.5238±0.0786|-5.32%|0.632±0.0263|0.5612±0.0634|-7.08%|
> |LD-CVAE|0.5960±0.0286|0.5418±0.0469|-5.42%|0.646±0.0309|0.5728±0.0309|-7.32%|
> |MSRL|0.6182±0.0015|0.6080±0.0118|-1.02%|0.664±0.0263|0.6478±0.0416|-1.62%|
>
>
> *We will add these results to the appendix of original manuscript.*
>
> [7] Xu, Hanwen, et al. "A whole-slide foundation model for digital pathology from real-world data." Nature 2024.
>
> [8] Shao, Daniel, et al. "Do Multiple Instance Learning Models Transfer?." ICML 2025.
>
> [9] Wang, Xiyue, et al. "A pathology foundation model for cancer diagnosis and prognosis prediction." Nature 2024.
>
> [10] Ding, Tong, et al. "A multimodal whole-slide foundation model for pathology." Nature Medicine 2025.

---

> > ### Author Response · Authors · 2025-11-24
> > **Response to Reviewer EbNx (3/3)**
> >
> > ## Response to Rigor of the Experimental Setup
> >
> > **1. The Distinction between Cancer Staging and Cancer Subtyping** The experiments on the TCGA BRCA and NSCLC datasets were **focused on cancer staging rather than subtyping** in our paper. Reference [11] ultimately reported results for cancer subtyping tasks on TCGA BRCA and NSCLC, achieving AUCs greater than 0.95. Subtyping involves determining the histological or molecular subtype of a tumor, a task where the histological information contained in a single WSI is often sufficient for judgment. In contrast, cancer staging assesses the extent of tumor spread based on the TNM standard, including primary tumor size, lymph node involvement, and distant metastasis. This typically requires integrating information from multiple slides, organs, and regions. The classification difficulty is significantly greater than that of subtyping. The current state-of-the-art (SOTA) metrics for cancer staging on TCGA BRCA and NSCLC generally show AUCs below 0.65 [12][13]. MSRL's performance has already reached an advanced level for these tasks.
> >
> > **2. Data Integrity and Fairness** We can assure that all public data utilized is authentic and has not been artificially altered. The results for all comparison methods are objectively reported. This ensures the fairness and validity of all experiments.
> >
> > ---
> > We would like to express our sincere gratitude once again for the time and effort you have dedicated to this review. We hope that our responses have satisfactorily addressed your concerns. Should you have any further questions, please do not hesitate to discussions with us.
> >
> > Best,
> >
> > Authors
> >
> > [11] Chen, Richard J., et al. "Towards a general-purpose foundation model for computational pathology." Nature medicine 2024.
> >
> > [12] Chan T H,  et al. Histopathology whole slide image analysis with heterogeneous graph representation learning. CVPR 2023.
> >
> > [13] Li J, et al. Dynamic graph representation with knowledge-aware attention for histopathology whole slide image analysis. CVPR 2024.

---

> > > ### Author Response · Authors · 2025-11-27
> > > **Response to comparison with DisPro**
> > >
> > > ---
> > > We have completed the comparison experiments with DisPro [1] and added the results to Table RA and Table RB.
> > >
> > > DisPro distills prognostic knowledge into the genomics prompt during training. However, its inference procedure relies on self-scores computed within each individual case to select and aggregate tokens. This design overlooks holistic case-level representations and inter-case relationships, which results in a c-index that is 1.44\% lower than MSRL’s as shown in Table RA. In addition, DisPro performs multi-stage uni-modal training, which makes it sensitive to the data distributions of both modalities and weakens its generalization ability at inference time. As shown in Table RB, DisPro’s c-index drops by 4.77% when evaluated on out-of-domain data. In contrast, MSRL only a 1.02% performance decrease under the same setting, which exhibits strong generalization capacity.
> > >
> > > ##### Table RA: The C-Index (mean ± std) on five survival prediction tasks. (A part of the Table 1 in the paper)
> > > |Method|Modality|BLCA|BRCA|STAD|HNSC|COADREAD|overall|
> > > |---|---|---|---|---|---|---|---|
> > > |**DisPro_multi**|**g.+h.**|**0.6267±0.0423**|**0.6931±0.0372**|**0.7097±0.0403**| **0.6390±0.0580**| **0.6770±0.0479**|**0.6691**|
> > > |MSRL_multi|g.+h.|0.6368±0.0327| 0.7012±0.0302| 0.7236±0.0411|0.6456±0.0263|0.6896±0.0301|0.6794|
> > > |||||||||
> > > |**DisPro**|**g.+h.→h.**|**0.6058±0.0269**| **0.6734±0.0352**|**0.6803±0.0424**|**0.6053±0.0610**|**0.6418±0.0342**|**0.6414**|
> > > |MSRL|g.+h.→h.|0.6192±0.0184|0.6808±0.0277|0.7050±0.0523|0.6182±0.0015|0.6554±0.0166|0.6558|
> > >
> > >
> > > ##### Table RB: The generalization analysis results, where CPTAC-HNSC and CPTAC-BRCA are external test sets and "Δ" denotes performance declines. (A part of the Table A1 in the paper)
> > > |method|TCGA-HNSC C-index|CPTAC-HNSC C-index|Δ|TCGA-BRCA AUC|CPTAC-BRCA AUC|Δ|
> > > |---|---|---|---|---|---|---|
> > > |FEATHER|0.5277±0.0302|0.5197±0.0214|-0.80%|0.627±0.0085|0.5782±0.0142|-4.88%|
> > > |CHIEF|0.5338±0.0388|0.5246±0.0301|-0.92%|0.602±0.0169|0.5608±0.0198|-4.12%|
> > > |GigaPath|0.5580±0.0330|0.5278±0.0545|-3.02%|0.625±0.0074|0.5824±0.0457|-4.26%|
> > > |TITAN|0.5652±0.0381|0.5341±0.0610|-3.11%|0.648±0.0044|0.6094±0.0541|-3.86%|
> > > ||||||||
> > > |MSRL_H|0.5676±0.0289|0.5461±0.0125|-2.15%|0.652±0.0095|0.6099±0.0401|-4.21%|
> > > |G-HANet|0.5770±0.0278|0.5238±0.0786|-5.32%|0.632±0.0263|0.5612±0.0634|-7.08%|
> > > |LD-CVAE|0.5960±0.0286|0.5418±0.0469|-5.42%|0.646±0.0309|0.5728±0.0309|-7.32%|
> > > |**DisPro**|**0.6053±0.0610**|**0.5576±0.0692**|**-4.77%**|--|--|--|
> > > |MSRL|0.6182±0.0015|0.6080±0.0118|-1.02%|0.664±0.0263|0.6478±0.0416|-1.62%|
> > >
> > > *As DisPro’s prompts are specifically designed for survival prediction, we therefore excluded it from the cancer staging tasks.*
> > >
> > > The revised manuscript has been uploaded, with all modifications highlighted in blue. If you have any remaining concerns, we would be happy to discuss them further.
> > >
> > > Best,
> > >
> > > Authors
> > >
> > > [1] Xu, Yingxue, et al. "Distilled Prompt Learning for Incomplete Multimodal Survival Prediction." CVPR 2025.

---

### Official Review · Reviewer_iCa8 · 2025-11-05

**Soundness:** 2
**Presentation:** 2
**Contribution:** 2
**Rating:** 2
**Confidence:** 5

**Summary:**

This paper proposes a multi-modal structural representation learning (MSRL) framework to address the issue of missing modality (it is genomics in this paper). It consists of a pre-training stage and a fine-tuning stage. In the first stage, i.e., pre-training, paired histology-gene data is collected to train GNN encoders for different modalities (including gene, histology, gene-histology fused) via a self-supervised graph contrastive learning strategy, where one node represents the corresponding features of one patient. In the second stage, a dual-branch fine-tuning strategy is adopted. It uses an online branch that takes histology WSIs as input to generate gene representation and fuses multi-modal representations (original histology + generated gene features) for prediction. To ensure the quality of gene representation generation, fused-feature, structure, and graph-level alignment are imposed, with an EMA-based model updating technique. Experiments on five datasets are conducted and their results show that the proposed framework obtains the best overall performance in compared baselines.

**Strengths:**

- The motivation of MSRL is clearly presented. Moreover, the analysis of the limitations of existing methods is rational.
- The proposed framework considers the potential relevance between different cases, which remains understudied in existing works.
- The core idea of the proposed framework seems interesting as it employs the WSI as the condition to generate gene features with several alignment constraints to ensure the generation quality.

**Weaknesses:**

My major concerns as as follows:
- The presentation of the Methods part: This part introduces many notations, individual steps, loss functions, new concepts, and new terminologies, making it hard to follow. The authors are strongly encouraged to improve it by carefully organizing each step in the algorithm and making sure their representation is self-contained and can convey information more efficiently.
- Quite-complicated solution: The proposed framework, overall, is complicated, with many techniques (self-supervised graph contrastive learning, mixup-based augmentation, dual-branch updating, etc) stacked, making it look like an over-engineering scheme.
- Unclear technical contributions in the first stage: It is unclear which algorithms are proposed by this work.
- Insufficient experimental evidence: the authors mentioned that their framework is proposed for a data-efficiency purpose. However, the experiments are missing. One could say this framework is not data-efficient as it requires pre-training on large-scale paired data. Furthermore, the authors should design new experiments (different modality-missing rates) to verify the data-efficiency advantage of the proposed framework. Besides, a state-of-the-art baseline (Xu et al., Distilled Prompt Learning for Incomplete Multimodal Survival Prediction, CVPR 2025) is not compared. The authors could also add some general, representative methods (those also address modality missing issues) for comparisons to increase the soundness.

In summary, this paper still needs substantial improvements in terms of presentation and key experimental designs to support its claims.

**Questions:**

- What do the authors mean by 'data-efficient' in the title? If it refers to missing data, designing new experiments (different modality-missing rates) would be better?
- Line 87: Could the authors explain in which scenarios genomic data is needed for cancer diagnosis? In most cases, histopathology WSIs (as the gold standard) are sufficient, right?
- Could Figure 2(b) be further improved to show how the missing modality is addressed more clearly? The current version is not clear enough.

---

> ### Author Response · Authors · 2025-11-24
> **Response to Reviewer iCa8 (1/5)**
>
> Dear Reviewer,
>
> we sincerely thanks for your thorough review and appreciate your recognition of the motivation. Your comments are very helpful for improving our paper. We have added more experiments and provided clearer explanations to address your concerns. Our detailed responses are as follows:
>
>
> ---
> ## Response to the Presentation of Methods and the Motivation of Techniques (1/2)
>
> Since the WSI of the case is a giga-pixel ultra-high resolution image, and the gene data also has thousands of dimensions, constructing case-level representations is a complex process. Therefore, we introduce techniques of multiple hierarchy to solve this problem. It is not a direct stacking of techniques. Each part is based on the relevant motivation of this paper and has experimental support to justify the necessity of its modules.
>
> **First Stage: Pre-training**
> The core goal of the first pre-training stage is to train the GSL's multimodal case relevance representation capability within an aligned space for WSI and genomic features. A significant volume of TCGA case data lacks unified annotation information. Therefore, we adopted unsupervised contrastive learning. This approach fully leverages the model's ability to extract multimodal data representations and learn inter-case relationships. We have added an ablation study detailing the contributions of each loss component within the self-supervised graph contrastive learning framework used in pre-training. The specific results are shown in Table R1. The results confirm the effectiveness of this approach.
>
> ##### Table R1: ablation results of the loss function during the pre-training on validation datasets
> |Dataset|All losses|$\mathcal{L}\_{intra}$+$\mathcal{L}\_{fuse}$|$\mathcal{L}\_{inter}$+$\mathcal{L}\_{fuse}$|$\mathcal{L}\_{inter}$+$\mathcal{L}\_{intra}$|$\mathcal{L}\_{inter}$|$\mathcal{L}\_{intra}$|$\mathcal{L}\_{fuse}$|
> |---|---|---|---|---|---|---|---|
> |BLCA|0.607|0.582(↓0.025)|0.571(↓0.036)|0.567(↓0.040)|0.522(↓0.085)|0.514(↓0.093)|0.556(↓0.051)|
> |BRCA|0.672|0.640(↓0.032)|0.646(↓0.026)|0.636(↓0.036)|0.538(↓0.134)|0.524(↓0.148)|0.563(↓0.109)|
> |STAD|0.729|0.656(↓0.073)|0.670(↓0.059)|0.646(↓0.083)|0.560(↓0.169)|0.557(↓0.172)|0.585(↓0.144)|
> |HNSC|0.622|0.603(↓0.019)|0.594(↓0.028)|0.569(↓0.053)|0.530(↓0.092)|0.536(↓0.086)|0.559(↓0.063)|
> |COADREAD|0.661|0.621(↓0.040)|0.629(↓0.032)|0.613(↓0.048)|0.545(↓0.116)|0.543(↓0.118)|0.573(↓0.088)|
>
> *We will add these results to the appendix of original manuscript.*

---

> ### Author Response · Authors · 2025-11-24
> **Response to Reviewer iCa8 (2/5)**
>
> ## Response to the Presentation of Methods and the Motivation of Techniques (2/2)
>
> **Second Stage: Fine-tuning**
> The primary objective of the second stage, fine-tuning, is to utilize authentic, complete multimodal data. This guides the WSI unimodal data to perform efficient task inference. To achieve this, we incorporated a teacher-student dual-branch architecture, where mixup-based augmentation and EMA updating are corresponding technical paradigms. To further clarify our methodology, we have reorganized the fine-tuning procedure. The revised and clearer flow is detailed below.
>
> We define the $Branch(\mathbf{h}, \mathbf{g}, D\_{buffer})$ as follows:
> 1. **Input**
>    - $\mathbf{h}$: slide features;
>    - $\mathbf{g}$: gene features;
>    - $D\_{buffer}$: feature buffer of training cases;
> 2. **Processing**
>    - $\mathbf{f} = concatenate(\mathbf{g}, \mathbf{h})$;
>    - $\mathbf{F} = Readout(D\_{buffer}, \mathbf{f})$, denoting the combination of the current cases and other cases in the buffer;
>    - $\mathbf{A}^r = GSL(\mathbf{F})$;
>    - $\mathbf{Z} = GCN(\mathbf{A}^r, \mathbf{F})$;
> 3. **Output**
>    - $\mathbf{f}$: node features;
>    - $\mathbf{A}^r$: refined adjacency matrix;
>    - $\mathbf{Z}$: graph feature.
>
> Then, the fine-tune stage is as follows:
> - step 1: Initialize each module with the pre-trained weights;
>
> - step 2: $\mathbf{h} = \phi\_H( X\_I^{aug}), \hat{\mathbf{h}} = \phi\_H( \hat{X}\_I^{aug}), \mathbf{g} = \phi\_G( X\_G)$;
>
> - step 3: $\mathbf{f}$, $\mathbf{A}^r$, $\mathbf{Z}$ = **$Branch\_{online}(\mathbf{h}, Inductor(\mathbf{h}), D\_{buffer})$**;
>
> - step 4: $\hat{\mathbf{f}}$, $\hat{\mathbf{A}}^r$, $\hat{\mathbf{Z}}$ = **$Branch\_{target}(\hat{\mathbf{h}}, \mathbf{g}, D\_{buffer})$**;
>
> - step 5: $D\_{buffer} = update(D\_{buffer}, \hat{\mathbf{f}})$;
>
> - step 6: $\mathcal{L}\_{falign}(\mathbf{f}, \hat{\mathbf{f}})$, $\mathcal{L}\_{galign}(\mathbf{Z}, \hat{\mathbf{Z}})$, $\mathcal{L}\_{salign}(\mathbf{A}^r, \hat{\mathbf{A}}^r)$, $\mathcal{L}\_{task}(\mathbf{Z}, y)$, where $y$ denotes the task labels;
>
> *We will incorporate this workflow into subsections 3.3 of the methodology section in the original manuscript. Furthermore, we will reorganize and streamline the content to ensure greater conciseness and clarity.*
>
> The ablation results for each loss component in the fine-tuning stage are presented in Table R2. These results show that Hierarchical Alignment is essential for ensuring the online branch's high performance and robustness.
>
> ##### Table R2: ablation results of the loss function during the fine-tuning on validation datasets (Table 4 in the paper)
> | |Three losses|$\mathcal{L}\_{galign}$+$\mathcal{L}\_{salign}$|$\mathcal{L}\_{falign}$+$\mathcal{L}\_{salign}$|$\mathcal{L}\_{falign}$+$\mathcal{L}\_{galign}$|$\mathcal{L}\_{falign}$|$\mathcal{L}\_{galign}$|$\mathcal{L}\_{salign}$|
> |---|---|---|---|---|---|---|---|
> |BLCA(N=357)|0.607|0.588(↓0.019)|0.599(↓0.008)|0.584(↓0.023)|0.570(↓0.037)|0.564(↓0.043)|0.573(↓0.034)|
> |BRCA(N=680)|0.672|0.644(↓0.028)|0.661(↓0.011)|0.623(↓0.049)|0.583(↓0.089)|0.577(↓0.095)|0.592(↓0.080)|
> |STAD(N=318)|0.729|0.694(↓0.035)|0.703(↓0.026)|0.687(↓0.042)|0.657(↓0.072)|Non-convergence|0.663(↓0.066)|
> |HNSC(N=392)|0.622|0.609(↓0.013)|0.617(↓0.005)|0.598(↓0.024)|0.563(↓0.059)|0.558(↓0.064)|0.587(↓0.035)|
> |COADREAD(N=298)|0.661|0.647(↓0.014)|0.656(↓0.006)|0.635(↓0.026)|0.558(↓0.103)|Non-convergence|0.577(↓0.084)|

---

> ### Author Response · Authors · 2025-11-24
> **Response to Reviewer iCa8 (3/5)**
>
> ## Response to Technical Contributions in the First Stage
>
> A core technical contribution of this stage is the construction of **case-level relevance**. This differs from previous approaches that modeled graphs based on image patches within a single case. This macroscopic relevance promotes the encoder's ability to learn more comprehensive, robust, and generalized representations.
>
> We have conducted an ablation study on the pre-training stage, which was included in the Table 3 in the original manuscript. Furthermore, we have added external validation using two datasets from CPTAC. Models trained on the TCGA dataset were tested directly on these external datasets. The experimental results are displayed in Table R3.
>
> The performance of MSRL significantly decreases after removing the pre-training stage. Its generalization capability is also greatly diminished. This practically validates the necessity of our pre-training approach. Furthermore, the existing foundation models in pathology fully demonstrate the importance of pan-cancer pre-training.[1][2][3][4]
>
> From a technical perspective, the genomics encoder used for buffer initialization in the second stage is trained by the first stage of pre-training. Without it, the representations stored in the buffer lack discriminative power, which reduces the performance on downstream tasks.
>
> ##### Table R3: The ablation results of pre-training stage, where "Δ" denotes performance declines.
> |method|TCGA-HNSC C-index|CPTAC-HNSC C-index|Δ|TCGA-BRCA AUC|CPTAC-BRCA AUC|Δ|
> |---|---|---|---|---|---|---|
> |MSRL|0.6182±0.0015|0.6080±0.0118|-1.02%|0.664±0.0263|0.6478±0.0416|-1.62%|
> |MSRL w/o Pre|0.5988±0.0253|0.5364±0.0408|-6.24%|0.639±0.0335|0.5856±0.0381|-5.34%|
>
> *We will add these results to the appendix of original manuscript.*
>
> [1] Xu, Hanwen, et al. "A whole-slide foundation model for digital pathology from real-world data." Nature 2024.
>
> [2] Shao, Daniel, et al. "Do Multiple Instance Learning Models Transfer?." ICML 2025.
>
> [3] Wang, Xiyue, et al. "A pathology foundation model for cancer diagnosis and prognosis prediction." Nature 2024.
>
> [4] Ding, Tong, et al. "A multimodal whole-slide foundation model for pathology." Nature Medicine 2025.

---

> > ### Author Response · Authors · 2025-11-24
> > **Response to Reviewer iCa8 (4/5)**
> >
> > ## Response to the Data-Efficiency Purpose and Experimental Evidence
> >
> > Our core objective is to achieve **data-efficiency during inference**, not to focus specifically on training using incompletely paired multimodal data.
> >
> > **1. Practical Scenario**
> > In the clinical diagnosis environment, Whole Slide Images (WSI) have become routine data. This is due to their low acquisition cost and convenience, resulting in a large volume of available data. Conversely, genomic data remains scarce because of its high acquisition cost, difficulty in collection, and patient privacy concerns. The inverse scenario, where genomic data is available but WSI is missing, is rare in routine clinical practice.
> >
> > **2. Motivation**
> > Our aim is to fully utilize existing complete multimodal data (public datasets and historical data from medical centers) for model training. However, during inference, we rely solely on the low-cost WSI unimodal data. This maximizes the diagnostic value of WSI, allowing it to reach performance comparable to multimodal inference. This fulfills our purpose of achieving data-efficiency during inference.
> >
> > **3. Experimental Setup**
> > Following your suggestion, we have included experiments detailing different genomic data missing rates. We did not originally design the method to handle missing modalities. Therefore, for this experiment, we adjusted the training strategy for the Target branch. If a case lacks genomic data, the Target branch does not participate in training and only the WSI-input Online branch is trained. The specific experimental results are presented in Table R4. Even when using only partially complete data, MSRL still surpasses all unimodal training methods. Furthermore, MSRL remains superior to existing methods for unimodal inference even at a 30% missing rate. This shows that MSRL can fully leverage existing multimodal data to learn comprehensive and robust representations. This greatly enhances WSI’s diagnostic value and data efficiency during the inference phase.
> >
> > ##### Table R4: The results of different genomic data missing rates.
> > |Method|Modality|BLCA|BRCA|STAD|HNSC|COADREAD|overall|
> > |---|---|---|---|---|---|---|---|
> > |CHIEF|h.|0.5606±0.0890|0.5762±0.0768|0.5668±0.0601|0.5338±0.0838|0.5822±0.0672|0.5639|
> > |TITAN|h.|0.5756±0.0864|0.6182±0.0711|0.6306±0.0838|0.5652±0.0381|0.6138±0.0340|0.6007|
> > |MSRL_H|h.|0.5774±0.0221|0.6398±0.0251|0.6626±0.0427|0.5676±0.0289|0.6182±0.0174|0.6131|
> > |||||||||
> > |G-HANet|g.+h.→h.|0.5806±0.0149|0.6418±0.0138|0.6782±0.0489|0.5770±0.0278|0.6216±0.0184|0.6246|
> > |LD-CVAE|g.+h.→h.|0.5954±0.0104|0.6430±0.0146|0.6938±0.0495|0.5960±0.0286|0.6280±0.0211|0.6313|
> > |||||||||
> > |MSRL missing 60%|g.+h.→h.|0.6008±0.0125|0.6508±0.0577|0.6918±0.0699|0.5896±0.0570|0.6186±0.0401|0.6303|
> > |MSRL missing 30%|g.+h.→h.|0.6132±0.0668|0.6752±0.0809|0.7034±0.0830|0.6026±0.0505|0.6448±0.0587|0.6478|
> > |MSRL missing 0%|g.+h.→h.|0.6192±0.0184|0.6808±0.0277|0.7050±0.0523|0.6182±0.0015|0.6554±0.0166|0.6558|
> >
> > *We will add these results to the Table 1 of original manuscript.*
> >
> >
> > **4. Comparison with Existing Methods**
> > We first directly compared our results with those reported in the original DisPro[5] paper. As shown in Table R5, MSRL's performance on TCGA BRCA, which is the closest dataset used by both, is closer to that of SurvPath than DisPro. This, to some extent, demonstrates MSRL's superiority. We are currently reproducing the DisPro results under our experimental settings to ensure a more objective and fair comparison. However, DisPro requires training three models for a single task, with each model involving fine-tuning an LLM. This leads to a longer experimental cycle under the five-fold cross-validation setting. We guarantee that these related experiments will be completed and results released within the next week.
> > ##### Table R5: The comparison of DisPro and MSRL.
> > ||DisPro|MSRL|
> > |---|---|---|
> > |Dataset |TCGA-BLCA(n=372)|TCGA-BLCA(n=357)|
> > |patch encoder|Vit (pretrained in UNI)|Vit (pre-trained in GigaPath)|
> > |WSI aggregation|LLM+Top-K pooling|LongNet (pre-trained in GigaPath)|
> > |Gene setting| 330 pathways|50 pathways|
> > |c-Index of SurvPath|0.6572|0.6288|
> > |c-Index of proposed method|0.6315|0.6192|
> > |performance gap with SurvPath|-2.57%|-0.96%|
> >
> >
> > [5] Xu, Yingxue, et al. "Distilled Prompt Learning for Incomplete Multimodal Survival Prediction." CVPR 2025.

---

> > > ### Author Response · Authors · 2025-11-24
> > > **Response to Reviewer iCa8 (5/5)**
> > >
> > > ## Response to Other Questions
> > >
> > > **1. The Significance of Genomic Data in Precision Cancer Diagnosis**
> > >
> > > - WSI contains morphological characteristics of tissue. This information is often sufficient for tasks like cancer subtyping. However, more precise diagnostic tasks—such as cancer grading (which relies on comprehensive clinical patient information and individual characteristics) or biomarker mutation status (which requires molecular testing and specific staining)—involve vast patient individuality. Accurate diagnosis is difficult if relying solely on WSI.
> > > - Numerous existing studies have validated that the introduction of genomic data significantly promotes the accuracy of precision cancer diagnosis, including survival prediction, cancer grading, and biomarker prediction.[6][7][8]
> > > - Our objective is to achieve highly efficient precision cancer diagnosis during the inference stage using only low-cost WSI. This fully enhances the data efficiency of WSI.
> > >
> > > **2. Regarding Figure 2(b)**
> > > In Figure 2(b), the **Inductor** module within the Online branch takes the WSI representation as input. Its output, the **genomics prompt**, serves as a placeholder for the missing genomic representation. The subsequent learning process uses inter-case relevance guidance and task supervision to learn this missing genomic representation. We will further clarify this mechanism within the figure.
> > >
> > > ---
> > > We would like to express our sincere gratitude once again for the time and effort you have dedicated to this review. We hope that our responses have satisfactorily addressed your concerns. Should you have any further questions, please do not hesitate to discussions with us.
> > >
> > > Best,
> > >
> > > Authors
> > >
> > >
> > > [6] Vaidya, Anurag, et al. "Molecular-driven foundation model for oncologic pathology." arXiv preprint (2025).
> > >
> > > [7] Jin, Cheng, et al. "Genome-Anchored Foundation Model Embeddings Improve Molecular Prediction from Histology Images." arXiv preprint (2025).
> > >
> > > [8] Xu, Yingxue, et al. "A multimodal knowledge-enhanced whole-slide pathology foundation model." arXiv preprint (2024).

---

> ### Author Response · Authors · 2025-11-27
> **Response to comparison with DisPro**
>
> We have completed the comparison experiments with DisPro [1] and added the results to Table RA and Table RB.
>
> DisPro distills prognostic knowledge into the genomics prompt during training. However, its inference procedure relies on self-scores computed within each individual case to select and aggregate tokens. This design overlooks holistic case-level representations and inter-case relationships, which results in a c-index that is 1.44\% lower than MSRL’s as shown in Table RA. In addition, DisPro performs multi-stage uni-modal training, which makes it sensitive to the data distributions of both modalities and weakens its generalization ability at inference time. As shown in Table RB, DisPro’s c-index drops by 4.77% when evaluated on out-of-domain data. In contrast, MSRL only a 1.02% performance decrease under the same setting, which exhibits strong generalization capacity.
>
> ##### Table RA: The C-Index (mean ± std) on five survival prediction tasks. (A part of the Table 1 in the paper)
> |Method|Modality|BLCA|BRCA|STAD|HNSC|COADREAD|overall|
> |---|---|---|---|---|---|---|---|
> |**DisPro_multi**|**g.+h.**|**0.6267±0.0423**|**0.6931±0.0372**|**0.7097±0.0403**| **0.6390±0.0580**| **0.6770±0.0479**|**0.6691**|
> |MSRL_multi|g.+h.|0.6368±0.0327| 0.7012±0.0302| 0.7236±0.0411|0.6456±0.0263|0.6896±0.0301|0.6794|
> |||||||||
> |**DisPro**|**g.+h.→h.**|**0.6058±0.0269**| **0.6734±0.0352**|**0.6803±0.0424**|**0.6053±0.0610**|**0.6418±0.0342**|**0.6414**|
> |MSRL|g.+h.→h.|0.6192±0.0184|0.6808±0.0277|0.7050±0.0523|0.6182±0.0015|0.6554±0.0166|0.6558|
>
>
> ##### Table RB: The generalization analysis results, where CPTAC-HNSC and CPTAC-BRCA are external test sets and "Δ" denotes performance declines. (A part of the Table A1 in the paper)
> |method|TCGA-HNSC C-index|CPTAC-HNSC C-index|Δ|TCGA-BRCA AUC|CPTAC-BRCA AUC|Δ|
> |---|---|---|---|---|---|---|
> |FEATHER|0.5277±0.0302|0.5197±0.0214|-0.80%|0.627±0.0085|0.5782±0.0142|-4.88%|
> |CHIEF|0.5338±0.0388|0.5246±0.0301|-0.92%|0.602±0.0169|0.5608±0.0198|-4.12%|
> |GigaPath|0.5580±0.0330|0.5278±0.0545|-3.02%|0.625±0.0074|0.5824±0.0457|-4.26%|
> |TITAN|0.5652±0.0381|0.5341±0.0610|-3.11%|0.648±0.0044|0.6094±0.0541|-3.86%|
> ||||||||
> |MSRL_H|0.5676±0.0289|0.5461±0.0125|-2.15%|0.652±0.0095|0.6099±0.0401|-4.21%|
> |G-HANet|0.5770±0.0278|0.5238±0.0786|-5.32%|0.632±0.0263|0.5612±0.0634|-7.08%|
> |LD-CVAE|0.5960±0.0286|0.5418±0.0469|-5.42%|0.646±0.0309|0.5728±0.0309|-7.32%|
> |**DisPro**|**0.6053±0.0610**|**0.5576±0.0692**|**-4.77%**|--|--|--|
> |MSRL|0.6182±0.0015|0.6080±0.0118|-1.02%|0.664±0.0263|0.6478±0.0416|-1.62%|
>
> *As DisPro’s prompts are specifically designed for survival prediction, we therefore excluded it from the cancer staging tasks.*
>
> The revised manuscript has been uploaded, with all modifications highlighted in blue. If you have any remaining concerns, we would be happy to discuss them further.
>
> Best,
>
> Authors
>
> [1] Xu, Yingxue, et al. "Distilled Prompt Learning for Incomplete Multimodal Survival Prediction." CVPR 2025.

---

### Official Review · Reviewer_xzsK · 2025-11-07

**Soundness:** 2
**Presentation:** 2
**Contribution:** 3
**Rating:** 4
**Confidence:** 4

**Summary:**

This paper proposes a graph structure learning-based approach to fuse genomics with histology for survival prediction. The approach aims to capture the relationship between cases by constructing a graph per modality and between modalities. It consists of a pre-training phase and fine-tuning phase. The pre-training phase learns graph structures for each modality, with contrastive loss preserving intra-modal representations that are invariant to slight pertubations to the graph structure, as well as inter-modal representations that seek to align nodal representations across modalities. Meanwhile, the fine-tuning phase trains two branches: an online branch which processes only the histologic image and generates the genomic representation with an SNN, and a target branch which processes both the WSI and true genomic representation. A hierarchical alignment module encourages the histology-only online branch to have the same output as the histology+genomic target branch. During inference, only the online branch (histology only) is used.

The method, called MSRL, is pre-trained on histology+genomic data from TCGA, and evaluated on a subset of these cases for survival prediction. Comparison against 13 total benchmark approaches (genomics-only, histology-only, histology + genomic,, and inferred genomic) demonstrate that MSRL consistently outperforms existing approaches across five survival tasks and four diagnostic tasks.

Ablation studies investigate relevant questions, including how MSRL performance changes when exposed to histology only versus both modalities at inference. The model is also benchmarked against a sufficient number of tasks and methods. However, pre-training and evaluation are performed entirely on cases from TCGA. Their method may be particularly susceptible to distribution shift due to its reliance on generated representations and encoding of all patient samples in their generated graph, and should therefore be evaluated on external cohorts such as CPTAC as well.

Additionally, MSRL utilizes extensive pre-training for the slide encoder, but performs benchmarks entirely against randomly initialized models. Additional comparisons with open-weight pre-trained slide encoders should be performed.

**Strengths:**

- The method is benchmarked against an appropriate number of tasks and organs
- The method is benhmarked against an appropriate number of approaches across a range of histology-only, histology + genomics, and inferred genomic approaches, demonstrating meaningful improvements compared to these benchmarks.
-  The proposed method is novel and addresses a new method to incorporate genomic information through a graph-based approach.
- The ablations are well-selected and provide helpful insight into the effect of each component on performance.

**Weaknesses:**

- Pre-training and validation is performed entirely on cases from TCGA. Performance on out of distribution cases, from cohorts external to TCGA, should be investigated. It is possible that MSRL, which both retains all sample-level embeddings from pretraining in the form of nodes, and generates simulated genomic information, will struggle to generalize to new clinical distributions.
- While MSRL utilizes pre-training, it is benchmarked against randomly initialized approaches. MSRL should also be benchmarked against existing lightweight slide encoders such as THREADS, CHIEF, and FEATHER.
- The methods consist of many steps and moving parts. The section is difficult to follow, with many elements introduced without sufficient motivation or context, described below:
- Line 224-226: Please describe the post-processing steps in the appendix
- Line 231-232: Clarify the meaning of this sentence. The motivation behind using contrastive learning with intra-modal InfoNCE is not clear. In particular, please clarify that the goal is invariance between each respective node, and why this invariance is desirable.
- Line 280: Please define the purpose of the “feature buffer” and what it represents.
- Line 276-277: Please clearly describe the purpose of the online branch and target branch.
- Line 289-290: Many new terms are introduced here. Please define more clearly either the Inductor module or the meaning of a “genomics prompt”.
- Line 295: Please define the meaning of the $Readout$ operation
- Line 305: make more clear what the hierarchical nature of this loss is.
- Line 312: clarify how alignment constraint helps learn “robust representations”. What about this loss makes the online branch more “robust”?
- In addition to describing the design motivations more clearly in the methods, the motivation for using GSL is not clearly stated in the introduction.
- There is no mention of GSL or graphs in related works

**Questions:**

While no cases from evaluation were used during pre-training, were the cases also stratified by patient? If a patient has multiple cases divided across pretraining and validation cohorts, then their genomic information may still be encoded in the pretraining stage. The splits used for pretraining and evaluation should be shared.

---

> ### Author Response · Authors · 2025-11-24
> **Response to  Reviewer xzsK (1/4)**
>
> Dear Reviewer,
>
> we sincerely thanks for your thorough review and appreciate your recognition of the contribution and performance of our method. Your comments are very helpful for improving our paper. We have added more experiments and provided clearer explanations to address your concerns. Our detailed responses are as follows:
>
>
> ---
> ## Response to pre-trained slide encoders
> Following a comprehensive review, we incorporated **CHIEF** [1] and **FEATHER** [2] as the recommended pre-trained lightweight slide encoders. Additionally, we included comparative experiments with **TITAN**[3], a recently released image-text pre-trained foundation model. However, comparisons with **THREADS** [4] were not feasible due to the unavailability of its source code and model weights. The specific experimental results are as shown in Tables R1 and R2:
>
>
> In survival prediction tasks, TITAN achieved optimal comprehensive performance among methods relying solely on WSI unimodal training. This suggests that integrating regional-level text pre-training effectively enhances the representation capability of WSI encoders, enabling the capture of more comprehensive information in complex downstream tasks. Conversely, FEATHER utilizes supervised pre-training based on extensive slide-level annotated slides for classification. Consequently, it demonstrated superior performance across four precision diagnosis tasks but proved less effective for survival prediction, likely due to the substantial divergence in task types.
>
> Overall, our proposed MSRL_H, incorporating WSI-Genomics structural information during pre-training, consistently outperforms existing pre-trained slide encoders. This validates that structural relevance between multimodal data can effectively enhance the representation capability of slide encoder.
>
> ##### Table R1: The results on test set of the survival prediction tasks.
>
> |Methods|Modality|BLCA|BRCA|STAD|HNSC|COADREAD|overall|
> |---|---|---|---|---|---|---|--|
> |FEATHER|h.|0.5306±0.0340|0.5698±0.0247|0.5570±0.0420|0.5277±0.0302|0.5796±0.0284|0.5530|
> |CHIEF|h.|0.5606±0.0890|0.5762±0.0768|0.5668±0.0601|0.5338±0.0838|0.5822±0.0672|0.5639|
> |TITAN|h.|0.5756±0.0864|0.6182±0.0711|0.6306±0.0838|0.5652±0.0381|0.6138±0.0340|0.6007|
> |MSRL_H|h.|0.5774±0.0221|0.6398±0.0251|0.6626±0.0427|0.5676±0.0289|0.6182±0.0174|0.6131|
> |||||||||
> |MSRL|g.+h.→h.|0.6192±0.0184|0.6808±0.0277|0.7050±0.0523|0.6182±0.0015|0.6554±0.0166|0.6558|
>
> ##### Table R2: The results on test set of the precision diagnosis tasks.
> |Model|BRCA staging ||NSCLC staging ||EGFR mutation ||HER2 status ||
> |---|---|---|---|---|---|---|---|---|
> ||AUC|F1| AUC| F1| AUC|F1|AUC|F1|
> |CHIEF|0.602±0.0169|0.566±0.0300|0.635±0.0270|0.589±0.0448|0.804±0.0340|0.713±0.0340|0.657±0.0110|0.517±0.0637|
> |FEATHER|0.627±0.0085|0.571±0.0056|0.646±0.0159|0.593±0.0150|0.816±0.0061|0.748±0.0240|0.689±0.0259|0.538±0.0620|
> |TITAN|0.648±0.0044|0.583±0.0438|0.639±0.0326|0.614±0.0166|0.822±0.0287|0.751±0.0150|0.693±0.0067|0.546±0.0157|
> || || || ||||
> |MSRL_H| 0.652±0.0095|0.586±0.0271| 0.655±0.0133| 0.625±0.0186| 0.826±0.0200| 0.758±0.0173|0.704±0.0569|0.550±0.0372|
> |MSRL|0.664±0.0263|0.593±0.0277|0.661±0.0102 |0.638±0.0108| 0.842±0.0206 |0.770±0.0165 |0.730±0.0223 |0.606±0.0274|
>
>
> *We will add these results to Table 1 and Table 2 of the original manuscript.*
>
> [1] Wang, Xiyue, et al. "A pathology foundation model for cancer diagnosis and prognosis prediction." Nature 2024.
>
> [2] Shao, Daniel, et al. "Do Multiple Instance Learning Models Transfer?." ICML 2025.
>
> [3] Ding, Tong, et al. "A multimodal whole-slide foundation model for pathology." Nature Medicine 2025.
>
> [4] Vaidya, Anurag, et al. "Molecular-driven foundation model for oncologic pathology." arXiv preprint arXiv:2501.16652 (2025).

---

> ### Author Response · Authors · 2025-11-24
> **Response to  Reviewer xzsK (2/4)**
>
> ## Response to Generalization of MSRL
>
> **1. Mechanism of Generalization**
>
> - **Our framework presents no additional difficulty in generalization.**
> Genomic data from out-of-distribution cases often suffers from complex sources and various protocols. Direct inclusion of such data can cause significant batch effects. In contrast, our method utilizes solely WSI unimodal inference. Its generalization only rely on the representation capability of the slide encoder. Therefore, the generalization capability of our method when facing out-of-distribution data is consistent with other WSI encoder-based inference frameworks and does not introduce additional challenges.
>
> - **Our cross-modal retrieval strategy even boosts the generalization ability.**
> Existing unimodal inference methods typically focus on reconstructing raw genomic data for individual cases, where the recontruction performance is also affected by OOD issues. In contrast, MSRL models genomic correlations between cases within an aligned representation space and then use the real source-domain genomic representation for muilti-modal prediciton. This strategy naturally avoids the multi-source heterogeneity inherent in raw genomic data during training. Therefore, it ensures better generalization than reconstruction-based methods during the inference phase.
>
> **2. External Evaluation Experiments**
> We constructed new external datasets to fully validate the generalizability of MSRL. The details are as follows:
>
> - **External Datasets:** We collected two public datasets from the CPTAC project: **CPTAC-HNSC** (comprising WSI data and clinical survival information for 106 patients) and **CPTAC-BRCA** (containing WSI data and diagnostic stage information for 111 patients). We ensured these datasets share no overlap with the TCGA project.
>
> - **Experimental Setting:** To validate the generalizability of MSRL and baseline methods on out-of-distribution cases, we directly applied models trained on TCGA-HNSC and TCGA-BRCA to the CPTAC-HNSC and CPTAC-BRCA datasets for testing, respectively. Additionally, we incorporated several foundation models known for high generalizability for comparison. The specific experimental results are shown in Table R3.
>
> - **Results Analysis:**
>     *  MSRL consistently achieved the best performance on both external datasets. Notably, MSRL_H secured the top results even under conditions limited to WSI unimodal training.
>     *  In scenarios addressing missing inference modalities, existing methods like G-HANet and LD-CAVE suffered performance drops exceeding 5% and 7% on CPTAC-HNSC and CPTAC-BRCA, respectively. In contrast, MSRL showed minimal declines of only 1.02% and 1.62%, outperforming most foundation models. These results demonstrate the superior generalizability of MSRL.
>     * Compared to in-domain performance on TCGA testing sets, pre-trained foundation models exhibited relatively minor performance declines on external data, with C-index reductions ranging from 0.8% to 3.11% on CPTAC-HNSC and AUC reductions between 3.86% and 4.88% on CPTAC-BRCA.
>
> ##### Table R3: The results on external test sets, where "Δ" denotes performance declines.
> |method|TCGA-HNSC C-index|CPTAC-HNSC C-index|Δ|TCGA-BRCA AUC|CPTAC-BRCA AUC|Δ|
> |---|---|---|---|---|---|---|
> |FEATHER|0.5277±0.0302|0.5197±0.0214|-0.80%|0.627±0.0085|0.5782±0.0142|-4.88%|
> |CHIEF|0.5338±0.0388|0.5246±0.0301|-0.92%|0.602±0.0169|0.5608±0.0198|-4.12%|
> |GigaPath|0.5580±0.0330|0.5278±0.0545|-3.02%|0.625±0.0074|0.5824±0.0457|-4.26%|
> |TITAN|0.5652±0.0381|0.5341±0.0610|-3.11%|0.648±0.0044|0.6094±0.0541|-3.86%|
> ||||||||
> |MSRL_H|0.5676±0.0289|0.5461±0.0125|-2.15%|0.652±0.0095|0.6099±0.0401|-4.21%|
> |G-HANet|0.5770±0.0278|0.5238±0.0786|-5.32%|0.632±0.0263|0.5612±0.0634|-7.08%|
> |LD-CVAE|0.5960±0.0286|0.5418±0.0469|-5.42%|0.646±0.0309|0.5728±0.0309|-7.32%|
> |MSRL|0.6182±0.0015|0.6080±0.0118|-1.02%|0.664±0.0263|0.6478±0.0416|-1.62%|
>
>
> *We will add these results to the appendix of original manuscript.*

---

> > ### Author Response · Authors · 2025-11-24
> > **Response to Reviewer xzsK (3/4)**
> >
> > ## Response to Presentation and Clarity (1/2)
> > **Lines 224–226:** The specific post-processing steps as follows:
> > - **Sparsification.** The obtained similarity matrix $S$ is typically dense and requires sparsification. We employ a KNN-based method: for each node, we retain the top $K$ connected edges and set the remaining connections to zero, specifically as follows:
> > $\mathbf{S}\_{ij}^{(sp)} = q\_{sp}(\mathbf{S}\_{ij})= \mathbf{S}\_{ij}  \ if \ {\mathbf{S}}\_{ij} \in \text{top-k}({\mathbf{S}}\_i) \ else \ 0 $
> >
> > - **Symmetrization and Activation.** To ensure bidirectional connections between nodes and guarantee non-negative edge values, we perform the following additional processing:
> > ${\mathbf{S}}^{(sym)} = q\_{sym}\left(q\_{act}\left({\mathbf{S}}^{(sp)}\right)\right) = \frac{\sigma_q\left({\mathbf{S}}^{(sp)}\right) + \sigma_q\left({\mathbf{S}}^{(sp)}\right)^{\mathrm{T}}}{2}$, where $\sigma(\cdot)$ is the activation function.
> >
> > - **Normalization.** To ensure the edge weights fall within the range $[0, 1]$, the final processing step is as follows:
> > $S\_{final} = q\_{norm} \left( {\mathbf{S}}^{(sym)} \right) = \left( {\mathbf{D}}^{(sym)} \right)^{-\frac{1}{2}} {\mathbf{S}}^{(sym)} \left( {\mathbf{D}}^{(sym)} \right)^{-\frac{1}{2}}$, where $D^{(sym)}$ is the degree matrix of $S^{(sym)}$.
> >
> > **Lines 231–232: Intra-modal Constraint (InfoNCE)**
> > In the original and augmented graph of the same modality, corresponding nodes still represent the same case and form a positive pair. To ensure that the node representations of the same case (the positive samples) are drawn closer, while those of different cases (the negative samples) are pushed apart, we utilize **InfoNCE** as the intra-modal constraint.
> >
> > **Lines 280 & 276–277: Feature Buffer and Online/Target Framework**
> > A key contribution of this paper is the use of authentic genomics data to guide inference under missing modality conditions, with the **feature buffer** and **Target-Online** being essential to this. Training samples for the downstream task possess complete WSI-Genomics data. The feature buffer stores the WSI-Genomics multimodal representations of these samples in the format of `{case_id: feature}`.
> >
> > During training, the **Online branch** is solely input with WSI, while the **Target branch** receives both WSI and paired Genomics data. The multimodal representations from both branches are then used for GSL graph construction with the authentic case representations stored in the buffer. It is vital that the real-data-fed Target branch constrains the Online branch during this process, optimizing the Online branch to learn inter-case multimodal structural representations. The Target branch output also updates the buffer, ensuring the buffer consistently retains authentic data representations. For inference, only the Online branch is used, performing task prediction based on its multimodal structural learning with the buffer.
> >
> >
> > **Lines 289–290: Inductor Module**
> > The "**Inductor module**" in the Online branch shares the similar structure as the genomic encoder in the Target branch and occupies a similar position. The key difference is that its input is solely the WSI representation. Its output, termed the "**genomics prompt**," serves as a placeholder for the missing genomics representation, thereby ensuring the completeness of the subsequent GSL pipeline.
> >
> >
> > **Line 295: Readout**
> > The **Readout** module, based on the `case_id` of the current batch samples, extracts the representations of all cases not included in the current batch from the buffer. These are then combined with the features $\mathbf{f}$ of the current batch, serving as the input for the next GSL step.

---

> > > ### Author Response · Authors · 2025-11-24
> > > **Response to Reviewer xzsK (4/4)**
> > >
> > > ## Response to Presentation and Clarity (2/2)
> > >
> > > **Lines 305 & 312: Hierarchical Alignment Loss**
> > > Our alignment loss imposes **constraints at three distinct levels**: the node representations prior to graph input, the adjacency matrix during graph construction, and the graph representations after node updates. This multi-level approach ensures that the Online branch, constrained by the real-data-fed Target branch, simultaneously learns both node representations and inter-case correlation structures. The ablation study for this loss component in the original paper is as shown in Table R4 (Table 4 in the original paper):
> > > ##### Table R4: ablation results of the loss function during the fine-tuning on validation datasets (Table 4 in the original paper)
> > > | |Three losses|$\mathcal{L}\_{galign}$+$\mathcal{L}\_{salign}$|$\mathcal{L}\_{falign}$+$\mathcal{L}\_{salign}$|$\mathcal{L}\_{falign}$+$\mathcal{L}\_{galign}$|$\mathcal{L}\_{falign}$|$\mathcal{L}\_{galign}$|$\mathcal{L}\_{salign}$|
> > > |---|---|---|---|---|---|---|---|
> > > |BLCA(N=357)|0.607|0.588(↓0.019)|0.599(↓0.008)|0.584(↓0.023)|0.570(↓0.037)|0.564(↓0.043)|0.573(↓0.034)|
> > > |BRCA(N=680)|0.672|0.644(↓0.028)|0.661(↓0.011)|0.623(↓0.049)|0.583(↓0.089)|0.577(↓0.095)|0.592(↓0.080)|
> > > |STAD(N=318)|0.729|0.694(↓0.035)|0.703(↓0.026)|0.687(↓0.042)|0.657(↓0.072)|Non-convergence|0.663(↓0.066)|
> > > |HNSC(N=392)|0.622|0.609(↓0.013)|0.617(↓0.005)|0.598(↓0.024)|0.563(↓0.059)|0.558(↓0.064)|0.587(↓0.035)|
> > > |COADREAD(N=298)|0.661|0.647(↓0.014)|0.656(↓0.006)|0.635(↓0.026)|0.558(↓0.103)|Non-convergence|0.577(↓0.084)|
> > >
> > > The results demonstrate that all these constraints are beneficial for the Online branch's representation learning and task inference.
> > >
> > > **About GSL (Graph Structure Learning)**
> > > A core motivation of this work is to construct inter-case relevance. Graph representation is ideal for fully describing the many-to-many associations between cases and is compatible with various modal representations as node features. However, static graph construction methods (such as KNN) rely on hard-to-define prior structures and are susceptible to noise. For instance, relationships between cancer patients are not always obvious. We therefore adopted the data-driven **Graph Structure Learning (GSL)** approach, necessitating a dynamic construction method. The ablation study in the original paper, which validated the necessity and effectiveness of GSL, is shown as in Table R5 (Table 3 in the original paper):
> > > ##### Table R5:The results of the structure ablation on five survival prediction datasets (Table 3 in the original paper)
> > > Model|BLCA|BRCA|STAD|HNSC|COADREAD|Overall
> > > ---|---|---|---|---|---|---
> > > KNN(euclidean)|0.569±0.029|0.628±0.021|0.632±0.036|0.562±0.030|0.611±0.011|0.600
> > > KNN(cosin)|0.577±0.023|0.637±0.025|0.641±0.041|0.568±0.031|0.618±0.014|0.608
> > > MSRL_random_GSL|0.598±0.021|0.669±0.033|0.681±0.032|0.599±0.025|0.637±0.015|0.637
> > > MSRL|0.619±0.018|0.681±0.028|0.705±0.052|0.618±0.002|0.655±0.017|0.656
> > >
> > > Furthermore, due to space limitations, the related work on GSL is relegated to the Appendix (see page 20).
> > >
> > > ---
> > >
> > > We will ultimately release all data splits, code, and model weights.
> > >
> > >
> > > Once again, thank you for your constructive feedback. We hope our response is helpful to better understand this work, and we look forward to further discussions with you.
> > >
> > > Best,
> > >
> > > Authors

---

> ### Comment · Reviewer_xzsK · 2025-11-24
>
> Thanks to the authors for their efforts to address my comments. I am still going through the results but wanted to comment early to provide authors time to respond.
>
> Could the authors please clarify how the slide encoders were evaluated? Was this linear probe or finetuning? What was the training recipe used?
>
> Additionally, I would like to clarify that my comments regarding the methodology being difficult to follow were intended to help the authors revise the text for the manuscript itself. While the authors have provided valuable additional experimentation, the presentation of the methods is also a substantial weakness. Similarly, the related works on graph-based and graph-structure-based learning should appear in the main text. An additional page is permitted at this revision stage to accommodate these additions. I strongly recommend the authors provide an updated manuscript based on reviewer feedback, with colored text to indicate changes that were made.

---

> > ### Author Response · Authors · 2025-11-25
> >
> > We sincerely appreciate your prompt response and kind reminder.
> >
> > For all foundation models, we adopted the strategy of fine-tuning the slide encoder. Specifically, we first extracted patch features using their respective patch encoders. Subsequently, we initialized the slide encoders using their published pre-trained weights (ABMIL for FEATHER and CHIEF, and ViT for TITAN). Finally, we fine-tuned these slide encoders on the training datasets for each downstream task and reported the final metrics on the test sets.
> >
> > We would like to express our gratitude once again for your constructive suggestions regarding the methodology. We are currently revising the text and will upload the updated manuscript as soon as possible. We will notify you immediately upon completion.

---

> > ### Author Response · Authors · 2025-11-27
> > **Response to the revised manuscript**
> >
> > Dear Reviewer,
> >
> > The revised manuscript has been uploaded, with all modifications highlighted in blue. If you have any remaining concerns, we would be happy to discuss them further.
> >
> > Best,
> >
> > Authors

---

### Author Response · Authors · 2025-11-27
**General response**

Dear AC and Reviewers,

We sincerely appreciate the time and effort you have devoted to reviewing our manuscript. Your constructive comments on writing clarity, literature coverage, methodological motivation, experimental design, and generalization analysis were extremely valuable. **We have thoroughly revised the paper and uploaded the updated manuscript, where the modifications are highlighted in blue.** The major changes are summarized as follows:

1. **Presentation and Clarity.**
We reorganized the Methods section to improve readability. This includes simplifying notations, clarifying the roles of variables and functions, consolidating shared components of the dual-branch architecture, presenting the workflow in a more modular manner, and clarifying the framework illustration in Figure 2. We also expanded the explanations of design motivations. These revisions appear in *Section 3 (starting at line 153)*.

2. **Generalization Analysis.**
To better assess the generalization of MSRL, we collected additional out-of-domain CPTAC data and conducted an external validation study. The experimental results show that MSRL exhibits a performance drop of only 1.02% in out-of-domain validation, which is substantially lower than the over 4% reduction observed in other data-efficient methods. This demonstrates the strong generalization capability of MSRL. The experimental setup is provided in *Appendix C (lines 885–886)*, and the results and analysis are reported in *Appendix E.1 (starting at line 985)*.

3. **Comparison Baselines.**
We introduced additional pre-trained slide encoders (FEATHER [1], CHIEF [2], and TITAN [3]), more multimodal fusion methods (MOTCat[4] and PIBD [5]), and distillation-based missing-modality inference approache (DisPro [6]). The experimental results indicate that MSRL, enhanced with inter-case relevance, outperforms existing methods for missing-modality inference and achieves performance comparable to multimodal fusion approaches. The comparative results and analysis are presented in *Section 4.2, along with Table 1 and Table 2 (starting at line 405)*.

4. **Missing-Modality Training.**
We expanded the experiments to evaluate different missing-rate settings for gene data. The experimental results show that even when trained with 30% of the gene data missing, MSRL still achieves better inference performance than existing methods trained on complete datasets. The details are available in *Appendix E.4 (starting at line 1115)*.

5. **Pre-training Ablation.**
We added ablation experiments on alternative pre-training losses. The results demonstrate that each loss component associated with GSL effectively enhances the representation capability of MSRL. These results are shown in *Appendix E.3 (starting at line 1077)*.

6. **Buffer Analysis.**
We analyzed the effect of buffer size, where the results indicate that the buffer is the key component for incorporating authentic data and ensures effective and efficient single-modality inference. These results are provided in *Appendix E.2 (starting at line 1041)*.

7. **Expanded Literature Review.**
We incorporated more related work on multimodal fusion [4][5] and distillation-based approaches [6][7], updated the motivation illustration in Figure 1, and added references on graph structure learning. These additions are included in *Section 2 (starting at line 104)*.


We hope the revised manuscript addresses all concerns and presents our work more clearly. We welcome any further questions and discussions from the reviewers.

Best regards,

Authors

[1] Shao, Daniel, et al. "Do Multiple Instance Learning Models Transfer?." ICML 2025.

[2] Wang, Xiyue, et al. "A pathology foundation model for cancer diagnosis and prognosis prediction." Nature 2024.

[3] Ding, Tong, et al. "A multimodal whole-slide foundation model for pathology." Nature Medicine 2025.

[4] Xu, Yingxue, and Hao Chen. "Multimodal optimal transport-based co-attention transformer with global structure consistency for survival prediction." CVPR 2023

[5] Zhang, Yilan, et al. "Prototypical information bottlenecking and disentangling for multimodal cancer survival prediction." ICLR 2024.

[6] Xu, Yingxue, et al. "Distilled Prompt Learning for Incomplete Multimodal Survival Prediction." CVPR 2025.

[7] Jin, Cheng, et al. "Genome-Anchored Foundation Model Embeddings Improve Molecular Prediction from Histology Images." arXiv preprint arXiv:2506.19681 (2025).

---

### Author Response · Authors · 2025-12-01
**Summary of responses**

**Dear PCs, SACs, ACs, and Reviewers,**

Thank you very much for your time and efforts in reviewing our manuscript. To assist the AC and help streamline the decision process, we summarize below the key points from the reviews and our corresponding revisions.

---

### **Strengths**

Overall, we appreciate the positive feedback provided by the reviewers. Specifically:

- **Novelty of the proposed method.**
  Recognized by reviewers ***xzsK***, ***EbNx***, ***vXzx***, ***vuS4***, and ***GyVr***.

- **Clear motivation and problem formulation.**
  Acknowledged by reviewer ***iCa8***, ***vXzx***, and ***sJW6***

- **Soundness of experimental validation.**
  Highlighted by reviewers ***xzsK***, ***EbNx***,***vXzx***, ***vuS4***, and ***sJW6***.

---

### **Concerns and Our Addressing**

The reviewers’ concerns primarily focus on three areas: the clarity of the method description, the insufficient analysis of model generalization, and the requirements for comparison baselines. We have thoroughly revised the manuscript according to these comments and updated the relevant sections accordingly. Our detailed responses are provided below：


### **1. Presentation and Clarity**

**Concern:**
Reviewers requested a clearer and more description of the proposed method. (xzsK: Weaknesses 3-11; iCa8: Weaknesses 1-2; vuS4: Weaknesses 4; GyVr: Weaknesses 1,3)

**Our Addressing:**
We reorganized the Methods section to improve readability. This includes simplifying notations, clarifying variable roles, reorganizing shared components of the dual-branch architecture, presenting a more modular workflow, and refining Figure 2. We also expanded the explanation of module-wise motivations.
Revisions appear in *Section 3 (starting at line 153)*.



### **2. Generalization Analysis**

**Concern:**
Reviewers sought stronger evidence of MSRL’s ability to generalize across datasets. (xzsK: Weaknesses 1; EbNx: Weaknesses 2,3; vuS4: Weaknesses 2; GyVr: Weaknesses 4; sJW6: Weaknesses 2)

**Our Addressing:**
We collected additional out-of-domain CPTAC slides and conducted external validation. MSRL shows only a 1.02% performance drop, compared with over 4% for other data-efficient approaches. This indicates strong generalization capability.
The setup is provided in *Appendix C (lines 885–886)*, and results in *Appendix E.1 (starting at line 985)*.



### **3. Comparison Baselines**

**Concern:**
Reviewers requested more comprehensive comparisons with existing foundation models and multimodal methods. (xzsK: Weaknesses 2; iCa8: Weaknesses 4; EbNx: Weaknesses 1,5; GyVr: Weaknesses 2, 6.6)

**Our Addressing:**
We added comparisons with additional pre-trained slide encoders (FEATHER [1], CHIEF [2], TITAN [3]), multimodal fusion methods (MOTCat [4], PIBD [5]), and distillation-based missing-modality inference (DisPro [6]). Results show that MSRL with inter-case relevance achieves superior missing-modality performance and remains competitive with multimodal fusion methods.
Details are reported in *Section 4.2* and *Table 1–2 (starting at line 405)*.

---

### **Overall Resolution**

We have provided point-by-point responses to all reviewers in the following comments, and the corresponding revisions have been incorporated into the manuscript, highlighted in blue.

All concerns raised by the reviewers have been addressed.


Best regards,

Authors

[1] Shao, Daniel, et al. "Do Multiple Instance Learning Models Transfer?." ICML 2025.

[2] Wang, Xiyue, et al. "A pathology foundation model for cancer diagnosis and prognosis prediction." Nature 2024.

[3] Ding, Tong, et al. "A multimodal whole-slide foundation model for pathology." Nature Medicine 2025.

[4] Xu, Yingxue, and Hao Chen. "Multimodal optimal transport-based co-attention transformer with global structure consistency for survival prediction." CVPR 2023

[5] Zhang, Yilan, et al. "Prototypical information bottlenecking and disentangling for multimodal cancer survival prediction." ICLR 2024.

[6] Xu, Yingxue, et al. "Distilled Prompt Learning for Incomplete Multimodal Survival Prediction." CVPR 2025.

---

### Meta-Review · Area_Chair_oG4X · 2026-01-05

**Summary:**

This paper proposes MSRL, a framework for data-efficient precision oncology that can learn inter-case relevance. All reviewers found the method novel and appreciate the large amount of experimental support. However, there are quite a few major concerns. In my opinion, the authors did a good job during the rebuttal phase, considering that there are seven reviewers with tons of comments. The authors wrote very comprehensive responses, added a lot of new experiments and finally addressed most major concerns, although there remain some concerns. A brief summary below:

1. Presentation and clarify. Four reviewers and myself found the paper hard to read/follow. The authors promised to improve it and did make some major changes to the paper, but it's not easy to tell whether the reviewers are satisfied by the updated version without a discussion with the reviewers.
2. Generalization. Almost all reviewers had concerns about generalization as everything was done within TCGA. The authors expanded their experiments and I think this concern was addressed.
3. Comparison baselines. Most reviewers proposed some baselines and the authors did new experiments for comparison and showed the superior performance of the proposed MSRL method.
4. Ablation analysis. Most reviewers asked for ablation analysis, which was addressed by the authors additional experiments.
5. Buffer. A few reviewers have concerns about the buffer from different perspectives, including clarification of the buffer, motivation  and role of t, technical details, and buffer management. I think most are addressed but the management, where the authors claimed "We plan to refine the specific buffer maintenance method in future work".

It's really a boarder line paper and it was a tough decision for the AC. I strongly recommend the authors to take the constructive comments and revise it carefully for the camera ready version.

**Reviewer Concerns:**

Reviewer xzsK:
1. Slide encoders (addressed)
2. Generalization (addressed)
3. Presentation and clarify (partially addressed)

Reviewer iCa8:
1. Presentation and complication (partially addressed)
2. Unclear contribution (addressed)
3. Insufficient experiments to support "data efficiency" (partially addressed)
5. Significance of genomic data (addressed)

Reviewer EbNx:
1. Missing reference for knowledge distillation (partially addressed)
2. Information leakage and generalizability (addressed)
3. Low AUC (addressed)

Reviewer vXzx:
1. Cross-modal kNN (addressed)
2. Ablation (addressed)
3. The role of buffer (addressed)

Reviewer vuS4:
1. Scalability and buffer management (partially addressed)
2. Fairness of comparison and robustness (addressed)
3. Lack of interpretability analysis (addressed)
4. Presentation and clarify (partially addressed)

Reviewer GyVr
1. Presentation and clarify (not addressed)
2. Details in the buffer (partially addressed)
3. Comparison methods (partially addressed)
4. Data contamination and generalization (addressed)
5. Missing modalities (addressed)
6. Ablation study (addressed)

Reviewer sJW6:
1. Generalization (addressed)
2. Missing data setting (addressed)
3. Evaluation fairness (addressed)

**Reviewer Scores:**

Reviewer xzsK: 4 --> 5

Reviewer iCa8: 2 --> 4

Reviewer EbNx: 2 --> 4

Reviewer vXzx: 6 --> 6

Reviewer vuS4: 4 --> 5

Reviewer GyVr: 4--> 4

Reviewer sJW6: 6 --> 6

---

### Decision · Program_Chairs · 2026-01-26

Accept (Poster)